# Uniform Last-Iterate Guarantee for Bandits and Reinforcement Learning

**Junyan Liu**
University of Washington
junyanl1@cs.washington.edu

**Yunfan Li**
University of California, Los Angeles
yunfanli@g.ucla.edu

**Ruosong Wang**[*]
CFCS and School of Computer Science
Peking University
ruosongwang@pku.edu.cn

**Lin F. Yang**[*]
University of California, Los Angeles
linyang@ee.ucla.edu

## Abstract

Existing metrics for reinforcement learning (RL) such as regret, PAC bounds, or uniform-PAC [Dann et al., 2017], typically evaluate the *cumulative* performance, while allowing the agent to play an arbitrarily bad policy at any finite time $t$. Such a behavior can be highly detrimental in high-stakes applications. This paper introduces a stronger metric, uniform last-iterate (ULI) guarantee, capturing both cumulative and instantaneous performance of RL algorithms. Specifically, ULI characterizes the instantaneous performance by ensuring that the per-round suboptimality of the played policy is bounded by a function, monotonically decreasing w.r.t. round $t$, preventing revisiting bad policies when sufficient samples are available. We demonstrate that a near-optimal ULI guarantee directly implies near-optimal cumulative performance across aforementioned metrics, but not the other way around. To examine the achievability of ULI, we first provide two positive results for bandit problems with finite arms, showing that elimination-based algorithms and high-probability adversarial algorithms with stronger analysis or additional designs, can attain near-optimal ULI guarantees. We also provide a negative result, indicating that optimistic algorithms cannot achieve near-optimal ULI guarantee. Furthermore, we propose an efficient algorithm for linear bandits with *infinitely many arms*, which achieves the ULI guarantee, given access to an optimization oracle. Finally, we propose an algorithm that achieves near-optimal ULI guarantee for the online reinforcement learning setting.

## 1 Introduction

In online decision-making problems with bandit feedback, a learner sequentially interacts with an unknown environment: in each round, the learner plays an policy and then observes the corresponding rewards of the played policy. Typically, the goal of the learner is to achieve good *cumulative performance*, commonly measured by regret or probably approximately correct (PAC) bound. For instance, in the online advertisement scenario, the goal of the website (learner) could be maximizing the cumulative click numbers [Li et al., 2010]. Hence, the website aims to minimize the regret that measures the cumulative clicks of the recommended advertisement compared to that of the unknown optimal advertisement. In addition to regret minimization, the goal could also be quickly identifying popular advertisements [Chen et al., 2014, Jin et al., 2019]. To this end, a PAC bound

---

[*]Corresponding authors

38th Conference on Neural Information Processing Systems (NeurIPS 2024).

is suitable here to measure the sample complexity (i.e., cumulative time steps) that the algorithm needs to identify those popular advertisements. To reap the benefits of both measures, Dann et al. [2017] propose a new performance measure called uniform-PAC, ensuring that for all $\epsilon > 0$, the total number of $\epsilon$-suboptimal policies played by the algorithm is bounded by a function polynomial in $1/\epsilon$. The uniform-PAC bound can simultaneously imply a high-probability sublinear regret bound and a polynomial sample complexity for any desired accuracy.

Although uniform-PAC provides a powerful framework to unify regret[2] and PAC bound, it still fails to capture the *instantaneous performance* of the learning algorithm. In particular, a uniform-PAC algorithm could play a bad policy in some late but finite round $t$, even if it enjoys a good cumulative performance. This drawback impedes the application of uniform-PAC algorithms into high-stakes fields. Clinical trials, for example, place high demands on instantaneous performance for every treatment test, since patients need to be assigned with increasingly better treatments when more experimental data are available [Villar et al., 2015]. Hence, two natural questions arise:

1. *Can we find a new metric that characterizes not only the cumulative performance but also the instantaneous performance?*

2. *If such a metric exists, is it* optimally *achievable by some algorithm?*

In this paper, we answer both questions affirmatively. Our main contributions are summarized as follows.

- We introduce a new metric called *uniform last-iterate* (ULI), which simultaneously characterizes cumulative and instantaneous performance of sequential decision-making algorithms. On one hand, ULI can characterize the instantaneous performance: the per-round suboptimality of any algorithm with ULI guarantee is upper bounded by a function, monotonically decreasing for late time $t$. On the other hand, we show that any algorithm with a (near-optimal) ULI guarantee is also (near-optimally) uniform-PAC, demonstrating that ULI can imply cumulative performance.

- To answer the question whether ULI is achievable, we examine three common types of bandit algorithms in the finite arm setting. First, we provide a stronger analysis to show that many existing elimination-based algorithms indeed enjoy a near-optimal ULI guarantee. Then, we propose a meta-algorithm that enables any high-probability adversarial bandit algorithms, with a mild condition, to achieve a near-optimal ULI guarantee, and we show that such condition naturally holds for many adversarial bandit algorithms. Finally, we provide a hardness result showing that optimistic algorithms (e.g., lil'UCB [Jamieson et al., 2014]) cannot achieve near-optimal ULI guarantee. As lil'UCB is near-optimally uniform-PAC, our hardness result also implies that ULI is *strictly stronger* than uniform-PAC.

- For linear bandits with infinitely-many arms, we propose an oracle-efficient[3] linear bandit algorithm with the ULI guarantee (with access to an optimization oracle). In particular, we propose an adaptive barycentric spanner technique, selecting finitely many base arms that can linearly represent all (possibly infinitely many) well-behaved arms. This technique generalizes the one in [Awerbuch and Kleinberg, 2008] for elimination-based algorithms by adaptively identifying spaces that active arms span. Leveraging the phased elimination algorithm [Lattimore et al., 2020], our algorithm can conduct the elimination over all arms by only playing a finite subset of arms and querying a linearly-constrained optimization oracle for only a polynomial number of times.

- Finally, we propose a new algorithm for tabular episodic Markov decision processes (MDPs), which achieves a near-optimal ULI guarantee. In particular, our algorithm adapts uncertainty-driven reward functions to encourage exploration of the transition model, which ensures accurate estimations of value functions across all policies. The final ULI guarantee is achieved by conducting policy elimination.

**Related work.** In online decision-making problems, regret and PAC bounds are widely adopted to evaluate the cumulative performance of algorithms. More concretely, one line of research [Auer et al., 2002a,b, Abbasi-Yadkori et al., 2011, Li et al., 2010, Jin et al., 2018] aims to minimize the regret which measures the difference between the cumulative rewards of the selected policies and that of the

---

[2]Throughout the paper, we focus on high-probability regret, and therefore, when we mention regret, it always refer to high-probability regret. We refer readers to Remark 2.8 for a discussion on expected regret.

[3]A linear bandit algorithm is oracle-efficient if it calls an optimization oracle per-round for at most polynomial number of times.

best policy in hindsight. The PAC guarantees are more common than the regret when studying the pure-exploration/best policy identification problems [Even-Dar et al., 2006, Kalyanakrishnan et al., 2012, Wagenmaker et al., 2022]. One of the popular PAC measures is $(\delta, \epsilon)$-PAC which suggests that with probability at least $1 - \delta$, the algorithm can output a near-optimal policy at most $\epsilon$ away from the optimal one by using a sample complexity polynomial in $1/\epsilon$. Later, Dann et al. [2017] introduce a new framework called uniform-PAC to unify both metrics and develop a uniform-PAC algorithm for episodic Markov decision processes (MDPs). Subsequent works design uniform-PAC algorithms for MDPs with linear function approximation [He et al., 2021] and bounded Eluder dimension [Wu et al., 2023]. Though uniform-PAC strengthens regret and $(\delta, \epsilon)$-PAC bound, it still fails to characterize the instantaneous performance of online algorithms, i.e., a uniform-PAC algorithm, even with a good cumulative performance, can play bad policies for some late rounds.

A seemingly related performance measure is last-iterate convergence (LIC) which has been studied for *optimizing* MDPs [Moskovitz et al., 2023, Ding et al., 2023] and they use the primal-dual approach to formulate the problem of identifying an optimal policy in the constrained MDPs from a game-theoretic perspective. These works often require additional knowledge of the value functions and the LIC does not characterize the unknown dynamics of the environment. However, in our problem, the dynamics need to be *learned* as the algorithm sequentially interacts with the environment.

## 2 Preliminaries

### 2.1 Framework

We consider a general online sequential decision-making framework where a learner interacts with an environment with a fixed decision set. At each round $t \in \mathbb{N}$, the learner makes a decision from the set and observes the corresponding reward(s). In what follows, we instantiate this framework to multi-armed bandits, linear bandits, and tabular episodic Markov decision processes (MDPs).

**Multi-armed bandits.** In the stochastic MAB setting, the arm (decision) set follows that $\mathcal{A} = [K] \triangleq \{1, \ldots, K\}$. Each arm $a \in [K]$ is associated with a fixed and unknown $[0, 1]$-bounded distribution[4] such that $\forall t$, reward $X_{t,a}$ is an i.i.d. sample from this distribution with mean $\mu_a = \mathbb{E}[X_{t,a}]$. Let $A_t$ be the arm played at round $t$ and $\Delta_a = \mu^\star - \mu_a$ be the suboptimality gap where $\mu^\star = \max_{a \in [K]} \mu_a$.

**Linear bandits.** In the stochastic linear bandits setup, we assume that the arm set $\mathcal{A} \subseteq \mathbb{R}^d$ is compact. The reward of played arm $A_t$ at round $t$ follows that $X_{t,A_t} = \langle \theta, A_t \rangle + \eta_t$ where $\theta \in \mathbb{R}^d$ is a fixed but unknown parameter, and $\eta_t$ is conditionally 1-subgaussian. Let $\Delta_a = \sup_{b \in \mathcal{A}} \langle \theta, b - a \rangle$. We follow standard assumptions that $\|\theta\|_2 \leq 1$, $\|a\|_2 \leq 1$ for all $a \in \mathcal{A}$, and $\Delta_a \leq 1$ for all $a \in \mathcal{A}$.

**Tabular episodic MDPs.** A tabular episodic MDP is formalized as $\mathcal{M} = (\mathcal{S}, \mathcal{A}, H, r, P, \mu)$ where $\mathcal{S}, \mathcal{A}$ are finite state and action spaces with $|\mathcal{S}| = S, |\mathcal{A}| = A$, $H$ is the horizon length, $r = \{r_h\}_{h=1}^H$ where $r_h : \mathcal{S} \times \mathcal{A} \to [0, 1]$ is a known reward function, $\{P_h\}_{h \in [H]}$ where $P_h : \mathcal{S} \times \mathcal{A} \to \Delta(\mathcal{S})$ is a transition function, and $\mu$ is the initial state distribution. At the beginning of each episode $t$, the learner executes a policy $\pi_t = \{\pi_{t,h} : \mathcal{S} \to \Delta(\mathcal{A})\}_{h=1}^H$. Then, starting from the initial state $s_{t,1} \sim \mu$, for each stage $h \in [H]$, the learner repeatedly takes an action $a_{t,h} \sim \pi_{t,h}(s_{t,h})$, observes reward $r_h(s_{t,h}, a_{t,h})$, and transits to the next state $s_{t+1,h} \sim P_h(\cdot | s_{t,h}, a_{t,h})$.

For any policy $\pi$ and stage $h$, we define action value function $Q_h^\pi(s, a)$ and value function $V_h^\pi(s)$ as

$$Q_h^\pi(s, a) = \mathbb{E}\left[\sum_{h'=h}^H r_{h'}(s_{h'}, a_{h'}) \mid s_h = s, a_h = a, \pi\right], \quad V_h^\pi(s) = \mathbb{E}\left[\sum_{h'=h}^H r_{h'}(s_{h'}, a_{h'}) \mid s_h = s, \pi\right].$$

The optimal action value function and value function at each stage $h$ are denoted by $V_h^\star(s) = \max_\pi V_h^\pi(s)$, and $Q_h^\star(s, a) = \max_\pi Q_h^\pi(s, a)$ respectively. Let $\Delta_\pi = \mathbb{E}_{s_1 \sim \mu}[V_1^\star(s_1) - V_1^\pi(s_1)]$.

**Suboptimality notations.** For MAB and linear bandits settings, the *instantaneous suboptimality* is $\Delta_t = \Delta_{A_t}$, and for episodic MDP, $\Delta_t = \Delta_{\pi_t}$. We use $\Delta = \inf_{a \in \Pi : \Delta_a > 0} \Delta_a$ to denote the minimum suboptimality gap. Notice that it is possible that $\Delta = 0$ when, for example, arm set $\mathcal{A}$ is a ball.

---

[4]Note that all MAB and linear stochastic bandit algorithms in this paper also work for $R$-subgaussian noise with minor adjustments. Only adversarial bandits algorithms in Section 3.1 require the boundedness assumption.

## 2.2 Limitations of Existing Metrics

Regret and $(\delta, \epsilon)$-PAC are widely adopted to measure the performance. The regret is defined as:

**Definition 2.1** (Regret). *For each fixed $T \in \mathbb{N}$, the regret $R_T$ is defined as $R_T = \sum_{t=1}^{T} \Delta_t$.*

Let $N_\epsilon = \sum_{t=1}^{\infty} \mathbb{I}\{\Delta_t > \epsilon\}$ be the number of plays of policies whose suboptimality gap is greater than $\epsilon$. The definition of $(\delta, \epsilon)$-PAC is given as:

**Definition 2.2** ($(\delta, \epsilon)$-PAC). *For any fixed $\delta, \epsilon \in (0, 1)$, an algorithm is $(\delta, \epsilon)$-PAC (w.r.t. function $F_{PAC}$) if there exists a function $F_{PAC}(\delta, \epsilon)$ polynomial in $\log(\delta^{-1})$ and $\epsilon^{-1}$ such that*

$$\mathbb{P}\left(N_\epsilon \leq F_{PAC}(\delta, \epsilon)\right) \geq 1 - \delta.$$

As shown by Dann et al. [2017], both regret and $(\delta, \epsilon)$-PAC have limitations. Specifically, an algorithm with sublinear regret bound may play suboptimal policies infinitely often. For the algorithm with $(\delta, \epsilon)$-PAC guarantee, it may not converge to the optimal policy when feeding the algorithm with more samples. Therefore, such an algorithm would play those policies with suboptimality gap, e.g., $\epsilon/2$ infinitely often. Motivated by these limitations, Dann et al. [2017] introduce uniform-PAC as follows.

**Definition 2.3** (Uniform-PAC). *An algorithm is uniform-PAC for some fixed $\delta \in (0, 1)$ if there exists a function $F_{UPAC}(\delta, \epsilon)$ polynomial in $\log(1/\delta)$ and $\epsilon^{-1}$, such that*

$$\mathbb{P}\left(\forall \epsilon > 0 : N_\epsilon \leq F_{UPAC}(\delta, \epsilon)\right) \geq 1 - \delta.$$

We also call $F_{\text{UPAC}}(\delta, \epsilon)$ the sample complexity function. Uniform-PAC is a stronger metric than regret and $(\delta, \epsilon)$-PAC since it leads to the following implications.

**Theorem 2.4** (Theorem 3 in [Dann et al., 2017]). *If an algorithm is uniform-PAC for some $\delta$ with function $F_{UPAC}(\delta, \epsilon) = \widetilde{\mathcal{O}}\left(\alpha_1/\epsilon + \alpha_2/\epsilon^2\right)$[5], where $\alpha_1, \alpha_2 > 0$ are constant in $\epsilon$ and depend on $\log(1/\delta)$ and $K$ for MAB, $d$ for linear bandits, and $S, A, H$ for MDPs then, the algorithm guarantees:*

- $\mathbb{P}\left(\lim_{t \to +\infty} \Delta_t = 0\right) \geq 1 - \delta$;
- $(\delta, \epsilon)$-*PAC with $F_{PAC}(\delta, \epsilon) = F_{UPAC}(\delta, \epsilon)$ for all $\epsilon > 0$;*
- *With probability at least $1 - \delta$, for all $T \in \mathbb{N}$, $R_T = \widetilde{\mathcal{O}}\left(\sqrt{\alpha_2 T} + \alpha_1 + \alpha_2\right)$.*

**Limitations of Uniform-PAC.** According to Theorem 2.4, uniform-PAC can imply a long-term convergence (the first bullet) and good cumulative performance (the second and the third bullets), but it does not capture the convergence rate of $\Delta_t$ for each round $t$. In other words, even if an algorithm enjoys uniform-PAC, it could still play a significantly bad policy for some very large but finite $t$. This would lead to catastrophic consequences in safety-critical applications.

## 2.3 New Metric: Uniform Last-Iterate Guarantee

To address the aforementioned issue, we introduce a new metric, formally defined below.

**Definition 2.5** (Uniform last-iterate (ULI)). *An algorithm is ULI for some $\delta \in (0, 1)$ if there exists a positive-valued function $F_{ULI}(\cdot, \cdot)$, such that*

$$\mathbb{P}\left(\forall t \in \mathbb{N} : \Delta_t \leq F_{ULI}(\delta, t)\right) \geq 1 - \delta,$$

*where $F_{ULI}(\delta, t)$ is polynomial in $\log(1/\delta)$ and proportional to the product of power functions of $\log t$ and $1/t$ (e.g., $F_{ULI}(\delta, t) = \texttt{polylog}(1/\delta)(\log t)^{\kappa_1} t^{-\kappa_2}$ for some $\kappa_1, \kappa_2 \geq 0$).*

According to Definition 2.5, the instantaneous suboptimality of any algorithm with the ULI guarantee can be bounded by a function $F_{\text{ULI}}(\delta, t)$. Moreover, $F_{\text{ULI}}(\delta, t)$ decreases monotonically for large $t$ if its power on $1/t$ is strictly positive, which captures the convergence rate of $\Delta_t$.

Note that the convergence rate of $\Delta_t$ is mainly determined by the power on $1/t$ in $F_{\text{ULI}}$. Moreover, as we will show shortly, an algorithm with the ULI guarantee automatically has a small regret bound. We therefore have the following lower bound on $F_{\text{ULI}}$.

**Theorem 2.6.** *For any bandit algorithm that achieves ULI guarantee for some $\delta$ with function $F_{ULI}(\delta, t)$, there exists a MAB instance such that $F_{ULI}(\delta, t) = \Omega\left(t^{-1/2}\right)$.*

---

[5]We use $\widetilde{\mathcal{O}}(\cdot)$ to hide polylog factors.

We provide the proof in Appendix A. In the rest of the paper, we say an algorithm is near-optimal ULI if it achieves the ULI guarantee with $F_{\mathrm{ULI}}(\delta, t) = \widetilde{\mathcal{O}}(1/\sqrt{t})$.

Then, we present the following theorem to show that ULI directly leads to uniform-PAC, implying that ULI also characterizes the cumulative performance of bandit algorithms.

**Theorem 2.7.** *Suppose an algorithm achieves the ULI guarantee for some $\delta$ with function $F_{ULI}(\delta, t) = \mathtt{polylog}(t/\delta) \cdot t^{-\kappa}$ where $\kappa \in (0, 1)$. Then, we have,*

- *the algorithm is uniform-PAC with function $F_{UPAC}(\delta, \epsilon) = \mathcal{O}\big(\epsilon^{-\frac{1}{\kappa}} \cdot \mathtt{polylog}(\delta^{-1}\epsilon^{-1})\big)$.*
- *with probability at least $1 - \delta$, $\forall T \in \mathbb{N}$, the regret $R_T$ is bounded by*

$$\mathcal{O}\left(\min\left\{\mathtt{polylog}(T/\delta) \cdot T^{1-\kappa}, \Delta^{1-1/\kappa}\mathtt{polylog}^2\big((\delta\Delta)^{-1}\big)\right\}\right),$$

*when the minimum suboptimality gap of the input instance $\Delta$ satisfies $\Delta > 0$.*

According to Theorem 2.7, if an algorithm is with near-optimal ULI guarantee (i.e., $\kappa = \frac{1}{2}$), then it implies the near-optimality for uniform-PAC bound (the first bullet point) and anytime sublinear high-probability regret bound (the second bullet point). On the other hand, an algorithm with near-optimal uniform-PAC bound does not necessarily enjoy a near-optimal ULI guarantee as shown in Section 3.2. The proof of Theorem 2.7 can be found in Appendix B.

**Remark 2.8.** *Although a near-optimal ULI guarantee implies an anytime sublinear high-probability regret bound, it cannot give an anytime sublinear expected regret bound. This is because any algorithm with ULI guarantee is also uniform-PAC, but [Dann et al., 2017, Theorem 1] implies that no algorithm can be uniform-PAC and achieve anytime sublinear expected regret bound simultaneously.*

## 3 Achieving Near-Optimal ULI in Bandits with Finite Arm-Space

In this section, we answer the question whether ULI is achievable for bandit problems. To this end, we examine three common types of bandit algorithms, including elimination-based algorithms, optimistic algorithms, and high-probability adversarial algorithms, in the finite arm setting, i.e., $|\mathcal{A}| = K$.

### 3.1 Elimination Framework Achieving Near-Optimal ULI Guarantee

To examine whether the elimination-type algorithms can achieve the ULI guarantee, we first provide an elimination framework in Algorithm 1 that ensures the ULI guarantee, and we then show that most elimination-based algorithms fall into this framework. The following result shows that with a proper function $f$ and a positive constant $\beta$, such an elimination framework ensures the ULI guarantee.

**Theorem 3.1.** *For any given $\delta \in (0, 1)$, if there exists function $f(\delta, t) = t^{-\kappa}\mathtt{polylog}(t/\delta)$ for some $\kappa \in (0, 1)$ and $\exists \beta > 0$, such that with probability $1 - \delta$, Eq. (1) holds for all $t$, then algorithm is ULI with $F_{ULI}(\delta, t) = \mathcal{O}(f(\delta, t))$.*

---

**Algorithm 1** Elimination framework for ULI

**Input**: $\delta \in (0, 1)$, set $\mathcal{A}$, function $f(\cdot, \cdot)$, and constant $\beta$.
**Initialize**: active arm set $\mathcal{A}_0 = \mathcal{A}$.
**for** $t = 1, 2, \ldots$ **do**
    Select an active set $\mathcal{A}_t$ based on available observations as ($a^*$ is one of optimal arms)

$$\mathcal{A}_t \subseteq \{a \in \mathcal{A}_{t-1} : \Delta_a \leq \beta \cdot f(\delta, t)\} \cup \{a^\star\}. \tag{1}$$

    Play an arm $A_t \in \mathcal{A}_t$ and observe reward $X_{t, A_t}$.

---

Theorem 3.1 suggests that Eq. (1) is a sufficient condition for elimination-based algorithms to achieve the ULI guarantee. Now, we show that existing elimination algorithms indeed fall into such a framework. We here consider successive elimination (SE) and phased elimination (PE). Notice that we consider SE only for the MAB setting (called SE-MAB, e.g., Algorithm 3 in [Even-Dar et al., 2006]) but PE for both the MAB setting (called PE-MAB, e.g., exercise 6.8 in [Lattimore and Szepesvári, 2020]) and the linear bandit setting (called PE-L, e.g., Algorithm 12 of Chapter 22 in [Lattimore and Szepesvári, 2020]). Since those algorithms are standard, we defer their pseudocodes

to Appendix D. Given Theorem 3.1, the following results show that the elimination framework can be instantiated by these algorithms with proper functions and therefore they achieve the ULI.

**Theorem 3.2.** *For any fixed $\delta \in (0, 1)$, elimination framework in Algorithm 1 can be instantiated by*

- *SE-MAB for MAB to achieve the ULI with $F_{ULI}(\delta, t) = \mathcal{O}\big(t^{-\frac{1}{2}}\sqrt{K\log(\delta^{-1}Kt)}\big)$.*
- *PE-MAB for MAB to achieve the ULI with $F_{ULI}(\delta, t) = \mathcal{O}\big(t^{-\frac{1}{2}}\sqrt{K\log(\delta^{-1}K\log(t+1))}\big)$.*
- *PE-L for linear bandits to achieve ULI with $F_{ULI}(\delta, t) = \mathcal{O}\big(t^{-\frac{1}{2}}\sqrt{d\log(\delta^{-1}K\log(t+1)\log d)}\big)$.*

**Achieving ULI by adversarial bandit algorithms.** Traditional elimination-based algorithms, including the ones mentioned above, typically require a carefully designed exploration strategy which is non-trivial even for linear bandits. Here, we provide an alternative way to achieve ULI by employing adversarial bandit algorithms to explore and then conduct the elimination. As shown in Appendix F, all adversarial bandit algorithms for both MAB and linear bandits that meet a certain condition can naturally be used to achieve the ULI guarantees similar to those of traditional elimination-based algorithms.

### 3.2 Lower Bound for Optimistic Algorithms

In this section, we present a lower bound to show that optimistic algorithms cannot achieve near-optimal ULI guarantee. The procedure of optimistic algorithms is summarized as follows. After playing each arm once, at each round $t$, the algorithm plays an arm $A_t$ that satisfies

$$A_t \in \underset{a \in \mathcal{A}}{\operatorname{argmax}} \left\{\widehat{\mu}_a(N_a(t)) + U_\delta(N_a(t))\right\}, \tag{2}$$

where $N_a(t)$ is the number of plays of arm $a$ before round $t$, $\widehat{\mu}_a(N_a(t)) = \frac{\sum_{s=1}^{t-1} X_{s, A_s}\mathbb{I}\{A_s=a\}}{N_a(t)}$ is the empirical mean of arm $a$ after $N_a(t)$ times play, and $U_\delta(N_a(t))$ is a positive bonus function which encourages the exploration.

We first consider optimistic algorithms e.g., upper confidence bound (UCB) [Lattimore and Szepesvári, 2020, Algorithm 3, Chapter 7], which enjoy (near)-optimal regret bounds. This type of algorithms typically uses the bonus function in the form of $\sqrt{\log(2t^2/\delta)/N_a(t)}$[6]. However, the $\log t$ term forces the algorithm to play suboptimal arms infinitely often, and thus they cannot achieve the ULI guarantee (refer to Appendix E.3 for a detailed proof). Similarly, other variants [Audibert and Bubeck, 2010, Degenne and Perchet, 2016] with $\log t$ term in bonus function, also cannot achieve the ULI guarantee.

We then consider another optimistic-type algorithm, lil'UCB [Jamieson et al., 2014] which obtains the order-optimal instance-dependent sample complexity and avoids $\log(t)$ term in $U_\delta(N_a(t))$. The bonus function of lil'UCB is as $\sqrt{\log\big(\delta^{-1}\log_+(N_a(t))\big)/N_a(t)}$ where $\log_+(x) = \log(\max\{x, e\})$. Our main result for lil'UCB is presented as follows. The full analysis of lil'UCB is deferred to Appendix E.

**Theorem 3.3.** *There exists a constant $\alpha \in (0, 1)$ that for all $\Delta \in (0, \alpha)$, running lil'UCB on the two-armed bandit instance with deterministic rewards and arm gap $\Delta$ gives $\exists t = \Omega\big(\Delta^{-2}\big)$ such that*

$$\Delta_t = \Omega\left(t^{-\frac{1}{4}}\sqrt{\log\log\big(\Delta^{-2}\log(\delta^{-1})\big) + \log(\delta^{-1})}\right).$$

Theorem 3.3 shows that lil'UCB is not near-optimal ULI, but it is unclear whether it can achieve the ULI guarantee. Recall from Theorem 3.2 that the elimination-based algorithms ensure that with high probability, $\forall t$, $\Delta_t = \widetilde{\mathcal{O}}\big(t^{-1/2}\big)$. Theorem 3.3 suggests that the convergence rate of lil'UCB is strictly worse than that of elimination-based algorithms, even if it enjoys a near-optimal cumulative performance [Jamieson et al., 2014]. In fact, this lower bound holds for all optimistic algorithms as long as the bonus function is in a similar form.

**Remark 3.4** (**ULI is strictly stronger than uniform-PAC**). *For lil'UCB, the number of times playing suboptimal arms is finite with an order-optimal instance-dependent sample complexity, which implies that lil'UCB is near-optimal uniform-PAC. Therefore, Theorem 3.3 also shows that an algorithm with near-optimal uniform-PAC does not necessarily enjoy near-optimal ULI guarantee.*

---

[6]We slightly adjust the bonus function to ensure that with probability at least $1 - \delta$, the confidence bound holds for all $t \in \mathbb{N}$.

---

**Algorithm 2** PE with adaptive barycentric spanner

---

**Input**: Compact arm set $\mathcal{A}$, confidence $\delta \in (0, 1)$, and constant $C > 1$.

**Initialize**: $\widehat{\theta}_1 = \{0, \ldots, 0\} \in \mathbb{R}^d$, $T_0 = 1$ and $\mathcal{B}_0 = \{e_1, \ldots, e_d\}$.

1  **for** $m = 1, 2, \ldots$ **do**

2     Set $T_m = 256 C^4 \cdot \frac{d^3}{4^{-m}} \log\left(\delta^{-1} d^3 4^m\right)$.

3     Invoke Algorithm 12 with $(\mathcal{A}, m, \mathcal{B}_{m-1}, T_m, C, \widehat{\theta}_m)$ to find a $C$-approximate barycentric spanner $\mathcal{B}_m$ for active arm set $\mathcal{A}_m$ where $\mathcal{A}_m$ is in Eq. (3).

4     Set $\pi_m(a) = \frac{1}{d}$ for each $a \in \mathcal{B}_m$.

5     Play each arm $a \in \mathcal{B}_m$ for $n_m(a) = \lceil T_m \pi_m(a) \rceil$ times.

6     Compute $V_m = I + \sum_{a \in \mathcal{B}_m} n_m(a) a a^\top$ and $\widehat{\theta}_{m+1} = V_m^{-1} \sum_{t \in \mathcal{T}_m} A_t X_{t, A_t}$ where $\mathcal{T}_m$ is a set that contains all rounds in phase $m$.

---

## 4   Achieving Near-Optimal ULI for Linear Bandits in Large Arm-Space

In this section, we propose a linear bandit algorithm that can handle the infinite number of arms. The compact arm set $\mathcal{A}$ is assumed to span $\mathbb{R}^d$ and $d$ is known.

### 4.1   Main Algorithm and Main Results

The starting point of our algorithm design is the phased elimination (PE) algorithm [Lattimore et al., 2020, Algorithm 12, Chapter 22]. However, PE in general is not feasible when the arm space is large (e.g., a continuous space). In this section, we present a carefully-designed algorithm to address the new challenges from large arm-spaces.

**Issues of PE for large arm-space.** PE needs to (i) compute (approximately) $G$-optimal design whose complexity scales linearly with $|\mathcal{A}|$ and (ii) compare the empirical mean of each arm, both of which are impossible when the arm set is infinite, e.g., $\mathcal{A}$ is a ball. A natural idea is to discretize $\mathcal{A}$, e.g., constructing a $\epsilon$-net, but the computational complexity has an exponential dependence on $d$, and the optimal arm does not necessarily lie in the net, which prevents the convergence to the optimal arm.

**High-level idea behind our solution.** To address the aforementioned issues, we propose an oracle-efficient linear bandit algorithm in Algorithm 2 which can eliminate bad arms by efficiently querying an optimization oracle. Our algorithm equips PE with a newly-developed *adaptive barycentric spanner* technique. The proposed technique selects a finite *representative arm set* to represent (possibly infinite) active arm set and adaptively adjusts the selection of arms across phases. By conducting the (approximate) $G$-optimal design [Kiefer and Wolfowitz, 1960] on the representative arm set and playing each arm in the set according to the design, the algorithm can acquire accurate estimations uniformly over all active arms. Moreover, the adaptive barycentric spanner approach can be implemented by efficiently querying an optimization oracle in polynomial times.

The definition of barycentric spanner [Awerbuch and Kleinberg, 2008] is presented as follows.

**Definition 4.1** ($C$-approximate barycentric spanner). *Let* $\mathcal{A} \subseteq \mathbb{R}^d$ *be a compact set. The set* $\mathcal{B} = [b_1, \ldots, b_d] \subseteq \mathcal{A}$ *is a $C$-approximate barycentric spanner for* $\mathcal{A}$ *if each* $a \in \mathcal{A}$ *can be expressed as a linear combination of points in* $\mathcal{B}$ *with coefficients in the range of* $[-C, C]$.

**Why adaptive barycentric spanner?** The primary reason that the non-adaptive barycentric spanner technique [Awerbuch and Kleinberg, 2008] fails to work for PE is that it requires the knowledge of the space that the active arm set spans. However, acquiring such knowledge is often difficult because the active arm set, which may span a *proper linear subspace* of $\mathbb{R}^d$, varies for different phases and the number of active arms could be infinite. Our algorithm shown in Algorithm 12 (whose pseudocode is deferred to Appendix G.2 due to space limit) can identify a barycentric spanner for active arm set adaptively for each phase, even if they do not span $\mathbb{R}^d$.

**Algorithm procedure.** Our algorithm proceeds in phases $m = 1, 2, \ldots$, and each phase consists of consecutive rounds. At the beginning of phase $m$, the algorithm invokes subroutine Algorithm 12 to identify a $C$-approximate barycentric spanner $\mathcal{B}_m$ which can linearly represent all arms in active arm set $\mathcal{A}_m$ where $\mathcal{A}_1 = \mathcal{A}$ and $\forall m \geq 2$

$$\mathcal{A}_m = \left\{ a \in \mathcal{A}_{m-1} : \left\langle \widehat{\theta}_m, a_m^\star - a \right\rangle \leq 2^{-m+1} \right\}, \tag{3}$$

---

**Algorithm 3** Tabular Episodic MDPs with ULI guarantee

---

**Input**: $\delta \in (0,1)$, set $\Pi_{\texttt{all}}$ containing all deterministic policies, absolute constant $c_1, c_2 > 0$.

**Initialize**: $\Pi_1 = \Pi_{\texttt{all}}$.

**for** $m = 1, 2, \ldots$ **do**

    Set $\delta_m = \delta/(2m^2)$ and duration $T_m = \left\lceil c_1 2^{2m} S^2 A H^4 \log^2 \left( c_2 2^{2m} S^2 A H^4 |\Pi_{\texttt{all}}|/\delta_m \right) \right\rceil$.

    Run Algorithm 4 with input $(\delta_m, \Pi_m, T_m)$ to obtain $\{\widetilde{V}_m^\pi\}_{\pi \in \Pi_m}$.

    Update active policy set $\Pi_{m+1} = \left\{ \pi \in \Pi_m : \max_{\pi' \in \Pi_m} \widetilde{V}_m^{\pi'} - \widetilde{V}_m^\pi \leq 2^{-m} \right\}$.

---

where $a_m^\star$ is the empirical best arm and $\widehat{\theta}_m$ is the estimation of unknown parameter $\theta$.

Then, the algorithm assigns $\pi_m(a) = \frac{1}{d}$ for each $a \in \mathcal{B}_m$. In fact, if we add $\cup_{i \in \mathcal{I}_m} \frac{e_i}{\sqrt{T_m}}$ (refer Appendix G for $e_i$ and $\mathcal{I}_m$) back to $\mathcal{B}_m$ and denote the new set by $\mathcal{B}_m'$, then $\pi_m : \mathcal{B}_m' \to \frac{1}{d}$ forms an optimal design. It is noteworthy that our algorithm only plays arms in $\mathcal{B}_m$ as these added elements do not necessarily exist in the arm set $\mathcal{A}_m$. As each added element $\frac{e_i}{\sqrt{T_m}}$ is close to zero, even if we only play arms in $\mathcal{B}_m$, we can still acquire accurate estimations uniformly over $\mathcal{A}_m$ (refer to Appendix G.6 for details):

$$\forall a \in \mathcal{A}_m : \quad \|a\|_{V_m^{-1}} = \sqrt{a^\top V_m^{-1} a} \leq C \cdot d/\sqrt{T_m}, \tag{4}$$

where $V_m = I + \sum_{a \in \mathcal{B}_m} n_m(a) a a^\top$ is the least squares matrix, used to estimate $\theta$.

According to the standard analysis of linear bandit algorithms, the estimation error of $\langle a, \theta \rangle$ is proportional to $\|a\|_{V_m^{-1}}$. Hence, the estimation errors of $\{\langle a, \theta \rangle\}_{a \in \mathcal{A}_m}$ can be uniformly bounded by $\widetilde{\mathcal{O}}\left(Cd/\sqrt{T_{m-1}}\right)$, which is known to the learner. Finally, after playing each arm $a \in \mathcal{B}_m$ for $n_m(a)$ times, the algorithm updates the empirical estimates $V_m$ and $\widehat{\theta}_{m+1}$, and then steps into the next phase.

The main results of Algorithm 2 for achieving the ULI guarantee and computational complexity are given as follows. The full proof can be found in Appendix G.

**Theorem 4.2.** *For any fixed $\delta \in (0,1)$, Algorithm 2 achieves the ULI guarantee with function $F_{ULI}(\delta, t) = \mathcal{O}\left(t^{-\frac{1}{2}} \sqrt{d^3 \log(\delta^{-1} dt)}\right)$. Moreover, in each phase, the number of calls to the optimization oracle given in Definition G.2 is $\mathcal{O}\left(d^3 \log_C d\right)$.*

Theorem 4.2 and Theorem 2.7 jointly suggest $\widetilde{\mathcal{O}}(d^{3/2}\sqrt{T})$ worst-case regret bound for the infinite-armed setting, which matches those of [Dani et al., 2008, Agrawal and Goyal, 2013, Hanna et al., 2023]. Compared with the lower bound $\Omega(d\sqrt{T})$ given by Dani et al. [2008], our regret bound suffers an extra $\sqrt{d}$ factor, caused by the spanner technique. Yet, it remains open to find a computationally efficient linear bandit algorithm that can handle the infinite arm setting with general compact $\mathcal{A}$ and matches the $\Omega(d\sqrt{T})$ lower bound.

Theorem 4.2 also shows that, for each phase, the spanner can be constructed by calling the oracle for only polynomial times. Compared to the computational efficiency of the algorithm in [Awerbuch and Kleinberg, 2008], the efficiency of our algorithm is only $d$ times worse than theirs.

## 5 Achieving Near-Optimal ULI in Tabular Episodic MDPs

In this section, we propose a novel algorithm that achieves a near-optimal ULI guarantee in tabular episodic MDPs setup. The algorithm is formally presented in Algorithm 3.

**High-level idea.** Algorithm 3 conducts policy elimination over a policy set $\Pi_{\texttt{all}}$ which enumerates all deterministic policies. The key challenge here is to acquire accurate estimations of value functions uniformly for all deterministic policies. Note that naïvely playing all deterministic policies will incur linear dependence on $|\Pi_{\texttt{all}}| = A^{SH}$ in $F_{\text{ULI}}$, which is exponentially large. Hence, the algorithm invokes subroutine Algorithm 4 to exhaustively explore the environment, which ensures accurate estimations of the transition model and only suffers *logarithmic dependence* on $|\Pi_{\texttt{all}}|$. Once the transition model can be well-approximated, the algorithm constructs an accurate estimation of the value function for every policy and then decides which policies should be eliminated.

---

**Algorithm 4** Uniform estimation for value functions

---

**Input**: confidence $\delta \in (0,1)$, policy set $\Pi \subseteq \Pi_{\mathtt{all}}$, duration $T$.
**Initialize**: randomly pick a policy $\pi_1 \in \Pi$, $N_{1,h}(s,a) = N_{1,h}(s,a,s') = 0$ for all $(h,s,a,s')$.
**for** $t = 1, \ldots, T$ **do**
    Observe initial state $s_{t,1} \sim \mu$.
    **for** $h = 1, \ldots, H$ **do**
        Take action $a_{t,h} = \pi_{t,h}(s_{t,h})$ and observe $s_{t,h+1} \sim \mathbb{P}_h(\cdot|s_{t,h}, a_{t,h})$.
        Increase counters $N_{t,h}(s_{t,h}, a_{t,h}, s_{t,h+1}) \overset{+}{\leftarrow} 1$ and $N_{t,h}(s_{t,h}, a_{t,h}) \overset{+}{\leftarrow} 1$.
    Update estimates $\widehat{\mathbb{P}}_{t,h}(s'|s,a) = \frac{N_{t,h}(s,a,s')}{\max\{1, N_{t,h}(s,a)\}}$ for all $(s,a,s') \in \mathcal{S} \times \mathcal{A} \times \mathcal{S}$.
    Update bonus function $b_t = \{b_{t,h}\}_{h \in [H]}$ where $b_{t,h}(\cdot, \cdot)$ is updated according to Eq. (5).
    Get $\{\widehat{V}_{t,1}^{\pi}(s_1)\}_{\pi \in \Pi}$ by invoking Algorithm 5 with input $(\Pi, b_t/H, \{\widehat{\mathbb{P}}_{T,h}\}_{h \in [H]}, b_t, s_{t,1})$.
    Update policy $\pi_{t+1} = \mathrm{argmax}_{\pi \in \Pi} \widehat{V}_{t,1}^{\pi}(s_1)$.
**for** $t = 1, \ldots, T$ **do**
    Get $\{\widehat{V}_{T,1}^{\pi}(s_{t,1})\}_{\pi \in \Pi}$ by invoking Algorithm 5 with input $(\Pi, r, \{\widehat{\mathbb{P}}_{T,h}\}_{h \in [H]}, 0, s_{t,1})$.
**Output**: $\{\widetilde{V}^{\pi}\}_{\pi \in \Pi}$ where $\widetilde{V}^{\pi} = \frac{1}{T} \sum_{t=1}^{T} \widehat{V}_{T,1}^{\pi}(s_{t,1})$.

---

---

**Algorithm 5** Construct estimated value function

---

**Input**: policy set $\Pi \subseteq \Pi_{\mathtt{all}}$, reward function $r$, transition $\widehat{\mathbb{P}}$, bonus function $b$, initial state $s_1$.
**for** $\pi \in \Pi$ **do**
    $\widehat{Q}_{H+1}^{\pi}(\cdot, \cdot) = 0$ and $\widehat{V}_{H+1}^{\pi}(\cdot) = 0$.
    **for** $h = H, H-1, \ldots, 1$ **do**
        $\widehat{Q}_h^{\pi}(\cdot, \cdot) = \min\left\{\left[\widehat{\mathbb{P}}_h \widehat{V}_{h+1}^{\pi}\right](\cdot, \cdot) + r_h(\cdot, \cdot) + b_h(\cdot, \cdot), H\right\}$ and $\widehat{V}_h^{\pi}(\cdot) = \widehat{Q}_h^{\pi}(\cdot, \pi(\cdot))$.
**Output**: $\{\widehat{V}_1^{\pi}(s_1)\}_{\pi \in \Pi}$.

---

More concretely, Algorithm 3 first accepts a set of all deterministic policies and then it proceeds in phases $m = 1, 2, \ldots$. In each phase $m$, subroutine Algorithm 4 is invoked to learn the transition model. Specifically, the subroutine inherits the structure of UCB-VI [Azar et al., 2017], but more importantly the algorithm pretends to be agnostic to the reward function and uses uncertainty-driven reward functions $\{b_{t,h}(s,a)/H\}_{t,h}$ where

$$b_{t,h}(s,a) = H\sqrt{\frac{2S \log \iota}{\max\{1, N_{t,h}(s,a)\}}} + \frac{2HS \log \iota}{3 \max\{1, N_{t,h}(s,a)\}} \quad \text{where} \quad \iota = \frac{10SAH|\Pi_{\mathtt{all}}|T}{\delta}, \quad (5)$$

where $N_{t,h}(s,a)$ is the number of times of visiting $(s,a)$ at stage $h$ up to episode $t$. Note that Eq. (5) captures the uncertainty of visitation a state-action pair, i.e., more visitations of $(s,a)$, larger $N_{t,h}(s,a)$, and less uncertainty. This modification, inspired by [Wang et al., 2020] forces the algorithm to aggressively explore the environment, in the sense that the algorithm prefers to play a policy that maximizes the uncertainty.

The following theorem shows that our algorithm achieves the ULI guarantee.

**Theorem 5.1.** *For any fixed $\delta \in (0,1)$, Algorithm 3 achieves the ULI guarantee with function $F_{ULI}(\delta, t) = \mathcal{O}\left(t^{-\frac{1}{2}}\sqrt{S^3 A H^5} \log\left(tSAH/\delta\right)\right)$.*

Theorem 5.1 shows near-optimality of Algorithm 3 w.r.t. episode $t$. Unfortunately, the regret bound implied by our ULI result incurs suboptimal dependence on $S, H$, and the logarithmic term, and our algorithm is not computational efficient. We leave these improvements for future work.

## 6 Conclusions and Future Work

In this paper, we propose a new metric, the uniform last-iterate (ULI) guarantee, which captures both instantaneous and cumulative performance of sequential decision-making algorithms. To answer

whether ULI is (optimally) achievable, we first examine three types of bandit algorithms in the finite-arm setting. Specifically, we provide *stronger* analysis to show that elimination-based algorithms naturally achieve near-optimal ULI guarantees. We also provide a reduction-based approach to enable any high-probability adversarial algorithms, with a mild condition, to achieve near-optimal ULI guarantees. We further provide a negative result for optimistic bandit algorithms showing that they cannot achieve near-optimal ULI guarantee. Furthermore, in the large arm space setting, we propose an oracle-efficient linear bandit algorithm, equipped with the novel adaptive barycentric spanner technique. Finally, we propose a new algorithm, which adapts uncertainty-driven reward functions into policy elimination to achieve a ULI guarantee in tabular episodic MDPs.

Some natural directions are to improve our current results, including proposing a ULI algorithm for linear bandits with infinite arms and proposing a computationally efficient algorithm, possibly based on the action-elimination approach, for tabular MDPs. We also summarize other interesting directions as follows.

- Design (computationally efficient) algorithms for MDPs with linear function approximation. The main challenge is to bypass any dependence on the number of states which is possibly infinite. Thus, generalizing our RL algorithm, which enumerates all deterministic policies, to linear MDPs does not work.

- Design (computationally efficient) algorithms for Episodic MDPs with only logarithmic dependence on $H$ (a.k.a. horizon-free). In our attempts, we use the doubling trick on $\epsilon$ in existing $(\delta, \epsilon)$-PAC horizon-free RL algorithms. In this case, we run the algorithm in phases with $\epsilon, \epsilon/2, \epsilon/4, \ldots$. The main difficulty is to leverage the information learned from the previous phase to guide the algorithm to play an improved policy at the next phase as required by ULI. Thus, we conjecture that there might exist fundamental barriers to simultaneously achieve ULI guarantee and a logarithmic dependence on $H$.

- It could be also interesting to investigate some empirical issues when deploying ULI algorithms in high-stakes fields. Typically, optimistic algorithms initially incur a lower regret than ULI algorithms, but their regret will exceed those of ULI algorithms at a certain juncture. At this juncture, ULI algorithms have eliminated all suboptimal arms, whereas optimistic algorithms continue exploring as time evolves. Hence, identifying this turning point could be beneficial for deploying ULI algorithms in high-stakes domains.

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

# Appendix

# A  Proof of Theorem 2.6

We prove this claim by contradiction. Suppose that there exists an algorithm that can achieve the ULI guarantee with function $F_{\mathrm{ULI}}(\delta, t) \leq c_0 \frac{1}{t^{\frac{1}{2}+\alpha}}$ for some $\alpha \in (0, \frac{1}{2})$ and constant $c_0 > 0$ which may depend on $\log(1/\delta)$ and $K$ for MAB or $d$ for linear bandits. We consider a $K$-armed bandit instance ($K \geq 2$) with Gaussian rewards and all suboptimality gaps are bounded by one.

By the definition of ULI, we have that with probability at least $1 - \delta$, for all $T \in \mathbb{N}$ with $T^{\frac{1}{2}-\alpha} > t_0$, we have

$$
\begin{aligned}
R_T &= \sum_{t=1}^{T} \Delta_t \\
&= \sum_{t=1}^{t_0} \Delta_t + \sum_{t=t_0+1}^{T} \Delta_t \\
&\leq t_0 + \sum_{t=t_0+1}^{T} F_{\mathrm{ULI}}(\delta, t) \\
&\leq t_0 + c_0 \sum_{t=t_0+1}^{T} t^{-\alpha-\frac{1}{2}} \\
&\leq T^{\frac{1}{2}-\alpha} + \frac{2c_0}{1-2\alpha} T^{\frac{1}{2}-\alpha},
\end{aligned}
$$

where the first inequality holds as we assume all suboptimal gaps are bounded by one, and $\Delta_t \leq F_{\mathrm{ULI}}(\delta, t)$ for all $t \geq t_0$.

Since $\frac{1}{2} - \alpha \in (0, \frac{1}{2})$, there will be a contradiction to the regret lower bound [Lattimore and Szepesvári, 2020, Corollary 17.3] for the Gaussian MAB setup.

# B  Proof of Theorem 2.7

**Theorem B.1** (Restatement of Theorem 2.7). *Suppose an algorithm achieves the ULI guarantee for some $\delta$ with function $F_{ULI}(\delta, t) = \mathtt{polylog}(t/\delta) \cdot t^{-\kappa}$ where $\kappa \in (0, 1)$. Then, we have,*

- *the algorithm is also uniform-PAC with*

$$
F_{UPAC}(\delta, \epsilon) = \mathcal{O}\left( \frac{\mathtt{polylog}(\delta^{-1}\epsilon^{-1})}{\epsilon^{1/\kappa}} \right).
$$

- *with probability at least $1 - \delta$, $\forall T \in \mathbb{N}$, the regret $R_T$ is bounded by*

$$
\mathcal{O}\left( \min\left\{ \mathtt{polylog}(T/\delta) \cdot T^{1-\kappa}, \frac{\mathtt{polylog}^2\left(\frac{1}{\delta\Delta}\right)}{\Delta^{1/\kappa-1}} \right\} \right),
$$

*when the minimum suboptimality gap of the input instance $\Delta$ satisfies $\Delta > 0$.*

## B.1  Proof of the first bullet: ULI guarantee implies uniform-PAC bound

In this subsection, we show the first bullet of Theorem 2.7. We define the following events for $F_{\mathrm{ULI}} = \mathtt{polylog}(t/\delta) \cdot t^{-\kappa}$:

$$
A = \{ \forall t \in \mathbb{N} : \Delta_t \leq F_{\mathrm{ULI}}(\delta, t) \}, \tag{6}
$$

which indicates the occurrence the ULI, and denote the event $B$ as:

$$
B = \left\{ \forall \epsilon > 0 : \sum_{t=1}^{\infty} \mathbb{I}\{\Delta_t > \epsilon\} \leq F_{\mathrm{UPAC}}(\delta, \epsilon) \right\}.
$$

For some $\delta \in (0,1)$, if event $A$ holds for an algorithm with probability at least $1 - \delta$, then the algorithm is with ULI guarantee. Therefore, to prove the claimed result, it suffices to show that conditioning on event $A$ that for any algorithm enjoys ULI guarantee with a function $F_{\mathrm{ULI}}(\delta, t)$, one can find a function $F_{\mathrm{UPAC}}(\delta, \epsilon)$ such that event $B$ also holds.

As for any fixed $\delta \in (0,1)$, $F_{\mathrm{ULI}}(\delta, t) = \texttt{polylog}(t/\delta) \cdot t^{-\kappa}$ is monotonically decreasing for large $t$, $\exists U \in \mathbb{R}_{>0}$ such that $F_{\mathrm{ULI}}(\delta, t) \leq U$ for all $t \in \mathbb{N}$ and $F_{\mathrm{ULI}}(\delta, t) = U$ is attainable by some $t$. Then, we consider any given $\epsilon > 0$ with two cases as follows.

**Case 1:** $\epsilon < U$. Again, by the fact that $F_{\mathrm{ULI}}(\delta, t) = \texttt{polylog}(t/\delta) \cdot t^{-\kappa}$ is monotonically decreasing for large $t$, there should exist a round $t \in \mathbb{N}$ such that $F_{\mathrm{ULI}}(\delta, s) \leq \epsilon$ for all $s \geq t$. Let $t_0$ be the round with the maximal index such that $F_{\mathrm{ULI}}(\delta, t_0) > \epsilon$, which gives

$$t_0 \leq \frac{\texttt{polylog}^{\frac{1}{\kappa}}(t_0/\delta)}{\epsilon^{\frac{1}{\kappa}}}.$$

As a result, the number of times that $\Delta_t > \epsilon$ occurs is at most

$$\sum_{t=1}^{\infty} \mathbb{I}\{\Delta_t > \epsilon\} \leq \sum_{t=1}^{\infty} \mathbb{I}\{F_{\mathrm{ULI}}(\delta, t) > \epsilon\} \leq t_0 \leq \frac{\texttt{polylog}^{\frac{1}{\kappa}}(t_0/\delta)}{\epsilon^{\frac{1}{\kappa}}}.$$

Then, one can find constants $c_0 > 0$ and $z \in (0, \kappa)$ such that $\texttt{polylog}(t/\delta) \leq c_0 \cdot \texttt{polylog}(1/\delta)t^z$ for all $t \in \mathbb{R}_{\geq 1}$, thereby $\texttt{polylog}(t_0/\delta) \leq c_0 \cdot \texttt{polylog}(1/\delta)t_0^z$. Combining $\texttt{polylog}(t_0/\delta) \leq c_0 \cdot \texttt{polylog}(1/\delta)t_0^z$ and $F_{\mathrm{ULI}}(\delta, t_0) > \epsilon$ gives

$$\epsilon \leq \frac{\texttt{polylog}(t_0/\delta)}{t_0^{\kappa}} \leq \frac{c_0 \cdot \texttt{polylog}(1/\delta)t_0^z}{t_0^{\kappa}} = c_0 \cdot \texttt{polylog}(1/\delta)t_0^{z-\kappa},$$

which immediately leads to

$$t_0 \leq \left( \frac{c_0 \cdot \texttt{polylog}(1/\delta)}{\epsilon} \right)^{\frac{1}{\kappa - z}}. \tag{7}$$

Then, we can further show that

$$\sum_{t=1}^{\infty} \mathbb{I}\{\Delta_t > \epsilon\} \leq t_0 \leq \frac{\texttt{polylog}^{\frac{1}{\kappa}}(t_0/\delta)}{\epsilon^{\frac{1}{\kappa}}} \leq \frac{\texttt{polylog}\left( \delta^{-1} \left( \frac{c_0 \cdot \texttt{polylog}(1/\delta)}{\epsilon} \right)^{\frac{1}{\kappa - z}} \right)}{\epsilon^{\frac{1}{\kappa}}} \triangleq F_{\mathrm{UPAC}}(\delta, \epsilon), \tag{8}$$

where the third inequality follows that $\texttt{polylog}(t/\delta)$ is monotonically increasing for all $t > 0$ and Eq. (7).

**Case 2:** $\epsilon \geq U$. Conditioning on event $A$ and by the definition of $U$, we have $\Delta_t \leq F_{\mathrm{ULI}}(\delta, t) \leq U \leq \epsilon$ for all $t \in \mathbb{N}$, which implies $\sum_{t=1}^{\infty} \mathbb{I}\{\Delta_t > \epsilon\} = 0$. Therefore, any positive function works there and we still choose $F_{\mathrm{UPAC}}(\delta, \epsilon)$ the same as above.

This argument holds for all $\epsilon > 0$. Therefore, if an algorithm satisfies the ULI guarantee, then $\mathbb{P}(A) \geq 1 - \delta$, and with the above statement, we can find a function $F_{\mathrm{UPAC}}$ such that $\mathbb{P}(B) \geq 1 - \delta$, which concludes the proof.

### B.2 Proof of the second bullet: ULI implies regret bounds

For some fixed $\delta \in (0,1)$, let us consider an algorithm enjoys the ULI guarantee with function $F_{\mathrm{ULI}}(\cdot, \cdot)$. The following analysis will condition on event $A$ given in Eq. (6).

**Gap-dependent bound.** Recall that $\Delta > 0$ is the minimum gap. We replace $\epsilon$ with $\Delta/2$ in the proof of in Appendix B.1, case 1, and similarly define $t_0 \in \mathbb{N}$ as the maximal round such that $F_{\mathrm{ULI}}(\delta, t_0) = \texttt{polylog}(t_0/\delta) \cdot t_0^{-\kappa} > \Delta/2$. Then, we apply a similar argument (in Appendix B.1,

case 1), which gives that that for some constants $z \in (0, \kappa)$ and $c_0 > 0$:

$$\sum_{s=1}^{\infty} \mathbb{I}\{\Delta_{A_s} > \Delta/2\}$$
$$\leq t_0$$
$$\leq \min\left\{\left(\frac{2c_0 \cdot \texttt{polylog}(1/\delta)}{\Delta}\right)^{\frac{1}{\kappa-z}}, \frac{\texttt{polylog}\left(\delta^{-1}\left(\frac{2c_0 \cdot \texttt{polylog}(1/\delta)}{\Delta}\right)^{\frac{1}{\kappa-z}}\right)}{(\Delta/2)^{\frac{1}{\kappa}}}\right\}, \quad (9)$$

where the last inequality holds by taking the minimum of Eq. (7) and Eq. (8). Therefore, the algorithm will incur no regret, i.e., $\Delta_t = 0$ for all $t \geq t_0$ For every $T \in \mathbb{N}$, the regret $R_T$ is bounded by

$$
\begin{aligned}
R_T &= \sum_{t=1}^{T} \Delta_t \\
&\leq \sum_{t=1}^{T} \texttt{polylog}(t/\delta) \cdot t^{-\kappa} \\
&\leq \sum_{t=1}^{t_0} \texttt{polylog}(t/\delta) \cdot t^{-\kappa} \\
&\leq \texttt{polylog}(t_0/\delta) \sum_{t=1}^{t_0} t^{-\kappa} \\
&\leq \frac{t_0^{1-\kappa} \cdot \texttt{polylog}(\delta^{-1}t_0)}{1-\kappa}, \quad (10)
\end{aligned}
$$

where the first inequality holds since we condition on the event $A$, the second inequality holds due to the definition of $t_0$, the third inequality uses the fact that $\texttt{polylog}(t/\delta)$ is monotonically increasing for all $t > 0$.

Then, one can further show

$$
\begin{aligned}
R_T &\leq \frac{t_0^{1-\kappa} \cdot \texttt{polylog}(\delta^{-1}t_0)}{1-\kappa} \\
&\leq \frac{\texttt{polylog}\left(\delta^{-1}\left(\frac{2c_0 \cdot \texttt{polylog}(1/\delta)}{\Delta}\right)^{\frac{1}{\kappa-z}}\right) \cdot \texttt{polylog}(\delta^{-1}t_0)}{(\Delta/2)^{1/\kappa-1}} \\
&\leq \frac{\texttt{polylog}\left(\delta^{-1}\left(\frac{2c_0 \cdot \texttt{polylog}(1/\delta)}{\Delta}\right)^{\frac{1}{\kappa-z}}\right) \cdot \texttt{polylog}\left(\delta^{-1}\left(\frac{2c_0 \cdot \texttt{polylog}(1/\delta)}{\Delta}\right)^{\frac{1}{\kappa-z}}\right)}{(\Delta/2)^{1/\kappa-1}},
\end{aligned}
$$

where the second inequality applies the second term of Eq. (9) and the last inequality uses the first term of Eq. (9).

Notice that these bounds in all three cases hold for all $T \in \mathbb{N}$, and thus the proof of gap-dependent bound is complete.

**Gap-independent bound.** We have

$$R_T = \sum_{t=1}^{T} \Delta_t \leq \sum_{t=1}^{T} \texttt{polylog}(t/\delta) \cdot t^{-\kappa} \leq \texttt{polylog}(T/\delta) \sum_{t=1}^{T} t^{-\kappa} \leq \frac{T^{1-\kappa} \cdot \texttt{polylog}(\delta^{-1}T)}{1-\kappa}.$$

# C  Proof of Theorem 3.1

For shorthand, we use $\mathcal{E}$ to denote the event that Eq. (1) holds for all $t \in \mathbb{N}$. From the assumption, $\mathcal{E}$ holds with probability at least $1 - \delta$. The following proof is straightforward by the definition of $\mathcal{E}$. We conditions on $\mathcal{E}$ and consider an arbitrary round $t$. If $A_t = a^\star$, the claim trivially holds as $\Delta_{A_t} = 0$. We then consider $A_t \neq a^\star$. Since Algorithm 1 pulls an arm $A_t \in \mathcal{A}_t$ at each $t$. From the definition of $\mathcal{E}$, we have $\Delta_{A_t} \leq \beta \cdot f(\delta, t)$. Therefore, Algorithm 1 achieves the ULI guarantee with $F_{\text{ULI}}(\delta, t) = \mathcal{O}\left(f(\delta, t)\right)$.

# D  Proof of Theorem 3.2

## D.1  ULI guarantee for SE-MAB algorithm

Consider the successive elimination (SE-MAB) algorithm (e.g., Algorithm 3 in [Even-Dar et al., 2006]) in the MAB setting where $\mathcal{A} = [K]$. We present SE-MAB algorithm in Algorithm 6 and define some notations. Let $N_a(t)$ be the number of times that arm $a$ has been pulled before round $t$ and let $\widehat{\mu}_a(s)$ be the empirical mean of arm $a$ after $s \in \mathbb{N}$ number of plays. We define the confidence width for each $s \in \mathbb{N}$ as

$$\text{wd}(s) = \sqrt{\frac{\log\left(4Ks^2/\delta\right)}{2s}}. \tag{11}$$

The SE-MAB algorithm always chooses the arm (from the current active arm set) with the minimum number of pulls. If there exist two arms $a, j \in [K]$ such that $N_a(t) = N_j(t)$, then, the algorithm chooses the arm with the smallest index.

---

**Algorithm 6** Successive elimination for multi-armed bandit (SE-MAB)

---

**Input**: confidence $\delta \in (0, 1)$.
**Initialize**: active arm set $\mathcal{A}_1 = [K]$ and sample every arm once to update $N_a(K + 1), \widehat{\mu}_a(N_a(K + 1)), \text{wd}(N_a(K + 1))$ for all $a \in [K]$.
**for** $t = K + 1, K + 2, \ldots$ **do**

> Play an arm $A_t = \operatorname{argmin}_{a \in \mathcal{A}_t} N_a(t)$ and observe reward $X_{t, A_t}$.
> Update counter $N_a(t+1) = N_a(t)+1$ for $A_t = a$ and $N_a(t+1) = N_a(t)$ for all $a \in \mathcal{A}_t - \{A_t\}$.
> Update empirical means $\widehat{\mu}_{A_t}(N_{A_t}(t + 1)) = \frac{\widehat{\mu}_{A_t}(N_{A_t}(t)) \cdot N_{A_t}(t) + X_{t, A_t}}{N_{A_t}(t)+1}$ and $\widehat{\mu}_a(N_a(t + 1)) = \widehat{\mu}_a(N_a(t))$ for all $a \in \mathcal{A}_t - \{A_t\}$.
> Update confidence width $\text{wd}(N_a(t + 1))$ based on Eq. (11) for all $a \in \mathcal{A}_t$.
> Update bad arm set $\mathcal{B}_t$ as
>
> $$\mathcal{B}_t = \{a \in \mathcal{A}_t : \exists j \in \mathcal{A}_t \text{ such that } \widehat{\mu}_j(N_j(t + 1)) - \text{wd}(N_j(t + 1)) - \widehat{\mu}_a(N_a(t + 1)) - \text{wd}(N_a(t + 1)) > 0\}.$$
>
> Update active arm set $\mathcal{A}_{t+1} = \mathcal{A}_t - \mathcal{B}_t$.

---

**Definition D.1.** *Let $\mathcal{E}$ be the event that $|\widehat{\mu}_a(N_a(t)) - \mu_a| \leq wd(N_a(t))$ holds for all $t \in \mathbb{N}$ and all $a \in \mathcal{A}_t$.*

**Lemma D.2.** $\mathbb{P}(\mathcal{E}) \geq 1 - \delta$.

*Proof.* Following the standard trick (e.g., [Audibert et al., 2010]), we rewrite the empirical mean for every $a, n$ as $\widehat{\mu}_a(n) = \frac{1}{n} \sum_{i=1}^{n} R_{i,a}$ where $R_{i,a}$ is the reward of $i$-th pull of arm $a$. Let us fix both $s \in \mathbb{N}$ and $a \in [K]$. By Hoeffding's inequality, with probability at least $1 - \delta'$

$$|\widehat{\mu}_a(s) - \mu_a| \leq \sqrt{\frac{\log\left(\frac{2}{\delta'}\right)}{2s}}.$$

Choosing $\delta' = \frac{\delta}{2Ks^2}$ and applying a union bound over $a \in \mathcal{A}_t$ ($|\mathcal{A}_t| \leq K, \forall t$), we have with probability at least $1 - \delta/(2s^2)$,

$$|\widehat{\mu}_a(s) - \mu_a| \leq \sqrt{\frac{\log\left(\frac{4Ks^2}{\delta}\right)}{2s}} = \text{wd}(s), \quad \forall a \in [K].$$

We apply a union bound over all $s \in \mathbb{N}$ and use the fact that $\sum_{s=1}^{\infty} \frac{\delta}{2s^2} \leq \delta$ to finish the proof. $\square$

**Lemma D.3.** *Suppose that $\mathcal{E}$ occurs where $\mathcal{E}$ is given in [Definition D.1](). For all $t \in \mathbb{N}$, $a^\star \in \mathcal{A}_t$ holds.*

*Proof.* We prove this by induction. For $t = 1$, $a^\star \in \mathcal{A}_1$ trivially holds. Suppose that $a^\star \in \mathcal{A}_t$. To show $a^\star \in \mathcal{A}_t$, it suffices to show that at the end of round $t$, arm $a^\star$ is deemed as a good arm. One can use the definition of $\mathcal{E}$ to show that for every $a \in \mathcal{A}_t$, the following holds.

$$0 \leq \mu^\star - \mu_a \leq \widehat{\mu}_{a^\star}(N_{a^\star}(t+1)) + \text{wd}(N_{a^\star}(t+1)) - \widehat{\mu}_a(N_a(t+1)) + \text{wd}(N_a(t+1)).$$

Note that the second inequality above can hold since the inductive hypothesis gives that $a^\star \in \mathcal{A}_t$. The above inequality shows that $a^\star$ will not be eliminated at the end of round $t$ thereby still being active at round $t + 1$. Once the induction is done, the proof is complete. $\square$

**Lemma D.4.** *Suppose that $\mathcal{E}$ occurs where $\mathcal{E}$ is given in [Definition D.1](). For all $t \in \mathbb{N}$ and all arms $a \in \mathcal{A}$ if $a \in \mathcal{A}_t$, then*

$$\Delta_a \leq \sqrt{\frac{14K \log \left(4Kt^2/\delta\right)}{t}}.$$

*Proof.* Consider any round $t \in \mathbb{N}$ and any active arm $a \in \mathcal{A}_t$. For simplicity, we assume that the first arm is the unique optimal arm[7]. If $a$ is an optimal arm, then $\Delta_a = 0$ and we hold the claim trivially. Then, it suffices to consider an arm $a \in \mathcal{A}_t$ with $\Delta_a > 0$. Notice that $\text{wd}(x)$ is monotonically decreasing for all $x \geq 1$ as long as $K \geq 2$. Thus, we find a minimum natural number $T_a \in \mathbb{N}$ such that

$$\Delta_a \geq 5\sqrt{\frac{\log \left(4KT_a^2/\delta\right)}{2T_a}} = 5\text{wd}(T_a). \tag{12}$$

With this definition, we have

$$\Delta_a < 5\text{wd}(s) \quad \forall 1 \leq s < T_a, \quad \text{and} \quad \Delta_a \geq 5\text{wd}(s) \quad \forall s \geq T_a.$$

Recall that $N_a(t)$ is the number of plays of arm $a$ before round $t$. We first show $N_a(t) \leq T_a$. As [Algorithm 6]() pulls arms in a round-robin fashion, if arm $a$ is pulled for $T_a$ times, then, the optimal arm $a^\star$ is also pulled for $T_a$ times, thereby $T_a = T_{a^\star}$. From [Lemma D.3](), optimal arm $a^\star$ is active for all $t \in \mathbb{N}$ and thus we can use it as a comparator.

$$\begin{aligned}
&\widehat{\mu}_{a^\star}(T_{a^\star}) - \text{wd}(T_{a^\star}) - \widehat{\mu}_a(T_a) - \text{wd}(T_a) \\
&\geq \mu^\star - 2\text{wd}(T_{a^\star}) - \mu_a - 2\text{wd}(T_a) && \text{(by } \mathcal{E}) \\
&= \mu^\star - 2\text{wd}(T_a) - \mu_a - 2\text{wd}(T_a) && \text{(by } T_a = T_{a^\star}) \\
&\geq \Delta_a - 2 \times \frac{\Delta_a}{5} - 2 \times \frac{\Delta_a}{5} \\
&= \frac{\Delta_a}{5} \\
&> 0.
\end{aligned}$$

According to elimination rule, this inequality shows that arm $a$ will be eliminated after $T_a$ pulling times, which suggests that $N_a(t) \leq T_a$. Since the $T_a$ is the minimum natural number which holds [Eq. (12)](), we use $N_a(t) \leq T_a$ to show

$$\Delta_a < 5\sqrt{\frac{\log \left(4K(N_a(t) - 1)^2/\delta\right)}{2(N_a(t) - 1)}} \leq 5\sqrt{\frac{\log \left(4Kt^2/\delta\right)}{2(N_a(t) - 1)}},$$

where the last inequality follows from the fact that $N_a(t) \leq t$. Rearranging the above, we have

$$N_a(t) \leq 1 + \frac{25 \log \left(4Kt^2/\delta\right)}{2\Delta_a^2} \leq \frac{13 \log \left(4Kt^2/\delta\right)}{\Delta_a^2}.$$

---

[7]This assumption is made only for simplicity, but our proof can be easily extended to multiple optimal arms by changing the constant.

Again, as Algorithm 6 pulls arms in a round-robin fashion, we have $t \leq K(N_a(t) + 1)$ and show

$$t \leq K(N_a(t) + 1) \leq \frac{14K \log \left(4Kt^2/\delta\right)}{\Delta_a^2},$$

which gives $\Delta_a \leq \sqrt{\frac{14K \log(4Kt^2/\delta)}{t}}$. Conditioning on event $\mathcal{E}$ the argument holds for all $t \in \mathbb{N}$, which thus completes the proof. $\qquad\square$

*Proof of Theorem 3.2 for SE-MAB.* Once Lemma D.3 and Lemma D.4 hold, Theorem 3.1 gives that for any fixed $\delta \in (0, 1)$, SE-MAB achieves the ULI guarantee with a function

$$F_{\mathrm{ULI}}(\delta, t) = \mathcal{O}\left(\sqrt{\frac{K \log \left(Kt/\delta\right)}{t}}\right).$$

Therefore, the proof of Theorem 3.2 for SE-MAB is complete. $\qquad\square$

## D.2 ULI guarantee for PE-MAB

In MAB setting with $\mathcal{A} = [K]$, we further consider phased elimination algorithm (e.g., Exercise 6.8 of [Lattimore and Szepesvári, 2020]) shown in Algorithm 7 (called PE-MAB). The algorithm proceeds with phases $\ell = 1, 2, \ldots$, and each phase $\ell$ includes consecutive rounds, with an exponential increase. In phase $\ell$, the algorithm sequentially pulls every arm $a \in [K]$ for $m_\ell$ times where

$$m_\ell = \left\lceil 2^{2\ell+1} \log \left(4K\ell^2/\delta\right)\right\rceil.$$

Once all arms are pulled $m_\ell$ times, the algorithm steps into the next phase $\ell + 1$. The counter $N_a(\ell)$ records the number of times that arm $a$ is pulled in phase $\ell$ and $\widehat{\mu}_a(m_\ell)$ is the empirical mean by using $m_\ell$ samples only from phase $\ell$.

---

**Algorithm 7** Phased elimination for multi-armed bandit (PE-MAB)

---

**Input**: confidence $\delta \in (0, 1)$.
**Initialize**: active arm set $\mathcal{A}_1 = [K]$.
**for** $\ell = 1, 2, \ldots$ **do**
  Play every arm $a \in \mathcal{A}_\ell$ for $m_\ell$ times and observe corresponding rewards.
  Update empirical means $\{\widehat{\mu}_a(m_\ell)\}_{a \in \mathcal{A}_\ell}$ only using samples from phase $\ell$.
  Update active arm set $\mathcal{A}_{\ell+1}$ as

$$\mathcal{A}_{\ell+1} = \mathcal{A}_\ell - \left\{a \in \mathcal{A}_\ell : \max_{j \in \mathcal{A}_\ell} \widehat{\mu}_j(m_\ell) - \widehat{\mu}_a(m_\ell) > 2^{-\ell}\right\}.$$

---

**Definition D.5.** *Let $\mathcal{E}$ be the event that $|\widehat{\mu}_a(m_\ell) - \mu_a| \leq 2^{-\ell-1}$ holds for all $\ell \in \mathbb{N}$ and all $a \in \mathcal{A}_\ell$.*

We first give the following lemmas, whose proof is similar to those of Lemma D.2 and Lemma D.3.
**Lemma D.6.** $\mathbb{P}(\mathcal{E}) \geq 1 - \delta$.

*Proof.* We first fix both $\ell \in \mathbb{N}$ and $a \in [K]$. By Hoeffding's inequality, with probability at most $\delta'$, we have

$$|\widehat{\mu}_a(m_\ell) - \mu_a| \geq \sqrt{\frac{\log \left(\frac{2}{\delta'}\right)}{2m_\ell}}.$$

Choosing $\delta' = \frac{\delta}{2K\ell^2}$ and applying union bounds over $a \in \mathcal{A}_\ell$ ($\mathcal{A}_\ell$ is at most $K$), with probability at most $\delta/2\ell^2$, we have

$$|\widehat{\mu}_a(m_\ell) - \mu_a| \geq \sqrt{\frac{\log \left(4K\ell^2/\delta\right)}{2m_\ell}} = 2^{-\ell-1}.$$

We apply union bounds over all $\ell \in \mathbb{N}$ and use the fact that $\sum_{\ell=1}^\infty \frac{\delta}{2\ell^2} \leq \delta$ to finish the proof. $\qquad\square$

**Lemma D.7.** *Suppose that $\mathcal{E}$ occurs where $\mathcal{E}$ is given in [Definition D.5](). For all $\ell \in \mathbb{N}$, $a^\star \in \mathcal{A}_\ell$.*

*Proof.* We prove this by induction. For the base case $\ell = 1$, the claim trivially holds. Suppose the claim holds for $\ell$, and then we will show that $a \in \mathcal{A}_{\ell+1}$. For all $\ell \in \mathbb{N}$ and all $a \in \mathcal{A}_\ell$, we have

$$0 \leq \mu^\star - \mu_a \leq \widehat{\mu}_{a^\star}(m_\ell) - \widehat{\mu}_a(m_\ell) + 2^{-\ell},$$

where the second inequality holds because the induction hypothesis ensures $a^\star \in \mathcal{A}_\ell$ and thus we can use the definition of $\mathcal{E}$. Based on the elimination rule, $a^\star \in \mathcal{A}_{\ell+1}$. Once the induction is done, the proof is complete. $\qquad\square$

**Lemma D.8.** *Suppose that $\mathcal{E}$ occurs where $\mathcal{E}$ is given in [Definition D.5](). For each arm $a$ with $\Delta_a > 0$, it will not be in $\mathcal{A}_\ell$ for all phases $\ell \geq \ell_a + 1$ where $\ell_a$ is the smallest phase such that $\frac{\Delta_a}{2} > 2^{-\ell_a}$.*

*Proof.* Consider any arm $a$ with $\Delta_a > 0$. We only need to consider $a \in \mathcal{A}_{\ell_a}$ and otherwise, the claimed result holds trivially. Recall that $\ell_a$ is the smallest phase such that $\frac{\Delta_a}{2} > 2^{-\ell_a}$ and one can show

$$
\begin{aligned}
&\max_{j \in \mathcal{A}_{\ell_a}} \widehat{\mu}_j(m_{\ell_a}) - \widehat{\mu}_a(m_{\ell_a}) - 2^{-\ell_a} \\
&\geq \widehat{\mu}_{a^\star}(m_{\ell_a}) - \widehat{\mu}_a(m_{\ell_a}) - 2^{-\ell_a} \\
&\geq \mu^\star - 2^{-\ell_a} - \mu_a - 2^{-\ell_a} \\
&> \Delta_a - 2 \times \frac{\Delta_a}{2} \\
&= 0,
\end{aligned}
$$

where the first inequality follows from [Lemma D.7]() that $a^\star \in \mathcal{A}_{\ell_a}$ and the second inequality holds due to $\mathcal{E}$.

According to the elimination rule, arm $a$ will not be in phases $\ell$ for all $\ell \geq \ell_a + 1$. $\qquad\square$

**Lemma D.9.** *Let $\ell(t)$ be the phase in which round $t$ lies. Then, $\ell(t) \leq \log_2(t+1)$ for all $t \in \mathbb{N}$.*

*Proof.* We prove this by contradiction. Suppose that $\exists t \in \mathbb{N}$ that $\ell(t) > \log_2(t+1)$. Note that we can further assume $\ell(t) \geq 2$ since one can easily verify that for all $t$ such that $\ell(t) = 1$, $\ell(t) \leq \log_2(t+1)$ must hold. We have

$$t \geq m_{\ell(t)-1} \geq 2^{2\ell(t)-1} \log\left(4K(\ell(t)-1)^2/\delta\right) > \frac{1}{2}(t+1)^2 \log\left(4K/\delta\right) > t,$$

where the third inequality bounds $\ell(t)$ in the logarithmic term by $\ell(t) \geq 2$ and bound the other $\ell(t) > \log_2(t+1)$ by assumption. Therefore, once a contradiction occurs, the proof is complete. $\quad\square$

**Lemma D.10.** *Let $\ell(t)$ be the phase in which round $t$ lies. Suppose that $\mathcal{E}$ occurs where $\mathcal{E}$ is given in [Definition D.5](). For all $t \in \mathbb{N}$ and all $a \in [K]$, if $a \in \mathcal{A}_{\ell(t)}$, then*

$$\Delta_a \leq \sqrt{\frac{256K \log\left(4K\left(\log_2(t+1)\right)^2/\delta\right)}{3t}}.$$

*Proof.* If $a \in \mathcal{A}_{\ell(t)}$ is optimal, then, $\Delta_a = 0$ and the claim trivially holds. In what follows, we only consider arm $a \in \mathcal{A}_{\ell(t)}$ with $\Delta_a > 0$. From [Lemma D.8](), if an arm $a \in \mathcal{A}_{\ell(t)}$ is with $\Delta_a > 0$, then, $\ell(t) \leq \ell_a$ where $\ell_a$ is defined in [Lemma D.8](). Thus, the total number of rounds that such an arm $a$ is active is at most

$$
\begin{aligned}
K \sum_{s=1}^{\ell(t)} m_s &= K \sum_{s=1}^{\ell(t)} \left\lceil 2^{2s+1} \log\left(4Ks^2/\delta\right)\right\rceil \\
&\leq 4K \log\left(4K\ell(t)^2/\delta\right) \sum_{s=1}^{\ell(t)} 2^{2s}
\end{aligned}
$$

$$\leq 4K \log \left(4K\ell(t)^2/\delta\right) \sum_{s=1}^{\ell_a} 2^{2s}$$

$$\leq \frac{16K \log \left(4K\ell(t)^2/\delta\right)}{3} \cdot 4^{\ell_a}$$

$$\leq \frac{256K \log \left(4K\ell(t)^2/\delta\right)}{3\Delta_a^2}$$

$$\leq \frac{256K \log \left(4K \left(\log_2(t+1)\right)^2/\delta\right)}{3\Delta_a^2},$$

where the first inequality follows from $\lceil x \rceil \leq 2x$ for all $x \geq 1$, the second inequality bounds $\ell(t) \leq \ell_a$, the fourth inequality follows from the definition of $\ell_a$, i.e., $\frac{\Delta_a}{2} \leq 2^{-(\ell_a-1)}$, and the last inequality follows from Lemma D.9 that $\ell(t) \leq \log_2(t+1)$.

Since this argument holds for each round $t$ and each arm $a \in \mathcal{A}_{\ell(t)}$ conditioning on $\mathcal{E}$, the proof is complete. $\qquad\square$

*Proof of Theorem 3.2 for PE-MAB.* Once Lemma D.7 and Lemma D.10 hold, Theorem 3.1 gives that for any fixed $\delta \in (0,1)$, PE-MAB achieves the ULI guarantee with a function

$$F_{\mathrm{ULI}}(\delta, t) = \mathcal{O}\left(t^{-\frac{1}{2}} \sqrt{K \log \left(K \log(t+1)/\delta\right)}\right).$$

Therefore, the proof of Theorem 3.2 for PE-MAB is complete. $\qquad\square$

## D.3 ULI guarantee for PE-linear

---

**Algorithm 8** Phased elimination for linear bandit (PE-linear)

---

**Input**: confidence $\delta \in (0, 1)$.
**Initialize**: active arm set $\mathcal{A}_1 = \mathcal{A}$.
**for** $\ell = 1, 2, \ldots$ **do**
  Find a design $\pi_\ell \in \Delta(\mathcal{A}_\ell)$ with

$$\max_{a \in \mathcal{A}_\ell} \|a\|_{G_\ell^{-1}}^2 \leq 2d, \quad \text{and} \quad |\mathrm{supp}(\pi_\ell)| \leq 4d \log \log(d) + 16, \tag{13}$$

  where $G_\ell = \sum_{a \in \mathcal{A}_\ell} \pi_\ell(a) aa^T$.
  Play every arm $a \in \mathcal{A}_\ell$ for $m_\ell(a) = \lceil \pi_\ell(a) m_\ell \rceil$ times and observe corresponding rewards.
  Update the empirical estimate as

$$\widehat{\theta}_\ell = V_\ell^{-1} \sum_{t \in \mathcal{T}_\ell} A_t X_{t, A_t}, \quad \text{where} \quad V_\ell = \sum_{a \in \mathcal{A}_\ell} m_\ell(a) aa^\top. \tag{14}$$

  Update active arm set

$$\mathcal{A}_{\ell+1} = \mathcal{A}_\ell - \left\{ a \in \mathcal{A}_\ell : \max_{b \in \mathcal{A}_\ell} \left\langle \widehat{\theta}_\ell, b - a \right\rangle > 2^{-\ell+1} \right\}.$$

---

We consider a phased elimination algorithm (e.g., algorithm in Chapter 22 of [Lattimore and Szepesvári, 2020]) for linear bandits setting with a finite arm set $\mathcal{A} = [K]$. The algorithm proceeds with phases $\ell = 1, 2, \ldots$, and in each phase $\ell$, the algorithm first computes a design $\pi_\ell \in \Delta(\mathcal{A}_\ell)$ over all active arms where $\Delta(\mathcal{A}_\ell)$ is the set of all Radon probability measures over set $\mathcal{A}_\ell$. Rather than computing an exact design in [Lattimore and Szepesvári, 2020], we follow Lattimore et al. [2020] to compute a nearly-optimal design (13), which can be efficiently implemented. Then, PE-linear plays each arm $a \in \mathcal{A}_\ell$ for $m_\ell(a)$ times and updates the active arm set by using the estimates in this phase.

Let us define $\mathcal{T}_\ell$ be a set that contains all rounds in phase $\ell$ and

$$m_\ell(a) = \lceil \pi_\ell(a) m_\ell \rceil, \quad \text{where} \quad m_\ell = \frac{4d}{2^{-2\ell}} \max \left\{ \log \left(4K\ell^2/\delta\right), \log \log d + 4 \right\}. \tag{15}$$

**Lemma D.11.** *For all $\ell \in \mathbb{N}$ and all $a \in \mathcal{A}_\ell$, $\|a\|^2_{V_\ell^{-1}} \le \frac{2d}{m_\ell}$ holds.*

*Proof.* For each $\ell \in \mathbb{N}$ and $a \in \mathcal{A}_\ell$, one can show

$$\|a\|^2_{V_\ell^{-1}} = a^T V_\ell^{-1} a \le a^T \left( m_\ell \sum_{a \in \mathcal{A}_\ell} \pi_\ell(a) a a^\top \right)^{-1} a \le \frac{2d}{m_\ell},$$

where the first inequality follows from Eq. (15) and the fact that if we let $A = m_\ell \sum_{a \in \mathcal{A}_\ell} \pi_\ell(a) a a^\top$ and $B = V_\ell$ then, $\|a\|_{A^{-1}} \ge \|a\|_{B^{-1}}$ holds since $A^{-1} \succeq B^{-1}$ and the second inequality uses Eq. (13). □

**Definition D.12.** *Let $\mathcal{E}$ be the event that $\left| \left\langle \widehat{\theta}_\ell - \theta, a \right\rangle \right| \le 2^{-\ell}$ holds for all $\ell \in \mathbb{N}$ and all $a \in \mathcal{A}_\ell$.*

**Lemma D.13.** $\mathbb{P}(\mathcal{E}) \ge 1 - \delta$.

*Proof.* Consider a fixed phase $\ell$ and a fixed arm $a \in \mathcal{A}_\ell$. We start from the following decomposition.

$$
\begin{aligned}
\left\langle \widehat{\theta}_\ell - \theta, a \right\rangle &= \left\langle V_\ell^{-1} \sum_{t \in \mathcal{T}_\ell} A_t X_{t, A_t} - \theta, a \right\rangle \\
&= \left\langle V_\ell^{-1} \sum_{t \in \mathcal{T}_\ell} A_t \left( A_t^T \theta + \eta_t \right) - \theta, a \right\rangle \\
&= \left\langle V_\ell^{-1} \sum_{t \in \mathcal{T}_\ell} A_t \eta_t, a \right\rangle \\
&= \sum_{t \in \mathcal{T}_\ell} \left\langle V_\ell^{-1} A_t, a \right\rangle \eta_t.
\end{aligned}
$$

By Eq. (20.2) of [Lattimore and Szepesvári, 2020], we have with probability at least $1 - \delta/(2\ell^2 K)$,

$$\left| \left\langle \widehat{\theta}_\ell - \theta, a \right\rangle \right| = \left| \sum_{t \in \mathcal{T}_\ell} \left\langle V_\ell^{-1} A_t, a \right\rangle \eta_t \right| \le \sqrt{2 \|a\|^2_{V_\ell^{-1}} \log(4\ell^2 K/\delta)} \le 2 \sqrt{\frac{d \log(4\ell^2 K/\delta)}{m_\ell}} \le 2^{-\ell},$$

where the last inequality applies Lemma D.11. Finally, applying union bounds over all $a, \ell$ completes the proof. □

**Lemma D.14.** *Suppose that $\mathcal{E}$ occurs where $\mathcal{E}$ is given in Definition D.12. We have $a^\star \in \mathcal{A}_\ell$ for all $\ell \in \mathbb{N}$.*

*Proof.* This can be proved by using induction on $\ell$ via the same reasoning of Lemma D.3. □

**Lemma D.15.** *Suppose that $\mathcal{E}$ occurs where $\mathcal{E}$ is given in Definition D.12. For each arm $a$ with $\Delta_a > 0$, it will not be in $\mathcal{A}_\ell$ for all phases $\ell \ge \ell_a + 1$ where $\ell_a$ is the smallest phase such that $\frac{\Delta_a}{4} > 2^{-\ell_a}$.*

*Proof.* Consider any arm $a$ with $\Delta_a > 0$. Let $\ell_a$ be the smallest phase such that $\frac{\Delta_a}{4} > 2^{-\ell_a}$ and we have

$$\max_{b \in \mathcal{A}_{\ell_a}} \left\langle \widehat{\theta}_{\ell_a}, b - a \right\rangle - 2^{-\ell_a + 1} \ge \left\langle \widehat{\theta}_{\ell_a}, a^\star - a \right\rangle - 2^{-\ell_a + 1} \ge \left\langle \theta, a^\star - a \right\rangle - 2^{-\ell_a + 2} > \Delta_a - 4 \times \frac{\Delta_a}{4} = 0,$$

where the first inequality follows from Lemma D.14 that $a^\star \in \mathcal{A}_\ell$ for all $\ell$.

According to the elimination rule, arm $a$ will not be in $\mathcal{A}_\ell$ for all phases $\ell \ge \ell_a + 1$. Since conditioning on $\mathcal{E}$, this argument holds for every arm $a$, the proof is complete. □

**Lemma D.16.** *Let $\ell(t)$ be the phase in which round $t$ lies. Then, $\ell(t) \le \log_2(t + 1)$ for all $t \in \mathbb{N}$.*

*Proof.* We prove this by contradiction. Suppose that $\exists t \in \mathbb{N}$ that $\ell(t) > \log_2(t+1)$. Note that we can further assume $\ell(t) \geq 2$ since one can easily verify that for all $t$ such that $\ell(t) = 1$, $\ell(t) \leq \log_2(t+1)$ must hold. We have

$$t \geq \sum_{a \in \mathcal{A}_{\ell(t)-1}} \left\lceil \pi_{\ell(t)-1} m_{\ell(t)-1} \right\rceil$$

$$\geq m_{\ell(t)-1} \geq \frac{4d}{2^{-2(\ell(t)-1)}} \left( \log \left( 4K(\ell(t)-1)^2/\delta \right) \right) > d(t+1)^2 \log\left( 4K/\delta \right) > t,$$

where the fourth inequality bounds $\ell(t)$ in the logarithmic term by $\ell(t) \geq 2$ and bound the other $\ell(t) > \log_2(t+1)$ by assumption. Therefore, once a contradiction occurs, the proof is complete. $\square$

**Lemma D.17.** *Let $\ell(t)$ be the phase in which round $t$ lies. Suppose that $\mathcal{E}$ occurs where $\mathcal{E}$ is given in [Definition D.12](). For all $t \in \mathbb{N}$ and all $a \in [K]$, if $a \in \mathcal{A}_{\ell(t)}$, then*

$$\Delta_a \leq \sqrt{\frac{512d\log\left( 4\log(d)K\left(\log_2(t+1)\right)^2/\delta \right) + 4}{3t}}.$$

*Proof.* If $a \in \mathcal{A}_{\ell(t)}$ is optimal, then, $\Delta_a = 0$ and the claim trivially holds. In what follows, we only consider arm $a \in \mathcal{A}_{\ell(t)}$ with $\Delta_a > 0$. From [Lemma D.15](), if an arm $a \in \mathcal{A}_{\ell(t)}$ is with $\Delta_a > 0$, then, $\ell(t) \leq \ell_a$ where $\ell_a$ is defined in [Lemma D.15](). Then, the total number of rounds that such an arm $a$ is active is at most

$$\sum_{s=1}^{\ell(t)} \sum_{a \in \mathcal{A}_s, \pi_s(a) \neq 0} \lceil \pi_s(a) m_s \rceil$$

$$\leq \sum_{s=1}^{\ell(t)} \left( 4d \log\log(d) + 16 + \sum_{a \in \mathcal{A}_s, \pi_s(a) \neq 0} \pi_s(a) m_s \right)$$

$$\leq 2 \sum_{s=1}^{\ell(t)} m_s$$

$$\leq 8d \max\left\{ \log\left( 4K\ell(t)^2/\delta \right), \log\log d + 4 \right\} \sum_{s=1}^{\ell(t)} \frac{1}{2^{-2s}}$$

$$\leq 8d \max\left\{ \log\left( 4K\left(\log_2(t+1)\right)^2/\delta \right), \log\log d + 4 \right\} \sum_{s=1}^{\ell_a} \frac{1}{2^{-2s}}$$

$$\leq \frac{512d}{3\Delta_a^2} \left( \log\left( 4\log(d)K\left(\log_2(t+1)\right)^2/\delta \right) + 4 \right),$$

where the first inequality applies $\lceil \pi_s(a) m_s \rceil \leq \pi_s(a) m_s + 1$ and [Eq. (13)](), the second inequality uses the definition of $m_s$ (see [Eq. (15)]()), the fourth inequality follows from $\ell(t) \leq \ell_a$ and [Lemma D.16]() gives $\ell(t) \leq \log_2(t+1)$, and the last inequality uses $\frac{\Delta_a}{4} \leq 2^{-(\ell_a-1)}$ to apply $\ell_a \leq \log_2\left( 8/\Delta_a \right)$.

Since this argument holds for each round $t$ and each arm $a \in \mathcal{A}_{\ell(t)}$ conditioning on $\mathcal{E}$, the proof is complete. $\square$

*Proof of [Theorem 3.2]() for PE-linear.* Once [Lemma D.14]() and [Lemma D.17]() hold, [Theorem 3.1]() gives that for any fixed $\delta \in (0,1)$, PE-linear achieves the ULI guarantee with a function

$$F_{\mathrm{ULI}}(\delta, t) = \mathcal{O}\left( t^{-\frac{1}{2}} \sqrt{d\log\left(\log(d)K\left(\log(t+1)\right)/\delta\right)} \right).$$

Therefore, the proof of [Theorem 3.2]() for PE-linear is complete. $\square$

# E    Omitted Details of Section 3.2

The bonus function that we here consider for lil'UCB is as:

$$U_\delta(x) = \sqrt{\frac{\log\log\left(\max\left\{x, e\right\}\right) + \log\left(6/\delta\right)}{x}}. \tag{16}$$

The choice of $U_\delta(x)$ in Eq. (16) is slightly different from that of [Jamieson et al., 2014] because we consider for any $\delta \in (0, 1)$ and they constrain the choice of $\delta$ in a more restricted range. The choice of $U_\delta(x)$ is motivated from another lemma of law of iterated logarithm, given in [Dann et al., 2017, Lemma F.1] with $\sigma^2 = 1/4$ as we here consider $[0, 1]$-bounded rewards. Note that the concentration bounds in [Dann et al., 2017] apply for the conditionally subgaussian random variables, and one can get the same result for i.i.d. subgaussian random variables, by simply replacing the Doob's maximal inequality by Hoeffding's maximal inequality (see [Jamieson et al., 2014, Lemma 3]).

One caveat here is that our analysis still works for other choices of $U_\delta(x)$ if one adjust constant or change $\log\log(\cdot)$ to $\log(\cdot)$.

---

**Algorithm 9** lil'UCB

---

**Input**: confidence $\delta \in (0, 1)$ and arm set $\mathcal{A}$.
**Initialize**: play each arm $a \in \mathcal{A}$ once to update $N_a(|\mathcal{A}| + 1)$ and $\widehat{\mu}_a(N_a(|\mathcal{A}| + 1))$ for all $a \in \mathcal{A}$.
**for** $t = |\mathcal{A}| + 1, |\mathcal{A}| + 2, \ldots$ **do**
  Play an arm
$$A_t = \operatorname*{argmax}_{a \in \mathcal{A}} \left\{\widehat{\mu}_a(N_a(t)) + U_\delta\left(N_a(t)\right)\right\},$$
  where $U_\delta(N_a(t))$ is given in Eq. (16).
  Update counters $N_a(t + 1) = N_a(t) + 1$ for $A_t = a$ and $N_a(t + 1) = N_a(t)$ for all $a \neq A_t$.
  Update empirical means $\widehat{\mu}_{A_t}(N_{A_t}(t + 1)) = \frac{\widehat{\mu}_{A_t}(N_{A_t}(t)) \cdot N_{A_t}(t) + X_{t, A_t}}{N_{A_t}(t) + 1}$ and $\widehat{\mu}_a(N_a(t + 1)) = \widehat{\mu}_a(N_a(t))$ for all $a \in \mathcal{A} - \{A_t\}$.

---

## E.1    Proof of Theorem 3.3

In the following proof, we consider the following instance.

**Definition E.1** (Two-armed bandit instance). *Consider a two-armed setting with $\mu_1 > \mu_2$. In each round, each arm generates deterministic rewards $\mu_1$ and $\mu_2$, respectively. The arm gap is $\Delta = \mu_1 - \mu_2$ and $\Delta \in (0, 0.6)$.*

According to Lemma E.2, the total number of plays of arm 2 is finite. Therefore, one can find a round $t_0 \in \mathbb{N}$ that the last play of arm 2 occurs. At the beginning of this round, the algorithm compares $\mu_1 + U_\delta(N_1(t_0))$ and $\mu_2 + U_\delta(N_2(t_0))$. Since arm 2 gets the last play at this round, we have

$$\mu_1 + U_\delta(N_1(t_0)) \leq \mu_2 + U_\delta(N_2(t_0)) \leq \mu_2 + (1 + f(\Delta))\Delta,$$

where the last inequality holds due to Lemma E.4 and $f(\Delta)$ is defined as

$$f(\Delta) := \Delta \cdot \frac{\sqrt{3}}{\sqrt{\log\log\left(\frac{\log(6/\delta)}{\Delta^2}\right) + \log(6/\delta)}}. \tag{17}$$

Rearranging the above, we have

$$U_\delta(N_1(t_0)) \leq f(\Delta)\Delta,$$

which immediately leads to

$$N_1(t_0) \geq \frac{\log\log\left(\frac{\log(6/\delta)}{f^2(\Delta)\Delta^2}\right) + \log(6/\delta)}{f^2(\Delta)\Delta^2}.$$

Moreover, since arm 2 is played at round $t_0$, $\Delta = \Delta_{A_{t_0}}$, which further implies that

$$t_0 = N_1(t_0) + N_2(t_0)$$

$$> N_1(t_0)$$

$$> \frac{\log\log\left(\frac{\log(6/\delta)}{f^2(\Delta)\Delta^2}\right) + \log(6/\delta)}{f^2(\Delta)\Delta^2}$$

$$\geq \frac{\left(\log\log\left(\frac{\log(6/\delta)}{\Delta^2}\right) + \log(6/\delta)\right)^2}{3\Delta^4}$$

$$= \frac{\left(\log\log\left(\frac{\log(6/\delta)}{\Delta^2}\right) + \log(6/\delta)\right)^2}{3\Delta_{A_{t_0}}^4},$$

where the third inequality uses $f(\Delta) \leq 1$ for all $\Delta \in (0, 0.6)$, and the last equality holds due to $\Delta = \Delta_{A_{t_0}}$ (recall that in round $t_0$, arm 2 gets the last play).

Rearranging the above, we have

$$\Delta_{A_{t_0}} > \frac{\sqrt{\log\log\left(\frac{\log(6/\delta)}{\Delta^2}\right) + \log(6/\delta)}}{(3t_0)^{\frac{1}{4}}} = \sqrt{\frac{(3t_0)^{\frac{1}{2}}\left(\log\log\left(\frac{\log(6/\delta)}{\Delta^2}\right) + \log(6/\delta)\right)}{3t_0}}.$$

Finally, by Lemma E.3, we have $t_0 = \Omega(\Delta^{-2})$, which completes the proof.

## E.2 Proof of Supporting Lemmas

**Lemma E.2.** *Suppose that we run Algorithm 9 for the two-armed bandit instance given in Definition E.1. For any $\delta \in (0, 1)$, suboptimal arm 2 will be played at most*

$$\left\lceil \frac{2\left(\log\log\left(\frac{16\log(6/\delta)}{\Delta^2}\right) + \log(6/\delta)\right)}{\Delta^2} \right\rceil.$$

*Proof.* Note that for any fixed $\delta \in (0, 1)$, function $U_\delta(x)$ is monotonically-decreasing for $x \geq 6$. To show the claimed result, it suffices to find an integer $x \geq 6$ such that

$$U_\delta(x) < \Delta.$$

Now, we consider the following choice.

$$N = \lceil n \rceil, \quad \text{where} \quad n = \frac{2\left(\log\log\left(\frac{16\log(6/\delta)}{\Delta^2}\right) + \log(6/\delta)\right)}{\Delta^2}$$

Notice that $N \geq n \geq 6$ for all $\Delta \in (0, 1]$. Then, one can show

$$U_\delta(N) \leq U_\delta(n)$$

$$= \Delta \cdot \sqrt{\frac{\log\log\left(\frac{2\left(\log\log\left(\frac{16\log(6/\delta)}{\Delta^2}\right)+\log(6/\delta)\right)}{\Delta^2}\right) + \log(6/\delta)}{2\log\log\left(\frac{16\log(6/\delta)}{\Delta^2}\right) + 2\log(6/\delta)}}$$

$$\leq \Delta \cdot \sqrt{\frac{\log\log\left(\frac{4\log\log\left(\frac{16\log(6/\delta)}{\Delta^2}\right)\cdot\log(6/\delta)}{\Delta^2}\right) + \log(6/\delta)}{2\log\log\left(\frac{16\log(6/\delta)}{\Delta^2}\right) + 2\log(6/\delta)}}$$

$$= \Delta \cdot \sqrt{\frac{\log\left(\log\left(\frac{1}{4}\log\log\left(\frac{16\log(6/\delta)}{\Delta^2}\right)\right) + \log\left(\frac{16\log(6/\delta)}{\Delta^2}\right)\right) + \log(6/\delta)}{2\log\log\left(\frac{16\log(6/\delta)}{\Delta^2}\right) + 2\log(6/\delta)}}$$

$$\leq \Delta \cdot \sqrt{\frac{2 \log \log \left( \frac{16 \log(6/\delta)}{\Delta^2} \right) + \log(6/\delta)}{2 \log \log \left( \frac{16 \log(6/\delta)}{\Delta^2} \right) + 2 \log(6/\delta)}}$$

$$\leq \Delta,$$

where the first inequality uses the fact that for any fixed $\delta \in (0,1)$, $U_\delta(x)$ is monotonically-decreasing for $x \geq 6$, the second inequality uses $x + y \leq 2xy$ for all $x, y > 1$ (here $x = \log(6/\delta)$ and $y = \log \log \left( \frac{16 \log(6/\delta)}{\Delta^2} \right)$), the last inequality first bounds $\frac{1}{4} \log \log \left( \frac{16 \log(6/\delta)}{\Delta^2} \right) \leq \frac{16 \log(6/\delta)}{\Delta^2}$ and then uses $2x \leq x^2$ for all $x \geq 2$ with $x = \log \left( \frac{16 \log(6/\delta)}{\Delta^2} \right) \geq 2$.

Thus, the proof is complete. $\qquad \square$

From Lemma E.2, the total number of plays of suboptimal arm is finite. Based on this fact, we present the following lemmas.

**Lemma E.3.** *Suppose that we run Algorithm 9 for the two-armed bandit instance given in Definition E.1. Let $n+1$ be the total number of plays of arm* 2*. Then,*

$$n \geq \left\lfloor \frac{\log \log \left( \frac{\log(6/\delta)}{\Delta^2} \right) + \log(6/\delta)}{\Delta^2} \right\rfloor \geq 6.$$

*Proof.* For shorthand, we define

$$m = \frac{\log \log \left( \frac{\log(6/\delta)}{\Delta^2} \right) + \log(6/\delta)}{\Delta^2}. \tag{18}$$

With this definition and $\Delta \in (0, 0.6)$, we have $m \geq 6$ and $\lfloor m \rfloor \geq 6$.

Now, we show that $n \geq \lfloor m \rfloor$. To this end, it suffices to show $U_\delta(\lfloor m \rfloor) > \Delta$. As $U_\delta(x)$ is monotonically-decreasing for all $x \geq 6$ and $m \geq \lfloor m \rfloor \geq 6$, we have

$$U_\delta(\lfloor m \rfloor) \geq U_\delta(m)$$

$$= \Delta \cdot \sqrt{\frac{\log \log \left( \frac{\log \log \left( \frac{\log(6/\delta)}{\Delta^2} \right) + \log(6/\delta)}{\Delta^2} \right) + \log(6/\delta)}{\log \log \left( \frac{\log(6/\delta)}{\Delta^2} \right) + \log(6/\delta)}}$$

$$> \Delta,$$

where the last inequality holds as $\frac{\log(6/\delta)}{\Delta^2} > \frac{\log(6)}{\Delta^2} > e$.

Once $U_\delta(\lfloor m \rfloor) > \Delta$ holds, $\lfloor m \rfloor$ cannot be the last play of arm 2 (i.e., at least the one before the last), which completes the proof. $\qquad \square$

**Lemma E.4.** *Suppose that we run Algorithm 9 for the two-armed bandit instance given in Definition E.1. Let $n+1$ be the total number of plays of arm* 2*. The following holds.*

$$U_\delta(n) \leq \Delta \left( 1 + \Delta \sqrt{\frac{3}{\log \log \left( \frac{\log(6/\delta)}{\Delta^2} \right) + \log(6/\delta)}} \right).$$

*Proof.* From Lemma E.3, we have $n \geq 6$, and thus $U_\delta(n) \geq U_\delta(n+1)$ as $U_\delta(x)$ is monotonically decreasing for all $x \geq 6$. Using the fact that for all $a \geq b \geq 0$, $a - b \leq \sqrt{a^2 - b^2}$ (here $a = U_\delta(n)$ and $b = U_\delta(n+1)$), one can show that

$$U_\delta(n) - U_\delta(n+1)$$

$$= \sqrt{\frac{\log \log \left( \max\{n, e\} \right) + \log(6/\delta)}{n}} - \sqrt{\frac{\log \log \left( \max\{n+1, e\} \right) + \log(6/\delta)}{n+1}}$$

$$\leq \sqrt{\frac{\log\log\left(\max\{n,e\}\right)+\log\left(6/\delta\right)}{n}-\frac{\log\log\left(\max\{n+1,e\}\right)+\log\left(6/\delta\right)}{n+1}}$$

$$=\sqrt{\frac{(n+1)\log\log\left(\max\{n,e\}\right)-n\log\log\left(\max\{n+1,e\}\right)+\log\left(6/\delta\right)}{n(n+1)}}$$

$$\leq \sqrt{\frac{\log\log\left(\max\{n,e\}\right)+\log\left(6/\delta\right)}{n(n+1)}}. \tag{19}$$

Since for any fixed $\delta \in (0,1)$, $U_\delta(x)$ is monotonically-decreasing for all $x \geq 6$, and thus by using the definition of $m$ (see Eq. (18)), one can show that

$$U_\delta(n)-U_\delta(n+1)$$

$$\leq \sqrt{\frac{\log\log\left(\lfloor m\rfloor\right)+\log\left(6/\delta\right)}{\lfloor m\rfloor\left(\lfloor m\rfloor+1\right)}}$$

$$\leq \sqrt{\frac{\log\log\left(m\right)+\log\left(6/\delta\right)}{m^2}}$$

$$=\frac{\Delta^2}{\log\log\left(\frac{\log(6/\delta)}{\Delta^2}\right)+\log(6/\delta)}\cdot\sqrt{\log\log\left(\frac{\log\log\left(\frac{\log(6/\delta)}{\Delta^2}\right)+\log(6/\delta)}{\Delta^2}\right)+\log\left(6/\delta\right)}$$

$$\leq \frac{\Delta^2}{\log\log\left(\frac{\log(6/\delta)}{\Delta^2}\right)+\log(6/\delta)}\cdot\sqrt{\log\left(2\log\left(\frac{\log(6/\delta)}{\Delta^2}\right)\right)+\log\left(6/\delta\right)}$$

$$\leq \frac{\Delta^2}{\log\log\left(\frac{\log(6/\delta)}{\Delta^2}\right)+\log(6/\delta)}\cdot\sqrt{3\log\log\left(\frac{\log(6/\delta)}{\Delta^2}\right)+\log\left(6/\delta\right)}$$

$$\leq \frac{\sqrt{3}\Delta^2}{\sqrt{\log\log\left(\frac{\log(6/\delta)}{\Delta^2}\right)+\log(6/\delta)}},$$

where the first inequality uses Eq. (19) with $n \geq \lfloor m\rfloor \geq e$ and $U_\delta(x)$ is monotonically decreasing for all $x \geq 6$, the second inequality upper-bounds $\lfloor m\rfloor \leq m$ in numerator and lower-bounds $\lfloor m\rfloor\left(\lfloor m\rfloor+1\right)\geq m^2$, the third inequality follows from

$$\frac{\log\log\left(\frac{\log(6/\delta)}{\Delta^2}\right)+\log(6/\delta)}{\Delta^2}\leq\frac{\frac{\log(6/\delta)}{\Delta^2}+\log(6/\delta)}{\Delta^2}\leq\left(\frac{\log(6/\delta)}{\Delta^2}\right)^2,$$

and the fourth inequality follows $2x \leq x^3$ for all $x \geq \sqrt{2}$ with $x = \log\left(\frac{\log(6/\delta)}{\Delta^2}\right)\geq\sqrt{2}$ ($\Delta \in (0,0.6)$).

After arm 2 gets $n+1$ times play, it will not be played anymore, which implies that $U_\delta(n+1)<\Delta$. Using this bound gives

$$U_\delta(n)\leq\Delta+\Delta^2\sqrt{\frac{3}{\log\log\left(\frac{\log(6/\delta)}{\Delta^2}\right)+\log(6/\delta)}}.$$

Therefore, we complete the proof. $\qquad\square$

### E.3 Traditional Optimistic Algorithms Fail to Achieve ULI Guarantee

We will show that the traditional optimistic algorithms fail to achieve the ULI guarantee due to the $\log t$ term in the bonus. At a high level, $\log t$ is increasing with time and forces the algorithm to play a bad arm indefinitely when $t$ evolves. We provide the formal proof in the following:

We prove the claim by contradiction. Consider a $K$-armed bandit instance with at least one suboptimal arm, and let $\Delta > 0$ be the minimum arm gap. Suppose there exists an optimism-based algorithm with $\log t$ in bonus term that can achieve the ULI guarantee in this setting. Then, for some fixed $\delta \in (0, 1)$, with probability $\geq 1 - \delta$, for all $t \in \mathbb{N}$, $\Delta_t \leq F_{ULI}(\delta, t)$. Based on Definition 2.5, we have $\lim_{t \to \infty} F_{ULI}(\delta, t) = 0$ and $F_{ULI}(\delta, t)$ is monotonically decreasing w.r.t. $t$ after a threshold. Thus, $\exists t_0 \in \mathbb{N}$ such that $F_{ULI}(\delta, t) < \Delta$ for all $t \geq t_0$. In other words, the algorithm cannot play any suboptimal arm after $t_0$-th round. Recall that the bonus term is $\sqrt{\log t / N_a(t)}$ where $N_a(t)$ is the number of plays of arm $a$ before round $t$. For any suboptimal arm $a$, $N_a(t)$ should not increase after $t_0$-th round, but $\log t$ keeps increasing. This leads the bonus of arm $a$ goes to infinity, which will incur a play of arm $a$ at a round after $t_0$. This makes a contradiction.

**Algorithm 10** Meta-algorithm towards ULI

**Input**: $\delta \in (0, 1)$, finite arm set $\mathcal{A}$, and base-algorithm `Alg` with function $g(\delta)$ that satisfies Condition. F.1.

**Initialize**: $\mathcal{A}_1 = [K]$.

1 **for** $m = 1, 2, \ldots$ **do**

2      Set duration $T_m = \lceil 576 g_m 2^m \rceil$ where $g_m = g\left(\frac{\delta}{2m^2}\right)$.

3      Run `Alg` with active arm set $\mathcal{A}_m$ for $T_m$ rounds and construct $\widehat{\ell}_{t,a} = \frac{\ell_{t,a} \cdot \mathbb{I}\{A_t = a\}}{p_{t,a}}$ for all $t, a$ in this phase.

4      Update active arm set $\mathcal{A}_{m+1} = \mathcal{A}_m - \mathcal{B}_m$ where:

$$\mathcal{B}_m = \left\{ a \in \mathcal{A}_m : \sum_{s \in \mathcal{T}_m} \left( \widehat{\ell}_{s,a} - \widehat{\ell}_{s,k} \right) > 7 \sqrt{g_m T_m} \right\}, \tag{20}$$

     and $\mathcal{T}_m$ is a set that contains all rounds in phase $m$ and $k$ is the empirical best arm:

$$k \in \underset{a \in \mathcal{A}_m}{\operatorname{argmin}} \sum_{s \in \mathcal{T}_m} \widehat{\ell}_{s,a}$$

# F Achieving ULI by Adversarial Bandit Algorithms

## F.1 Meta-algorithm Enabling Adversarial Algorithms to Achieve ULI

In this subsection, we propose a meta-algorithm shown in Algorithm 10 which enables any high-probability adversarial algorithm, with a mild condition, to achieve the ULI guarantee. Then, we show that existing high-probability adversarial bandit algorithms naturally meet this condition in both MAB and linear bandit settings. For notational simplicity, we follow the convention of adversarial analysis to use loss $\ell_{t,a} = 1 - X_{t,a}$. Note that all algorithms in this subsection require $[0, 1]$-boundedness assumption on loss.

At a high-level, our meta-algorithm keeps running a base-algorithm `Alg` to play arms and collects rewards. The meta-algorithm uses collected rewards to construct the importance-weighted (IW) estimator $\widehat{\ell}_{t,a} = \frac{\ell_{t,a} \cdot \mathbb{I}\{A_t = a\}}{p_{t,a}}$ for each $t, a$, which helps to eliminate bad arms, and then runs `Alg` on a reduced arm space. To achieve the ULI guarantee, our meta-algorithm requires the input base-algorithm to satisfy the following:

**Condition F.1.** *An algorithm* `Alg` *runs for given consecutive $T$ rounds with a finite arm set $\mathcal{A}$ and a fixed $\delta \in (0, 1)$. At each round $t \in [T]$,* `Alg` *maintains a distribution $p_t$ over $\mathcal{A}$ and samples an arm $A_t \sim p_t$. With probability at least $1 - \frac{3}{5}\delta$,* `Alg` *ensures for all $a \in \mathcal{A}$,*

$$\sum_{t=1}^{T} (\ell_{t,A_t} - \ell_{t,a}) \leq \sqrt{g(\delta)T} - 2 \left| \sum_{t=1}^{T} \left( \widehat{\ell}_{t,a} - \ell_{t,a} \right) \right|, \tag{21}$$

*where $g(\delta)$ is a positive-valued function, monotonically decreasing for all $\delta \in (0, 1)$, polynomial in $\log(1/\delta)$, and $g(\delta) \geq \log\left(10|\mathcal{A}|/\delta\right), \forall \delta \in (0, 1)$.*

In fact, many high-probability adversarial bandit algorithms naturally meet Condition. F.1, but require stronger analysis. In particular, we show that EXP3.P [Auer et al., 2002b] for MAB holds this condition with $g(\delta) = \mathcal{O}\left(|\mathcal{A}| \log(|\mathcal{A}|/\delta)\right)$ (omitted proof can be found in Appendix F) and Lee et al. [2021] show that GEOMETRICHEDGE.P [Bartlett et al., 2008] with John's exploration meets the condition for linear bandits with $g(\delta) = \mathcal{O}\left(d \log(|\mathcal{A}|/\delta)\right)$. Hence, feeding those algorithms to Algorithm 10 yields the following:

**Theorem F.2.** *For any fixed $\delta \in (0, 1)$, if Algorithm 10 uses*

- EXP3.P *as a base-algorithm, then, for $K$-armed bandit, ULI guarantee is achieved with*

$$F_{ULI}(\delta, t) = \mathcal{O}\left(t^{-\frac{1}{2}} \sqrt{K \log\left(\delta^{-1} K \log(t + 1)\right)}\right).$$

- GEOMETRICHEDGE.P *as a base-algorithm then for linear bandits with $K$ arms, ULI guarantee is achieved with*

$$F_{ULI}(\delta, t) = \mathcal{O}\big(t^{-\frac{1}{2}}\sqrt{d\log\big(\delta^{-1}K\log(t+1)\big)}\big).$$

Theorem F.2 suggests that Algorithm 10 enables EXP3.P and GEOMETRICHEDGE.P to achieve near-optimal ULI guarantees for MAB and linear bandits, respectively. Moreover, their ULI guarantees are as good as those of conventional elimination-based algorithms (refer to Theorem 3.2).

Our main objective in this section is to prove the following results.

**Theorem F.3** (Restatement of Theorem F.2)**.** *For any fixed $\delta \in (0, 1)$, if Algorithm 10 uses*

- EXP3.P *as a base-algorithm, then, for $K$-armed bandit, ULI guarantee is achieved with*

$$F_{ULI}(\delta, t) = \mathcal{O}\big(t^{-\frac{1}{2}}\sqrt{K\log\big(\delta^{-1}K\log(t+1)\big)}\big).$$

- GEOMETRICHEDGE.P *as a base-algorithm then for linear bandits with $K$ arms, ULI guarantee is achieved with*

$$F_{ULI}(\delta, t) = \mathcal{O}\big(t^{-\frac{1}{2}}\sqrt{d\log\big(\delta^{-1}K\log(t+1)\big)}\big)$$

We decompose the proof of Theorem F.2 into two parts. In the first part, we first show that any algorithm with Condition. F.1 for some $\delta \in (0, 1)$ can achieve the ULI guarantee. Then, we only need to show that EXP3.P and GEOMETRICHEDGE.P satisfy Condition. F.1, which completes the proof of Theorem F.2.

The following result suggests that showing an algorithm enjoys the ULI guarantee is reduced to show that this algorithm meets Condition. F.1.

**Theorem F.4.** *For any fixed $\delta \in (0, 1)$, if Algorithm 10 uses a base-algorithm that satisfies Condition. F.1, then, it achieves the ULI guarantee that*

$$\mathbb{P}\left(\forall t \in \mathbb{N} : \Delta_{A_t} = \mathcal{O}\left(\sqrt{\frac{g\big(\delta^{-1}\log^2(t+1)\big)}{t}}\right)\right) \geq 1 - \delta.$$

We sketch the proof of Theorem F.4 as follows. The full proof can be found in Appendix F.2.

*Proof Sketch.* The high-level objectives are to show the sufficient condition of Eq. (1) given in Algorithm 1 to achieve the ULI. We first show that the optimal arm will not be eliminated, i.e., $a^\star \in \mathcal{A}_m$ for all phases $m \in \mathbb{N}$. To show this, we only need show

$$\forall m \in \mathbb{N} \quad \sum_{s \in \mathcal{T}_m}\left(\widehat{\ell}_{s,a^\star} - \widehat{\ell}_{s,k}\right) \leq 7\sqrt{g_m T_m}, \text{ where } k \in \underset{a \in \mathcal{A}_m}{\operatorname{argmin}} \sum_{s \in \mathcal{T}_m}\widehat{\ell}_{s,a}, \tag{22}$$

which does not meet the elimination rule in Eq. (20).

For analysis purpose, we decompose Eq. (22) as

$$\sum_{s \in \mathcal{T}_m}\left(\widehat{\ell}_{s,a^\star} - \widehat{\ell}_{s,k}\right) = \sum_{s \in \mathcal{T}_m}\left(\widehat{\ell}_{s,a^\star} - \ell_{s,a^\star}\right) + \sum_{s \in \mathcal{T}_m}\left(\ell_{s,a^\star} - \ell_{s,k}\right) + \sum_{s \in \mathcal{T}_m}\left(\ell_{s,k} - \widehat{\ell}_{s,k}\right). \tag{23}$$

The first and the third term in Eq. (23) can be handled thanks to the bound on $\left|\sum_t(\widehat{\ell}_{t,a} - \ell_{t,a})\right|$ in Eq. (21), and the second term in Eq. (23) can be handled by invoking the standard concentration inequality as losses are drawn from fixed distributions.

Then, we show that if an arm is still active at a round $t$, then, for all active arms $a \in \mathcal{A}_t$, $\Delta_a$ is bounded by a function, monotonically decreasing for large $t$. This can be proved again via a similar decomposition of Eq. (23) to get

$$\sum_{s \in \mathcal{T}_m}\left(\widehat{\ell}_{s,a} - \widehat{\ell}_{s,k}\right) \geq 0.5 T_m \Delta_a - 5\sqrt{g_m T_m}.$$

Therefore, for some large $T_m$ such that $0.5 T_m \Delta_a - 5\sqrt{g_m T_m} > 7\sqrt{g_m T_m}$, bad arms will be eliminated. Putting two pieces together, we complete the proof. $\square$

With Theorem F.4 in hand, our next objective is to show that EXP3.P and GEOMETRICHEDGE.P are able to satisfy Condition. F.1. For GEOMETRICHEDGE.P with John's exploration (which improves the original bound [Bartlett et al., 2008]), Lee et al. [2021] have already shown that it achieves the condition, and the rest result will show that EXP3.P also holds it. The full proof can be found in Appendix F.3.

**Proposition F.5.** *In MAB setting, for any* $\delta \in (0, 1)$, EXP3.P *with given arm set* $\mathcal{A}$, *satisfies Condition. F.1 with* $g(\delta) = \mathcal{O}\left(|\mathcal{A}| \log\left(|\mathcal{A}|/\delta\right)\right)$.

The key that EXP3.P as well as GEOMETRICHEDGE.P can fulfill Condition. F.1 is because it adds a small amount of probability for uniform exploration to each arm. Therefore, the importance-weighted (IW) estimator can be lower-bounded and the term $\left|\sum_t(\widehat{\ell}_{t,a} - \ell_{t,a})\right|$ will not be too large.

Apart from EXP3.P for MAB and GEOMETRICHEDGE.P for linear bandits, Lee et al. [2021] show that refined version of SCRIBLE [Lee et al., 2020] can be used as a base-algorithm for Algorithm 10 to achieve the ULI guarantee for linear bandits, but the ULI guarantee is inferior to that of GEOMETRICHEDGE.P (i.e., inferior $F_{\text{ULI}}$ w.r.t. $d$).

## F.2 Proof of Theorem F.4

**Lemma F.6.** *If Algorithm 10 accepts a base-algorithm which satisfies Condition. F.1 as an input, then, with probability at least* $1 - \frac{3}{5}\delta$, *for all* $m \in \mathbb{N}$ *and* $a \in \mathcal{A}_m$,

$$\sum_{s \in \mathcal{T}_m} (\ell_{s,A_s} - \ell_{s,a}) \leq \sqrt{g_m T_m} - 2 \left|\sum_{s \in \mathcal{T}_m} \left(\ell_{s,a} - \widehat{\ell}_{s,a}\right)\right|. \tag{24}$$

*Proof.* We first consider a fixed phase $m$. Since the base-algorithm satisfies Condition. F.1, if the base-algorithm runs for consecutive $T_m$ and active arm set $\mathcal{A}_m \subseteq \mathcal{A}$, then, with probability at least $1 - \frac{3}{5}\delta'$, for all $a \in \mathcal{A}_m$

$$\sum_{s \in \mathcal{T}_m} (\ell_{s,A_s} - \ell_{s,a}) \leq \sqrt{g(\delta')T_m} - 2 \left|\sum_{s \in \mathcal{T}_m} \left(\ell_{s,a} - \widehat{\ell}_{s,a}\right)\right|. $$

By setting $\delta' = \delta/(2m^2)$ for phase $m$ and applying a union bound over all $m \in \mathbb{N}$, we complete the proof. $\qquad \square$

Recall from the second bullet of Condition. F.1 that

$$g(\delta') \geq \log\left(10|\mathcal{A}|/\delta'\right), \quad \forall \delta' \in (0, 1).$$

As the above holds for all $\delta' \in (0, 1)$, according to the definition $g_m = g(\delta/(2m^2))$, we also have that

$$g_m \geq \log\left(\frac{20m^2|\mathcal{A}|}{\delta}\right), \quad \forall m \in \mathbb{N}. \tag{25}$$

**Lemma F.7.** *With probability at least* $1 - \delta/5$, *for all* $m \in \mathbb{N}$ *and* $a \in \mathcal{A}_m$, *we have*

$$\left|\sum_{s \in \mathcal{T}_m} \ell_{s,a} - \sum_{s \in \mathcal{T}_m} (1 - \mu_a)\right| \leq \sqrt{\frac{g_m T_m}{2}}. \tag{26}$$

*Proof.* Consider any fixed $m$ and $a \in \mathcal{A}_m$. By applying Hoeffding's inequality with probability at least $1 - \delta'$,

$$\left|\sum_{s \in \mathcal{T}_m} \ell_{s,a} - \sum_{s \in \mathcal{T}_m} (1 - \mu_a)\right| \leq \sqrt{\frac{T_m \log(2/\delta')}{2}}.$$

Choosing $\delta' = \delta/(10|\mathcal{A}|m^2)$, applying union bounds over all $m \in \mathbb{N}$ and $a \in \mathcal{A}_m$, and using Eq. (25), we complete the proof. $\qquad \square$

**Lemma F.8.** *With probability at least $1 - \delta/5$, for all $m \in \mathbb{N}$, we have*

$$\left| \sum_{s \in \mathcal{T}_m} \ell_{s,A_s} - \sum_{s \in \mathcal{T}_m} \sum_{a \in \mathcal{A}_m} p_{s,a}(1 - \mu_a) \right| \leq \sqrt{8g_m T_m}. \tag{27}$$

*Proof.* Consider a fixed $m$. For each round $s$ in this phase, let us define

$$M_s = \ell_{s,A_s} - \mathbb{E}_s\left[\ell_{s,A_s}\right] = \sum_{a \in \mathcal{A}_m} \left(\ell_{s,a} B_{s,a} - p_{s,a}(1 - \mu_a)\right).$$

By applying Azuma-inequality with the fact that $|M_s - M_{s-1}| \leq 2$ for all $s$ and also using a union bound over all $m \in \mathbb{N}$, with probability at least $1 - \delta/5$, for all $m \in \mathbb{N}$

$$\left| \sum_{s \in \mathcal{T}_m} \ell_{s,A_s} - \sum_{s \in \mathcal{T}_m} \sum_{a \in \mathcal{A}_m} p_{s,a}(1 - \mu_a) \right| \leq \sqrt{8T_m \log(20m^2/\delta)} \leq \sqrt{8g_m T_m},$$

where the last inequality uses Eq. (25), which thus completes the proof. $\qquad\square$

**Definition F.9.** *Let $\mathcal{E}$ be the event in which the concentration inequalities in Lemma F.6, Lemma F.7, and Lemma F.8 hold simultaneously.*

By a union bound, we have the following fact.

**Fact.** With the definition of $\mathcal{E}$ in Definition F.9, $\mathcal{E}$ occurs with probability at least $1 - \delta$.

**Lemma F.10.** *Suppose that $\mathcal{E}$ occurs where $\mathcal{E}$ is defined Definition F.9, and then for all $m \in \mathbb{N}$ and $a \in A_m$,*

$$2 \left| \sum_{s \in \mathcal{T}_m} \left( \ell_{s,a} - \widehat{\ell}_{s,a} \right) \right| \leq 5\sqrt{g_m T_m} + T_m \Delta_a.$$

*Proof.* Conditioning on $\mathcal{E}$, for each $m \in \mathbb{N}$ and $a \in A_m$, one can show

$$2 \left| \sum_{s \in \mathcal{T}_m} \left( \ell_{s,a} - \widehat{\ell}_{s,a} \right) \right|$$

$$\leq \sqrt{g_m T_m} + \sum_{s \in \mathcal{T}_m} \left( \ell_{s,a} - \ell_{s,A_s} \right)$$

$$\leq \sqrt{g_m T_m} - \sum_{s \in \mathcal{T}_m} \sum_{j \in \mathcal{A}_m} p_{s,j}(1 - \mu_j) + \sqrt{8g_m T_m} + \sum_{s \in \mathcal{T}_m} (1 - \mu_a) + \sqrt{\frac{g_m T_m}{2}}$$

$$\leq 5\sqrt{g_m T_m} + \sum_{s \in \mathcal{T}_m} \sum_{j \in \mathcal{A}_m} p_{s,j}(\mu_j - \mu_a)$$

$$\leq 5\sqrt{g_m T_m} + T_m \Delta_a,$$

where the first inequality holds due to Lemma F.6, the second inequality uses Lemma F.7 and Lemma F.8, and the last inequality bounds $\mu_j \leq \mu^\star$ for all $j$. $\qquad\square$

**Corollary F.11.** *Suppose that $\mathcal{E}$ occurs where $\mathcal{E}$ is defined Definition F.9, and then for all $m \in \mathbb{N}$ and $a \in \mathcal{A}$, we have*

$$\sum_{s \in \mathcal{T}_m} \ell_{s,a^\star} - \sum_{s \in \mathcal{T}_m} \ell_{s,a} \leq -T_m \Delta_a + \sqrt{2g_m T_m}, \tag{28}$$

$$\sum_{s \in \mathcal{T}_m} \ell_{s,a} - \sum_{s \in \mathcal{T}_m} \ell_{s,a^\star} \leq T_m \Delta_a + \sqrt{2g_m T_m}. \tag{29}$$

*Proof.* The proof is immediate by applying Lemma F.7. $\qquad\square$

**Lemma F.12.** *Suppose that $\mathcal{E}$ occurs where $\mathcal{E}$ is defined Definition F.9, and then $a^\star \in \mathcal{A}_m$ for all $m \in \mathbb{N}$.*

*Proof.* We prove the claimed result by induction. For the base case, the claim holds for the first phase $m = 1$ as $\mathcal{A}_1 = \mathcal{A}$. Suppose that $a^\star \in \mathcal{A}_m$. Then, we show that $a^\star$ will not be eliminated at the end of phase $m$, thereby active for phase $m + 1$. Then, we have

$$
\begin{aligned}
&\sum_{s \in \mathcal{T}_m} \left( \widehat{\ell}_{s,a^\star} - \widehat{\ell}_{s,k} \right) \\
&= \sum_{s \in \mathcal{T}_m} \left( \widehat{\ell}_{s,a^\star} - \ell_{s,a^\star} \right) + \sum_{s \in \mathcal{T}_m} \left( \ell_{s,a^\star} - \ell_{s,k} \right) + \sum_{s \in \mathcal{T}_m} \left( \ell_{s,k} - \widehat{\ell}_{s,k} \right) \\
&\leq \frac{5}{2} \sqrt{g_m T_m} + \left( -T_m \Delta_k + \sqrt{2 g_m T_m} \right) + \frac{1}{2} \left( 5 \sqrt{g_m T_m} + T_m \Delta_k \right) \\
&= \frac{13}{2} \sqrt{g_m T_m} - 0.5 T_m \Delta_k \\
&< 7 \sqrt{g_m T_m},
\end{aligned}
$$

where the first inequality uses Lemma F.10 together with $\Delta_{a^\star} = 0$ and Corollary F.11.

Once the induction is done, we complete the proof (recall the elimination rule in Algorithm 10).  □

**Lemma F.13.** *Suppose that $\mathcal{E}$ occurs where $\mathcal{E}$ is defined Definition F.9. For each arm $a \in \mathcal{A}$ with $\Delta_a > 0$, it will not be in $\mathcal{A}_m$ for all $m \geq m_a + 1$, where $m_a$ is the smallest phase such that $2^{m_a} > \frac{1}{\Delta_a^2}$.*

*Proof.* Consider fixed phase $m$ and arm $a \in \mathcal{A}$ with $\Delta_a > 0$. Suppose that arm $a$ is still active in an phase $m$. One can show

$$
\begin{aligned}
&\sum_{s \in \mathcal{T}_m} \left( \widehat{\ell}_{s,a} - \widehat{\ell}_{s,k} \right) \\
&\geq \sum_{s \in \mathcal{T}_m} \left( \widehat{\ell}_{s,a} - \widehat{\ell}_{s,a^\star} \right) \\
&= \sum_{s \in \mathcal{T}_m} \left( \widehat{\ell}_{s,a} - \ell_{s,a} \right) + \sum_{s \in \mathcal{T}_m} \left( \ell_{s,a} - \ell_{s,a^\star} \right) + \sum_{s \in \mathcal{T}_m} \left( \ell_{s,a^\star} - \widehat{\ell}_{s,a^\star} \right) \\
&\geq \frac{-1}{2} \left( 5 \sqrt{g_m T_m} + T_m \Delta_a \right) + \left( T_m \Delta_a - \sqrt{2 g_m T_m} \right) + \frac{-5}{2} \sqrt{g_m T_m} \\
&\geq 0.5 T_m \Delta_a - 5 \sqrt{g_m T_m}, \quad\quad\quad\quad\quad\quad\quad\quad\quad\quad\quad (30)
\end{aligned}
$$

where the first inequality holds since Lemma F.12 implies that $a^\star \in \mathcal{A}_m$ for all $m \in \mathbb{N}$, and the second inequality uses Lemma F.10 together with $\Delta_{a^\star} = 0$ and Corollary F.11.

Let $m_a$ be the minimum phase such that $2^{m_a} > \frac{1}{\Delta_a^2}$ (i.e., $2^{m_a - 1} \leq \frac{1}{\Delta_a^2}$), which further gives that

$$
T_{m_a} = \lceil 576 g_{m_a} 2^{m_a} \rceil \geq 576 g_{m_a} 2^{m_a} > \frac{576 g_{m_a}}{\Delta_a^2}.
$$

Hence, by the definition of $m_a$, we have $T_m > \frac{576 g_m}{\Delta_a^2}$ for all $m \geq m_a$, which gives that $T_m \Delta_a > 24 \sqrt{g_m T_m}$ for all $m \geq m_a$. Plugging this into Eq. (30), we arrive at

$$
\sum_{s \in \mathcal{T}_m} \left( \widehat{\ell}_{s,a} - \widehat{\ell}_{s,k} \right) \geq 0.5 T_m \Delta_a - 5 \sqrt{g_m T_m} > 7 \sqrt{g_m T_m},
$$

which implies that arm $a$ will not be active in all phases $m \geq m_a + 1$ according to the elimination rule.

Finally, one can repeat this argument for each $a \in \mathcal{A}$ with $\Delta_a > 0$ conditioning on $\mathcal{E}$.  □

**Lemma F.14.** *Let $m(t)$ be the phase in which round $t$ lies. Then, $m(t) \leq \log_2(t + 1)$ for all $t \in \mathbb{N}$.*

*Proof.* We prove this by contradiction. Suppose that $\exists t \in \mathbb{N}$ that $m(t) > \log_2(t + 1)$. Note that we can further assume $m(t) \geq 2$ since one can easily verify that for all $t$ such that $m(t) = 1$,

$m(t) \le \log_2(t+1)$ must hold. Recall that in phase $m(t)$, each active arm will be played for $m_{\ell(t)}$ times, we have

$$t \ge T_{m(t)-1} \ge 576 g_{m(t)-1} 2^{m(t)-1} > 288(t+1) g_{m(t)-1} > t,$$

where the third inequality bounds $\ell(t) > \log_2(t+1)$ by assumption. Therefore, once a contradiction occurs, the proof is complete. $\square$

**Lemma F.15.** *Let $m(t)$ be the phase in which round $t$ lies. Suppose that $\mathcal{E}$ occurs. For all $t \in \mathbb{N}$ and all $a \in \mathcal{A}$, if $a \in \mathcal{A}_{m(t)}$, then,*

$$\Delta_a \le \sqrt{\frac{4608 g(\delta/(2\log_2^2(t+1)))}{t}}.$$

*Proof.* If $a \in \mathcal{A}_{m(t)}$ is optimal, then, $\Delta_a = 0$ and the claim trivially holds. In what follows, we only consider arm $a \in \mathcal{A}_{m(t)}$ with $\Delta_a > 0$. Then, $t$ can be bounded by

$$t \le \sum_{n=1}^{m(t)} \lceil 576 g_n 2^n \rceil$$

$$\le 1152 \sum_{n=1}^{m(t)} g_n 2^n$$

$$\le 1152 g_{m(t)} \sum_{n=1}^{m(t)} 2^n$$

$$\le 1152 g_{m(t)} \sum_{n=1}^{m_a} 2^n$$

$$\le \frac{4608 g_{m(t)}}{\Delta_a^2},$$

where the second inequality simply bounds $\lceil 576 g_n 2^n \rceil \le 2 \times 576 g_n 2^n$ for all phases $n$, the fourth inequality holds because Lemma F.13 implies that if $a \in \mathcal{A}_{m(t)}$, then, $m(t) \le m_a$ holds, and the last inequality uses $\frac{1}{\Delta_a^2} \ge 2^{m_a - 1}$.

Since for any fixed $\delta \in (0,1)$, $g(\delta/x^2)$ is monotonically increasing for $x \ge 1$ (recall Condition. F.1) and Lemma F.14 gives $m(t) \le \log_2(t+1)$, we have

$$g_{m(t)} = g(\delta/(2m(t)^2)) \le g(\delta/(2\log_2^2(t+1))),$$

which gives $t \le \frac{4608 g(\delta/(2\log_2^2(t+1)))}{\Delta_a^2}$. Conditioning on $\mathcal{E}$, this argument holds for each $t, a$, which completes the proof. $\square$

*Proof of Theorem F.4.* Once Lemma F.12 and Lemma F.15 hold, and $g(x)$ is polynomial in $\log(1/x)$, Theorem 3.1 gives that for any fixed $\delta \in (0,1)$, Algorithm 10 achieves the ULI guarantee with a function

$$F_{\mathrm{ULI}}(\delta, t) = \mathcal{O}\left(\sqrt{\frac{g(\delta/(2\log_2^2(t+1)))}{t}}\right).$$

According to the first bullet of Condition. F.1, the proof is complete. $\square$

## F.3 Proof of Proposition F.5

In this section, we prove that EXP3.P [Auer et al., 2002b] meets Condition. F.1. In our setting, arm set is $\mathcal{A} = [K]$, and the loss $\ell_{t,a}$ generated by each arm $a$ at each round $t$ is assumed to be $[0,1]$-bounded.

---

**Algorithm 11** EXP3.P

---

**Input**: Time horizon $T$, arm set $[K]$, confidence $\delta \in (0, 1)$.
**Initialize**: $\forall a \in [K]$, $w_{1,a} = 1$ and parameters $\eta = \eta_\delta(T), \gamma = \gamma_\delta(T), \beta = \beta_\delta(T)$ according to Eq. (33).
**for** $t = 1, 2, \ldots, T$ **do**

Play an arm $A_t \in [K]$ from distribution $p_t = [p_{t,1}, \ldots, p_{t,K}]$ and observe loss $\ell_{t,A_t}$ where

$$p_{t,a} = (1 - \gamma)\frac{w_{t,a}}{W_t} + \frac{\gamma}{K} \quad \text{where} \quad W_t = \sum_{a \in [K]} w_{t,a}. \tag{31}$$

Update $w_{t+1,a} = w_{t,a} \exp(-\eta \widetilde{\ell}_{t,a})$ for all $a \in [K]$ with

$$\widetilde{\ell}_{t,a} = \widehat{\ell}_{t,a} - \frac{2\beta}{p_{t,a}}, \quad \text{where} \quad \widehat{\ell}_{t,a} = \frac{\ell_{t,a} B_{t,a}}{p_{t,a}}, \text{ and } B_{t,a} = \mathbb{I}\{A_t = a\}. \tag{32}$$

---

The pseudocode of EXP3.P is given in Algorithm 11 and we briefly review its procedure. Ahead of time, EXP3.P accepts a fixed time horizon $T \in \mathbb{N}$ arm set $[K]$, and confidence $\delta \in (0, 1)$ as inputs. At each round $t \in [T]$, EXP3.P pulls an arm $A_t \sim p_t$ from a distribution $p_t = [p_{t,1}, \ldots, p_{t,K}]$, and then observes the loss $\ell_{t,A_t}$. The probability of playing an arm $a$ at round $t$ is $(1 - \gamma)\frac{w_{t,a}}{W_t} + \frac{\gamma}{K}$ where $\gamma > 0$ is a fixed parameter, which encourages the exploration, $w_{t,a}$ is the weight of arm $a$, and $W_t = \sum_{a \in [K]} w_{t,a}$. After pulling the arm, the algorithm uses the observed reward to construct the shifted IW-estimators $\widetilde{\ell}_{t,a}$ according to Eq. (32), and finally uses the shifted IW-estimators to update the weight $w_{t,a}$ for each arm.

The parameters of $\gamma_\delta(T), \eta_\delta(T), \beta_\delta(T)$ are as a function $T$, given as

$$\gamma_\delta(T) = \min\left\{\frac{1}{2}, \sqrt{\frac{K\log(10K/\delta)}{T}}\right\}, \quad \beta_\delta(T) = \eta_\delta(T) = \frac{\gamma_\delta(T)}{K}. \tag{33}$$

Let $\mathbb{E}_t[\cdot]$ be the conditional expectation given the history prior to round $t$.

**Lemma F.16** (Exercise 5.15 of [Lattimore and Szepesvári, 2020])**.** *Let $\{X_t\}_{t=1}^T$ be a sequence of random variables adapted to filtration $\{\mathcal{F}_t\}_{t=1}^T$ and let $\beta > 0$ such that $\beta X_t \leq 1$ almost surely for all $t \in [T]$. With probability at least $1 - \delta$,*

$$\sum_{t=1}^T (X_t - \mathbb{E}_t[X_t]) \leq \beta \sum_{t=1}^T \mathbb{E}_t[X_t^2] + \frac{\log(1/\delta)}{\beta}.$$

**Corollary F.17.** *With probability at least $1 - \delta/5$, for all $a \in [K]$,*

$$\sum_{t=1}^T \left(\widetilde{\ell}_{t,a} - \ell_{t,a}\right) \leq \frac{\log(5K/\delta)}{\beta}.$$

*Proof.* Consider a fixed arm $a$. For all $t, a$, we have $\ell_{t,a} \in [0, 1]$, and thus

$$\mathbb{E}_t\left[\widehat{\ell}_{t,a}^2\right] = \mathbb{E}_t\left[\frac{\ell_{t,a}^2 B_{t,a}^2}{p_{t,a}^2}\right] = \mathbb{E}_t\left[\frac{\ell_{t,a}^2 B_{t,a}}{p_{t,a}^2}\right] \leq \mathbb{E}_t\left[\frac{B_{t,a}}{p_{t,a}^2}\right] = \frac{1}{p_{t,a}}. \tag{34}$$

Then, we check $\beta\widehat{\ell}_{t,a} \leq 1$ almost surely for all $t \in [T]$. Specifically, we use $p_{t,a} \geq \gamma/K$ to show

$$\beta\widehat{\ell}_{t,a} = \beta\frac{B_{t,a}\ell_{t,a}}{p_{t,a}} \leq \beta\frac{1}{p_{t,a}} \leq \frac{\beta K}{\gamma} = 1,$$

where the last equality holds due to $\beta = \gamma/K$ according to Eq. (33).

For the fixed $a$, applying Lemma F.16 with $X_t = \widehat{\ell}_{t,a}$, we have that with probability at least $1 - \delta'$

$$\sum_{t=1}^T \left(\widetilde{\ell}_{t,a} - \ell_{t,a}\right) = \sum_{t=1}^T \left(\widehat{\ell}_{t,a} - \ell_{t,a}\right) - \sum_{t=1}^T \frac{2\beta}{p_{t,a}}$$

$$\leq \beta \sum_{t=1}^{T} \frac{1}{p_{t,a}} + \frac{\log(1/\delta')}{\beta} - \sum_{t=1}^{T} \frac{2\beta}{p_{t,a}}$$

$$\leq \frac{\log(1/\delta')}{\beta}.$$

Choosing $\delta' = \delta/(5K)$ and applying a union bound over all $a \in [K]$ yield the claimed bound. $\qquad\square$

**Lemma F.18.** *With probability at least $1 - \delta/5$, for all $a \in [K]$, we have*

$$\left| \sum_{t=1}^{T} \ell_{t,a} - \sum_{t=1}^{T} (1 - \mu_a) \right| \leq \sqrt{\frac{T \log(10K/\delta)}{2}}. \tag{35}$$

*Proof.* Consider any fixed arm $a$. By applying Hoeffding's inequality with probability at least $1 - \delta'$,

$$\left| \sum_{t=1}^{T} \ell_{s,a} - \sum_{t=1}^{T} (1 - \mu_a) \right| \leq \sqrt{\frac{T \log(2/\delta')}{2}}.$$

Choosing $\delta' = \delta/(5K)$ and applying a union bound over all arms, we complete the proof. $\qquad\square$

**Lemma F.19** (Freedman's inequality). *Let $\{X_t\}_{t=1}^{T}$ be a martingale difference sequence with respect to filtration $\{\mathcal{F}_t\}_{t=1}^{T}$ and $|X_t| \leq M$ almost surely for all $t$. Then, for any $\delta \in (0, 1)$, with probability at least $1 - \delta$,*

$$\left| \sum_{t=1}^{T} X_t \right| \leq \frac{2M}{3} \log(2/\delta) + \sqrt{2 \log(2/\delta) \sum_{t=1}^{T} \mathbb{E}_t[X_t^2]}.$$

**Corollary F.20.** *With probability at least $1 - \delta/5$, the following holds for all $a$,*

$$\left| \sum_{t=1}^{T} \left( \widehat{\ell}_{t,a} - (1 - \mu_a) \right) \right| \leq \frac{2 \log(10K/\delta)}{3} \frac{K}{\gamma} + \sqrt{2 \log(10K/\delta) \sum_{t=1}^{T} \frac{1}{p_{t,a}}}.$$

*Proof.* Consider a fixed arm $a$. Let $M_{t,a} = \widehat{\ell}_{t,a} - (1 - \mu_a)$ and $\{M_{t,a}\}_{t=1}^{T}$ is a martingale difference sequence. We have $\mathbb{E}_t[M_{t,a}] = 0$, $|M_{t,a}| \leq \frac{K}{\gamma}$ and also

$$\sqrt{\sum_{t=1}^{T} \mathbb{E}_t \left[ M_{t,a}^2 \right]} \leq \sqrt{\sum_{t=1}^{T} \mathbb{E}_t \left[ \widehat{\ell}_{t,a}^2 \right]} \leq \sqrt{\sum_{t=1}^{T} \frac{1}{p_{t,a}}},$$

where the last step holds due to Eq. (34).

By Lemma F.19 and a union bound over all $a$, with probability at least $1 - \delta'$ for all $a$

$$\left| \sum_{t=1}^{T} \left( \widehat{\ell}_{t,a} - (1 - \mu_a) \right) \right| \leq \frac{2 \log(2K/\delta')}{3} \frac{K}{\gamma} + \sqrt{2 \log(2K/\delta') \sum_{t=1}^{T} \frac{1}{p_{t,a}}}.$$

Choosing $\delta' = \delta/5$ completes the proof. $\qquad\square$

**Definition F.21.** *Let $\mathcal{E}_0$ be the event in which all inequalities of Corollary F.17, Lemma F.18, and Corollary F.20 hold simultaneously. With this definition, $\mathcal{E}_0$ occurs with probability at least $1 - \frac{3\delta}{5}$.*

**Lemma F.22.** *Let $\theta_0$ be an arbitrary constant such that $\theta_0 \geq 1$. Suppose that $\mathcal{E}_0$ occurs where $\mathcal{E}_0$ is defined in Definition F.21, and then for all $a \in [K]$, we have*

$$-\sum_{t=1}^{T} \frac{\beta}{p_{t,a}} \leq -\theta_0 \left| \sum_{t=1}^{T} \left( \widehat{\ell}_{t,a} - \ell_{t,a} \right) \right| + \frac{2K\theta_0 \log(10K/\delta)}{3\gamma} + \frac{\theta_0^2 \log(10K/\delta)}{\beta} + \theta_0 \sqrt{\frac{T \log(10K/\delta)}{2}} \tag{36}$$

$$\sum_{t=1}^{T} \left( \widetilde{\ell}_{t,a} - \ell_{t,a} \right) \leq \frac{2K \log(10K/\delta)}{3\gamma} + \frac{\theta_0 \log(10K/\delta)}{\beta} + \sqrt{\frac{T \log(10K/\delta)}{2}} - \beta \sum_{t=1}^{T} \frac{1}{p_{t,a}}, \tag{37}$$

*Proof.* Let $\theta_0 > 0$ be an arbitrary constant. By Corollary F.20, for all $a \in [K]$

$$\left| \sum_{t=1}^{T} \left( \widehat{\ell}_{t,a} - (1 - \mu_a) \right) \right| \leq \frac{2 \log(10K/\delta)}{3} \frac{K}{\gamma} + \sqrt{2 \log(10K/\delta) \sum_{t=1}^{T} \frac{1}{p_{t,a}}}$$

$$= \frac{2 \log(10K/\delta)}{3} \frac{K}{\gamma} + \sqrt{2 \frac{\theta_0 \log(10K/\delta)}{\beta} \cdot \frac{\beta}{\theta_0} \sum_{t=1}^{T} \frac{1}{p_{t,a}}}$$

$$\leq \frac{2 \log(10K/\delta)}{3} \frac{K}{\gamma} + \frac{\theta_0 \log(10K/\delta)}{\beta} + \frac{\beta}{\theta_0} \sum_{t=1}^{T} \frac{1}{p_{t,a}}, \tag{38}$$

where the second inequality uses $\sqrt{2ab} \leq a + b$ for all $a, b \geq 0$.

Moreover, by the triangle inequality, we have

$$\left| \sum_{t=1}^{T} \left( \widehat{\ell}_{t,a} - \ell_{t,a} \right) \right| \leq \left| \sum_{t=1}^{T} \left( \widehat{\ell}_{t,a} - (1 - \mu_a) \right) \right| + \left| \sum_{t=1}^{T} (\ell_{t,a} - (1 - \mu_a)) \right|$$

$$\leq \frac{2 \log(10K/\delta)}{3} \frac{K}{\gamma} + \frac{\theta_0 \log(10K/\delta)}{\beta} + \frac{\beta}{\theta_0} \sum_{t=1}^{T} \frac{1}{p_{t,a}} + \sqrt{\frac{T \log(10K/\delta)}{2}}, \tag{39}$$

where the last inequality applies Lemma F.18 and uses Eq. (38).

Rearranging the above gives

$$-\sum_{t=1}^{T} \frac{\beta}{p_{t,a}} \leq -\theta_0 \left| \sum_{t=1}^{T} \left( \widehat{\ell}_{t,a} - \ell_{t,a} \right) \right| + \frac{2\theta_0 \log(10K/\delta)}{3} \frac{K}{\gamma} + \frac{\theta_0^2 \log(10K/\delta)}{\beta} + \theta_0 \sqrt{\frac{T \log(10K/\delta)}{2}}.$$

Finally, if we constrain $\theta_0 \geq 1$, then

$$\sum_{t=1}^{T} \left( \widetilde{\ell}_{t,a} - \ell_{t,a} \right)$$

$$= \sum_{t=1}^{T} \left( \widehat{\ell}_{t,a} - \ell_{t,a} \right) - \sum_{t=1}^{T} \frac{2\beta}{p_{t,a}}$$

$$\leq \frac{2 \log(10K/\delta)}{3} \frac{K}{\gamma} + \frac{\theta_0 \log(10K/\delta)}{\beta} + \sqrt{\frac{T \log(10K/\delta)}{2}} + \frac{1}{\theta_0} \sum_{t=1}^{T} \frac{\beta \ell_{t,a}}{p_{t,a}} - \sum_{t=1}^{T} \frac{2\beta}{p_{t,a}}$$

$$\leq \frac{2 \log(10K/\delta)}{3} \frac{K}{\gamma} + \frac{\theta_0 \log(10K/\delta)}{\beta} + \sqrt{\frac{T \log(10K/\delta)}{2}} - \sum_{t=1}^{T} \frac{\beta}{p_{t,a}},$$

where the first inequality uses Eq. (39) and the last one bounds $\theta_0 \geq 1$.

Since this argument holds for all $a \in [K]$ conditioning on $\mathcal{E}_0$, we complete the proof. $\square$

**Lemma F.23.** *The following holds for all $t \in [T]$.*

$$\sum_{a=1}^{K} p_{t,a} \widetilde{\ell}_{t,a}^2 \leq \sum_{a=1}^{K} \widetilde{\ell}_{t,a} + \frac{4K^2\beta^2}{\gamma}.$$

*Proof.* For any fixed $t$, we can show

$$\sum_{a=1}^{K} p_{t,a} \widetilde{\ell}_{t,a}^2 = \sum_{a=1}^{K} p_{t,a} \left( \widehat{\ell}_{t,a} - \frac{2\beta}{p_{t,a}} \right)^2$$

$$= \sum_{a=1}^{K} p_{t,a} \left( \widehat{\ell}_{t,a}^2 - \frac{4\beta}{p_{t,a}} \widehat{\ell}_{t,a} + \frac{4\beta^2}{p_{t,a}^2} \right)$$

$$= \sum_{a=1}^{K} p_{t,a} \widehat{\ell}_{t,a} \left( \widehat{\ell}_{t,a} - \frac{4\beta}{p_{t,a}} \right) + \sum_{a=1}^{K} \frac{4\beta^2}{p_{t,a}}$$

$$\leq \sum_{a=1}^{K} p_{t,a} \widehat{\ell}_{t,a} \widetilde{\ell}_{t,a} + \frac{4K^2\beta^2}{\gamma}$$

$$\leq \sum_{a=1}^{K} \widetilde{\ell}_{t,a} + \frac{4K^2\beta^2}{\gamma},$$

where the first inequality uses $\widehat{\ell}_{t,a} - \frac{4\beta}{p_{t,a}} \leq \widehat{\ell}_{t,a} - \frac{2\beta}{p_{t,a}} = \widetilde{\ell}_{t,a}$ and $p_{t,a} \geq \frac{\gamma}{K}$, and the last inequality bounds $p_{t,a} \widehat{\ell}_{t,a} \leq 1$ for each $a \in [K]$. Since the claim deterministically holds for all $t$, the proof is complete. $\qquad \square$

**Lemma F.24.** *The following holds for all arms $k \in [K]$.*

$$\sum_{t=1}^{T} \sum_{a=1}^{K} p_{t,a} \widetilde{\ell}_{t,a} - \sum_{t=1}^{T} \widetilde{\ell}_{t,k} \leq \frac{\log(K)}{\eta} + 2\eta \sum_{a=1}^{K} \sum_{t=1}^{T} \widetilde{\ell}_{t,a} + 4TK\beta^2.$$

*Proof.* Let us define

$$\Phi_0 = 0, \quad \Phi_t = \frac{1-\gamma}{\eta} \log \left( \frac{1}{K} \sum_{a=1}^{K} \exp \left( -\eta \sum_{s=1}^{t} \widetilde{\ell}_{s,a} \right) \right), \quad \forall t \geq 1.$$

With this definition, one can show for $t \geq 2$

$$\Phi_t - \Phi_{t-1} = \frac{1-\gamma}{\eta} \log \left( \frac{\sum_{a=1}^{K} \exp \left( -\eta \sum_{s=1}^{t} \widetilde{\ell}_{s,a} \right)}{\sum_{a=1}^{K} \exp \left( -\eta \sum_{s=1}^{t-1} \widetilde{\ell}_{s,a} \right)} \right)$$

$$= \frac{1-\gamma}{\eta} \log \left( \frac{\sum_{a=1}^{K} \exp \left( -\eta \sum_{s=1}^{t-1} \widetilde{\ell}_{s,a} \right)}{\sum_{a=1}^{K} \exp \left( -\eta \sum_{s=1}^{t-1} \widetilde{\ell}_{s,a} \right)} \exp \left( -\eta \widetilde{\ell}_{t,a} \right) \right)$$

$$= \frac{1-\gamma}{\eta} \log \left( \sum_{a=1}^{K} \frac{w_{t,a}}{W_t} \exp \left( -\eta \widetilde{\ell}_{t,a} \right) \right)$$

$$\leq \frac{1-\gamma}{\eta} \log \left( \sum_{a=1}^{K} \frac{w_{t,a}}{W_t} \left( 1 - \eta \widetilde{\ell}_{t,a} + \eta^2 \widetilde{\ell}_{t,a}^2 \right) \right)$$

$$= \frac{1-\gamma}{\eta} \log \left( 1 + \eta \sum_{a=1}^{K} \frac{w_{t,a}}{W_t} \left( \eta \widetilde{\ell}_{t,a}^2 - \widetilde{\ell}_{t,a} \right) \right)$$

$$= \frac{1-\gamma}{\eta} \log \left( 1 + \frac{\eta}{1-\gamma} \sum_{a=1}^{K} \left( p_{t,a} - \frac{\gamma}{K} \right) \left( \eta \widetilde{\ell}_{t,a}^2 - \widetilde{\ell}_{t,a} \right) \right)$$

$$\leq -\sum_{a=1}^{K} \left( p_{t,a} - \frac{\gamma}{K} \right) \widetilde{\ell}_{t,a} + \eta \sum_{a=1}^{K} \left( p_{t,a} - \frac{\gamma}{K} \right) \widetilde{\ell}_{t,a}^2$$

$$\leq -\sum_{a=1}^{K} p_{t,a} \widetilde{\ell}_{t,a} + \frac{\gamma}{K} \sum_{a=1}^{K} \widetilde{\ell}_{t,a} + \eta \sum_{a=1}^{K} p_{t,a} \widetilde{\ell}_{t,a}^2,$$

where the first inequality uses $e^{-x} \leq 1 - x + x^2$ whenever $x \geq -1$ (here we repeat $x = \eta \widetilde{\ell}_{t,a}$ for every $a$), and the second inequality follows from $\log(1 + x) \leq x$ for all $x > -1$.

By summing over all $t$ and using Lemma F.23 with $\eta = \frac{\gamma}{K}$, the above result yields

$$\sum_{t=1}^{T}\sum_{a=1}^{K} p_{t,a}\widetilde{\ell}_{t,a} \le \sum_{t=1}^{T}(\Phi_{t-1} - \Phi_t) + \eta\sum_{t=1}^{T}\sum_{a=1}^{K}\widetilde{\ell}_{t,a} + \eta\sum_{t=1}^{T}\sum_{a=1}^{K} p_{t,a}\widetilde{\ell}_{t,a}^2$$

$$\le \sum_{t=1}^{T}(\Phi_{t-1} - \Phi_t) + 2\eta\sum_{t=1}^{T}\sum_{a=1}^{K}\widetilde{\ell}_{t,a} + 4TK\beta^2.$$

As $\Phi_0 = 0$ and $1 - \gamma \le 1$, we have for an arbitrary arm $k \in [K]$,

$$\sum_{t=1}^{T}(\Phi_{t-1} - \Phi_t) = -\Phi_T$$

$$\le \frac{(1-\gamma)\log(K)}{\eta} - \frac{1-\gamma}{\eta}\log\left(\sum_{a=1}^{K}\exp\left(-\eta\sum_{t=1}^{T}\widetilde{\ell}_{t,a}\right)\right)$$

$$\le \frac{(1-\gamma)\log(K)}{\eta} - \frac{1-\gamma}{\eta}\log\left(\exp\left(-\eta\sum_{t=1}^{T}\widetilde{\ell}_{t,k}\right)\right)$$

$$\le \frac{\log(K)}{\eta} + \sum_{t=1}^{T}\widetilde{\ell}_{t,k}.$$

As this argument deterministically holds for all $k \in [K]$, we complete the proof. $\qquad\square$

*Proof of Proposition F.5.* The following analysis will condition on event $\mathcal{E}_0$ which is defined in Definition F.21. As mentioned in Definition F.21, this event occurs with probability at least $1 - \frac{3}{5}\delta$. We first note that for all $t, a$

$$\sum_{a=1}^{K} p_{t,a}\widetilde{\ell}_{t,a} = \sum_{a=1}^{K} p_{t,a}\left(\frac{\ell_{t,a}B_{t,a}}{p_{t,a}} - \frac{2\beta}{p_{t,a}}\right) = \sum_{a=1}^{K}(\ell_{t,a}B_{t,a} - 2\beta) = \ell_{t,A_t} - 2\beta K. \tag{40}$$

Recall that we choose parameters as

$$\gamma = \min\left\{\frac{1}{2}, \sqrt{\frac{K\log(10K/\delta)}{T}}\right\}, \quad \beta = \eta = \frac{\gamma}{K}. \tag{41}$$

In what follows, we consider $T > 4K\log(10K/\delta)$[8]. In this case, we have $\eta, \gamma, \beta \in (0, \frac{1}{2}]$.

Then, for an arbitrary arm $k \in [K]$, we pick $\theta_0 = 2$ and show

$$\sum_{t=1}^{T}(\ell_{t,A_t} - \ell_{t,k})$$

$$= \sum_{t=1}^{T}\left(\sum_{a=1}^{K} p_{t,a}\widetilde{\ell}_{t,a} - \ell_{t,k}\right) + 2\beta KT$$

$$\le \underbrace{\sum_{t=1}^{T}\left(\sum_{a=1}^{K} p_{t,a}\widetilde{\ell}_{t,a} - \widetilde{\ell}_{t,k}\right)}_{\text{TERM 1}} + \underbrace{2\beta KT + \frac{2\log(10K/\delta)}{3}\frac{K}{\gamma} + \frac{2\log(10K/\delta)}{\beta} + \sqrt{\frac{T\log(10K/\delta)}{2}}}_{\text{TERM 2}} + \underbrace{(-\beta)\sum_{t=1}^{T}\frac{\ell_{t,k}}{p_{t,k}}}_{\text{TERM 3}},$$

where the equality uses Eq. (40) and the inequality applies Lemma F.22.

We first apply Lemma F.24 to bound TERM 1 as

$$\text{TERM 1} \le \frac{\log(K)}{\eta} + 2\eta\sum_{a=1}^{K}\sum_{t=1}^{T}\widetilde{\ell}_{t,a} + 4TK\beta^2$$

---

[8]With the choice of $g(\delta)$ in Proposition F.5, this requirement always holds if EXP3.P is a base-algorithm in Algorithm 10.

$$\leq \frac{\log(K)}{\eta} + 2\eta \sum_{a=1}^{K} \left( \sum_{t=1}^{T} \ell_{t,a} + \frac{\log(10K/\delta)}{\beta} \right) + 4TK\beta^2$$

$$\leq \frac{\log(K)}{\eta} + 2\eta KT + \frac{2\eta K \log(10K/\delta)}{\beta} + 4TK\beta^2$$

$$\leq 3\sqrt{KT \log(10K/\delta)} + 2K \log(10K/\delta) + 4\log(10K/\delta)$$

$$\leq 6\sqrt{KT \log(10K/\delta)},$$

where the second inequality uses Corollary F.17, the third inequality uses $\sum_i \ell_{t,a} \leq K$, and the last inequality uses $T \geq 4K \log(10K/\delta)$ (i.e., $K \leq \sqrt{\frac{KT}{4\log(10K/\delta)}}$).

Then, we use $K/\gamma = 1/\eta$ to bound TERM 2:

$$\text{TERM 2} = 2\beta KT + \frac{2\log(10K/\delta)}{3\eta} + \frac{2\log(10K/\delta)}{\beta} + \sqrt{\frac{T\log(10K/\delta)}{2}}$$

$$= 2\sqrt{KT \log(10K/\delta)} + \frac{2}{3}\sqrt{KT \log(10K/\delta)} + 2\sqrt{KT \log(10K/\delta)} + \sqrt{\frac{T\log(10K/\delta)}{2}}$$

$$\leq \frac{14}{3}\sqrt{KT \log(10K/\delta)} + \sqrt{\frac{T\log(10K/\delta)}{2}}.$$

Finally, we apply Eq. (36) with $\theta_0 = 2$ to bound TERM 3.

$$\text{TERM 3} \leq -2 \left| \sum_{t=1}^{T} \left( \widehat{\ell}_{t,k} - \ell_{t,k} \right) \right| + \frac{4\log(10K/\delta)}{3\eta} + \frac{4\log(10K/\delta)}{\beta} + \sqrt{2T\log(10K/\delta)}$$

$$= -2 \left| \sum_{t=1}^{T} \left( \widehat{\ell}_{t,k} - \ell_{t,k} \right) \right| + \frac{16}{3}\sqrt{KT \log(10K/\delta)} + \sqrt{2T\log(10K/\delta)}.$$

Putting bounds of TERM 1, TERM 2, and TERM 3 together gives

$$\sum_{t=1}^{T} (\ell_{t,A_t} - \ell_{t,k}) \leq -2 \left| \sum_{t=1}^{T} \left( \widehat{\ell}_{t,k} - \ell_{t,k} \right) \right| + 16\sqrt{KT \log(10K/\delta)} + \frac{3\sqrt{2}}{2}\sqrt{T\log(10K/\delta)}$$

$$\leq -2 \left| \sum_{t=1}^{T} \left( \widehat{\ell}_{t,k} - \ell_{t,k} \right) \right| + 19\sqrt{KT \log(10K/\delta)}.$$

As this argument holds for all $k \in [K]$ conditioning on $\mathcal{E}_0$, the proof is thus complete. $\square$

## F.4  Proof of Theorem F.2

For EXP3.P, we apply $g(\delta) = \mathcal{O}(K \log(K/\delta))$ given in Proposition F.5 into Theorem F.4 to get the claimed result.

As for GEOMETRICHEDGE.P, we consider an improved version in [Lee et al., 2021] which uses John's exploration and shows that $g(\delta) = d \log(K/\delta)$. Note that this is slightly different from $d\log(K \log_2 T/\delta)$ presented in [Lee et al., 2021]. The extra $\log_2 T$ term is caused by Lemma 2 of [Bartlett et al., 2008], a concentration inequality for martingales, but one can invoke Lemma F.19 to avoid it. By applying such a function $g(\delta)$ into Theorem F.4, we obtain the claimed result.

---

**Algorithm 12** Adaptive barycentric spanner

---

**Input**: compact arm set $\mathcal{A}$, phase $m$, spanner $\mathcal{B}_{m-1}$, parameters $T_m$, $C$, and estimation $\widehat{\theta}_m$.
**Initialize**: matrix $A = [a_1, \ldots, a_d]$ by setting $a_i = \frac{e_i}{\sqrt{T_m}} \in \mathbb{R}^d$ for all $i \in [d]$, and $\mathcal{I}_m = [d]$.

1   Let $B_{m-1} \in \mathbb{R}^{d \times |\mathcal{B}_{m-1}|}$ be a matrix whose the $i$-th column vector is the $i$-th element of set $\mathcal{B}_{m-1}$.
2   **if** $|\mathcal{B}_{m-1}| < d$ **then**
3      Use Gaussian elimination to get $M_m \in \mathbb{R}^{(d-|\mathcal{B}_{m-1}|) \times d}$ such that $\mathrm{Span}(\mathcal{B}_{m-1}) = \{x \in \mathbb{R}^d :$
     $M_m x = \vec{0}\}$.
4   **else**
5      Remove constraints in (42), (43) related to $B_{m-1}, M_m$.
6   Query oracle to get empirical best arm $a_m^\star$, the solution of

$$\operatorname*{argmax}_{a \in \mathcal{A}} \left\langle \widehat{\theta}_m, a \right\rangle \tag{42}$$

$$\text{s.t. } M_m a = \vec{0}, \ -\vec{C} \leq \left( B_{m-1}^\top B_{m-1} \right)^{-1} B_{m-1}^\top a \leq \vec{C}.$$

7   **for** $i = 1, \ldots, d$ **do**
8      Set $s_i = \text{LI-Argmax}(\mathcal{A}, A, \widehat{\theta}_m, \mathcal{B}_{m-1}, m, C, a_m^\star)$.
9      **if** $s_i \neq \text{NULL}$ **then**
10        Update $a_i = s_i$ and $\mathcal{I}_m = \mathcal{I}_m - \{i\}$.

11   **for** $i = 1, \ldots, d$ **do**
12      Set $s_i = \text{LI-Argmax}(\mathcal{A}, A, \widehat{\theta}_m, \mathcal{B}_{m-1}, m, C, a_m^\star)$.
13      **if** $s_i \neq \text{NULL}$ **then**
14        **if** $|\det(s_i, A_{-i})| \geq C |\det(A)|$ **or** $i \in \mathcal{I}_m$ **then**
15          Update $\mathcal{I}_m = \mathcal{I}_m - \{i\}$ and $a_i = s_i$.
16          Restart this for-loop with current parameters.

17   **Return:** $\{a_i\}_{i=1}^d - \cup_{i \in \mathcal{I}_m} \frac{e_i}{\sqrt{T_m}}$.

---

# G   Omitted Details of Section 4

## G.1   Notations

We use $e_i \in \mathbb{R}^d$ to denote a vector whose $i$-th coordinate is one and all others are zero. For any two vectors $x, y \in \mathbb{R}^d$, $x \leq y$ indicates that $x_i \leq y_i$ holds for each coordinate $i$. For a positive definite matrix $A \in \mathbb{R}^{d \times d}$, the weighted 2-norm of vector $x \in \mathbb{R}^d$ is given by $\|x\|_A = \sqrt{x^\top A x}$. For matrix $A = [a_1, \cdots, a_d] \in \mathbb{R}^{d \times d}$ with each $a_i \in \mathbb{R}^d$, we use $A_{-i}$ to denote the $(d-1)$-tuple of vectors $[a_1, \cdots, a_{i-1}, a_{i+1}, \cdot, a_d]$. For matrices $A, B$, we use $A \succ B$ to indicate that $A - B$ is positive definite. For two sets $\mathcal{A}, \mathcal{B}$, we use $\mathcal{A} - \mathcal{B}$ to indicate the exclusion. For any scalar $C \in \mathbb{R}$, we use $\vec{C}$ to denote a vector, with all coordinates equal to $C$.

## G.2   Key Technique: Adaptive Barycentric Spanner

**Algorithm description.** Algorithm 12 aims to identify a finite arm set $\mathcal{B}_m$ to linearly represent the (possibly infinite) active arm set $\mathcal{A}_m$, so that playing arms in $\mathcal{B}_m$ allows us to obtain an accurate estimation of $\langle a, \theta \rangle$ for each $a \in \mathcal{A}_m$.

Algorithm 12 initializes a matrix $A = [a_1, \ldots, a_d]$ whose column vectors are linearly independent, and $a_i = e_i / \sqrt{T_m}$. At the beginning, the algorithm solves (42) to find an empirical best arm $a_m^\star$ (line 1-6). Then, in two for-loops (line 7-10 and line 11-16), the algorithm tries to replace each column vector of $A$ with an active arm $a \in \mathcal{A}_m$ by invoking the subroutine LI-Argmax in Algorithm 13, while keeping column vectors of $A$ linearly independent. For each column $i$, LI-Argmax attempts to find $\max_{a \in \mathcal{A}_m} |\det(a, A_{-i})|$[9] by solving (43) with $a_m^\star$. If $\max_{a \in \mathcal{A}_m} |\det(a, A_{-i})| = 0$, then,

---

[9]It is equivalent to $\max_{a \in \mathcal{A}_m} |\langle u, a \rangle|$ where $u$ can be found by rank-one update. See [Zhu et al., 2022] for more details.

---

**Algorithm 13** Linear-independent argmax (LI-Argmax)

---

**Input**: compact arm set $\mathcal{A}$, matrix $A$, estimation $\widehat{\theta}_m$, spanner $\mathcal{B}_{m-1}$, and phase $m$, parameter $C$, and arm $a_m^\star$.

1 Find $u \in \mathbb{R}^d$ s.t. $\langle u, x \rangle = \det(x, A_{-i})$, $\forall x \in \mathbb{R}^d$.

2 Query oracle to obtain $a^+$, the solution of

$$\operatorname*{argmax}_{a \in \mathcal{A}} \langle u, a \rangle$$

$$\text{s.t. } \left\langle \widehat{\theta}_m, a_m^\star - a \right\rangle \le 2^{-m+1},$$

$$M_m a = \vec{0}, \; -\vec{C} \le \left( B_{m-1}^\top B_{m-1} \right)^{-1} B_{m-1}^\top a \le \vec{C}. \tag{43}$$

3 Query oracle to get $a^-$ from (43) by replacing $u$ with $-u$.

4 **if** $\langle u, a^+ \rangle = 0$ and $\langle u, a^- \rangle = 0$ **then**
⌊ **Return:** NULL.

5 **else**
⌊ **Return:** $\operatorname{argmax}_{b \in \{a^+, a^-\}} |\langle u, b \rangle|$.

---

the algorithm returns NULL, since there does not exist an active arm that can maintain the linear independence. In this case, the algorithm keeps $e_i/\sqrt{T_m}$ in the $i$-th column of $A$. Otherwise, $a_i$ will be updated to $s_i \in \operatorname{argmax}_{a \in \mathcal{A}_m} |\det(a, A_{-i})|$. We use set $\mathcal{I}_m$ to record the indices of columns where the replacement fails. According to the conditions in line 14, the second for-loop will be repeatedly restarted to ensure that the linear combination of elements in $\mathcal{B}_m$ can represent all active arms in $\mathcal{A}_m$ with coefficients in the range of $[-C, C]$. Also note that, as shown in Theorem 4.2, LI-Argmax will be oracle-efficient as it only queries the oracle $\widetilde{\mathcal{O}}\left(d^3\right)$ times.

Formally, we have the following result.

**Lemma G.1.** *Suppose $C > 1$ and $\mathcal{B}_m$ is the output of Algorithm 12 in phase $m$. Then, for all $m \in \mathbb{N}$, $\mathcal{B}_m$ is a $C$-approximate barycentric spanner of $\mathcal{A}_m$.*

### G.3 Computational Analysis

Recall from the previous subsection that Algorithm 12 and LI-Argmax needs to find the empirical best arm $a_m^\star$ and $\operatorname{argmax}_{a \in \mathcal{A}_m} |\det(a, A_{-i})|$, both of which rely on the access to the following optimization oracle.

**Definition G.2** (Optimization oracle). *Given a compact set $\mathcal{A} \subseteq \mathbb{R}^d$, the oracle can solve problems of the form*

$$\operatorname*{argmax}_{a \in \mathcal{A}} \langle \theta, a \rangle, \quad \text{s.t. } Ua = \vec{\beta}_1, \quad Va \le \vec{\beta}_2,$$

*for any $\beta_1, \beta_2 \in \mathbb{R}$, $\theta \in \mathbb{R}^d$, $U \in \mathbb{R}^{\tau_1 \times d}$, $V \in \mathbb{R}^{\tau_2 \times d}$, where $\tau_1, \tau_2$ are at most $\mathcal{O}(d)$. If the optimal solution is not unique, the oracle returns any one of them.*

We note that the constrained optimization oracles are commonly-used and also crucial for elimination-type approaches e.g., [Bibaut et al., 2020, Li et al., 2022]. Although this oracle is slightly powerful than that used for non-elimination based approaches [Dani et al., 2008, Agarwal et al., 2014], linear constrained oracle, in fact, can be implemented efficiently in many cases (e.g., a common assumption that $\mathcal{A}$ is a ball [Plevrakis and Hazan, 2020]) via the ellipsoid method. We refer readers to [Bibaut et al., 2020, Plevrakis and Hazan, 2020] for more discussions.

With the optimization oracle in hand, our goal here is to query the oracle to solve the optimization problem $\operatorname{argmax}_{a \in \mathcal{A}_m} |\det(a, A_{-i})|$, which can be rewritten as follows based on Eq. (3).

$$\operatorname*{argmax}_{a \in \mathcal{A}_{m-1}} |\det(a, A_{-i})| \text{s.t.} \left\langle \widehat{\theta}_m, a_m^\star - a \right\rangle \le 2^{-m+1}. \tag{44}$$

Problem (44) also needs to find the empirical best arm $a_m^\star$. A natural idea to compute the empirical best arm $a_m^\star$ is to solve $\operatorname{argmax}_{a \in \mathcal{A}_{m-1}} \langle \widehat{\theta}_m, a \rangle$. Notice that both finding $a_m^\star$ and solving Eq. (44)

require to deal with constraint $a \in \mathcal{A}_{m-1}$. However, this constrain prevents us from querying the optimization oracle due to the following reason.

**Issue:** $a \in \mathcal{A}_{m-1}$ **may cause a large (even infinite) number of constraints.** One can rewrite $a \in \mathcal{A}_{m-1}$ as:

$$\left\{ a \in \mathcal{A} : \left\langle \widehat{\theta}_\tau, a_\tau^\star - a \right\rangle \leq 2^{-\tau+1}, \forall \tau \leq m-1 \right\}. \tag{45}$$

However, the evolution of phases blows up the number of constraints in (45) (e.g., when $m = \Omega(\exp(d))$), which prevents LI-Argmax from directly querying the optimization oracle defined in Definition G.2, since it requires the number of constraints at most $\mathcal{O}(d)$. To address this issue, one needs to remove the dependence on $m$.

**Solution: enlarging active arm set** $\mathcal{A}_{m-1}$. Our solution is to slightly enlarge the active arm set $\mathcal{A}_{m-1}$ in Eq. (45) so that the enlarged set can be expressed by *finitely-many linear constraints*, independent of $m$. Before showing the way to enlarging $\mathcal{A}_{m-1}$, we first give the following definition.

**Definition G.3** (*$C$-bounded spanner*). *For any given set $\mathcal{S} \subseteq \mathbb{R}^d$ and constant $C > 0$, $Span_{[-C,C]}(\mathcal{S})$, defined as follows, is a set that contains all possible linear combinations from $\mathcal{S}$ with coefficients within $[-C, C]$.*

$$Span_{[-C,C]}(\mathcal{S}) = \left\{ \sum_{s \in \mathcal{S}} c_s \cdot s : \forall c_s \in [-C, C] \right\}, \tag{46}$$

*where $c_s$ is the coefficient associated with element $s$.*

We enlarge $\mathcal{A}_{m-1}$ to $\mathcal{A} \cap \text{Span}_{[-C,C]}(\mathcal{B}_{m-1})$, and one can easily verify that this is true since $\mathcal{A}_{m-1} \cap \mathcal{A} = \mathcal{A}_{m-1}$ and Lemma G.1 implies that $\mathcal{A}_{m-1} \subseteq \text{Span}_{[-C,C]}(\mathcal{B}_{m-1})$.

Now, it remains to show that $\text{Span}_{[-C,C]}(\mathcal{B}_{m-1})$ can be expressed by $\mathcal{O}(d)$ number of linear constraints. The following lemma gives the desired result.

**Lemma G.4.** *Suppose that $|\mathcal{B}_{m-1}| < d$. For $M_m$ and $B_{m-1}$ given in Algorithm 12, $Span_{[-C,C]}(\mathcal{B}_{m-1})$ can be equivalently written as*

$$\left\{ a \in \mathbb{R}^d : M_m a = \vec{0}, -\vec{C} \leq \left( B_{m-1}^\top B_{m-1} \right)^{-1} B_{m-1}^\top a \leq \vec{C} \right\}.$$

Hence, LI-Argmax can invoke the optimization oracle in Definition G.2 to solve problems of $\text{argmax}_{a \in \mathcal{A}_{m-1}} \langle \widehat{\theta}_m, a \rangle$ and (44) approximately by solving (42) and (43), respectively.

**Computational efficiency.** We mainly focus on two costly steps in Algorithm 12. The first step is to use Gaussian elimination to obtain the matrix $M_m \in \mathbb{R}^{n \times d}$ for some $n \leq d$. This step can be done with at most $\mathcal{O}(d^3)$ complexity and is implemented at most once for each phase $m$. The other expensive step is to query the optimization oracle to construct a $C$-approximate barycentric spanner. The number of calls to the oracle depends on the number of times that Algorithm 12 invokes LI-Argmax subroutine. The LI-Argmax will be invoked for $d$ times in the first for-loop. For the second for-loop, LI-Argmax will be invoked for $\widetilde{\mathcal{O}}(d^3)$ times. Compared with $\widetilde{\mathcal{O}}(d^2)$ in [Awerbuch and Kleinberg, 2008], the extra $d$ in our complexity comes from the additional restart-condition $i \in \mathcal{I}_m$ in line 14 of Algorithm 12. As $|\mathcal{I}_m| \leq d$ and $\mathcal{I}_m$ is non-increasing when update, this condition will be met for at most $d$ times.

### G.4 Proof of Lemma G.4

**Lemma G.5** (Restatement of Lemma G.4). *Suppose that $|\mathcal{B}_{m-1}| < d$. For $M_m$ and $B_{m-1}$ given in Algorithm 12, the set $Span_{[-C,C]}(\mathcal{B}_{m-1}) = \mathcal{H}_m$ where*

$$\mathcal{H}_m = \left\{ a \in \mathbb{R}^d : M_m a = \vec{0}, -\vec{C} \leq \left( B_{m-1}^\top B_{m-1} \right)^{-1} B_{m-1}^\top a \leq \vec{C} \right\}. \tag{47}$$

*Proof.* For this proof, we show that $a \in \text{Span}_{[-C,C]}(\mathcal{B}_{m-1})$ if and only if $a \in \mathcal{H}_m$.

We first show that if $a \in \text{Span}_{[-C,C]}(\mathcal{B}_{m-1})$, then $a \in \mathcal{H}_m$. As matrix $M_m \in \mathbb{R}^{(d-|\mathcal{B}_{m-1}|) \times d}$ is defined by $\text{Span}(\mathcal{B}_{m-1}) = \{x \in \mathbb{R}^d : M_m x = \vec{0}\}$, $a \in \text{Span}_{[-C,C]}(\mathcal{B}_{m-1}) \subseteq \text{Span}(\mathcal{B}_{m-1})$ gives

that $M_m a = \vec{0}$. Recall that $B_{m-1} \in \mathbb{R}^{d \times |\mathcal{B}_{m-1}|}$ is a matrix whose the $i$-th column is the $i$-th element of set $\mathcal{B}_{m-1}$ and $\text{rank}(B_{m-1}) = |\mathcal{B}_{m-1}|$ because $\mathcal{B}_{m-1}$ is linearly independent. Therefore, for each $a \in \text{Span}_{[-C,C]}(\mathcal{B}_{m-1})$, there exists a unique vector[10] $x_a \in \mathbb{R}^{|\mathcal{B}_{m-1}|}$ such that $B_{m-1} x_a = a$ and the value of each coordinate of $x_a$ is no larger than $C$. As $\mathcal{B}_{m-1}$ is linearly independent, $B_{m-1}$ is a full-column rank matrix, which gives that $x_a = \left(B_{m-1}^\top B_{m-1}\right)^{-1} B_{m-1}^\top a$. As a result, if $a \in \text{Span}_{[-C,C]}(\mathcal{B}_{m-1})$, then, $-\vec{C} \le \left(B_{m-1}^\top B_{m-1}\right)^{-1} B_{m-1}^\top a \le \vec{C}$, which concludes the proof of this argument.

Then, we show that if $a \in \mathcal{H}_m$, then, $a \in \text{Span}_{[-C,C]}(\mathcal{B}_{m-1})$. If $M_m a = \vec{0}$, then, we have $a \in \text{Span}(\mathcal{B}_{m-1})$ due to the definition of $M_m$. As $\text{rank}(B_{m-1}) = |\mathcal{B}_{m-1}|$, for each $a \in \text{Span}(\mathcal{B}_{m-1})$, there exists a unique vector $x_a \in \mathbb{R}^{|\mathcal{B}_{m-1}|}$ such that $B_{m-1} x_a = a$, which gives that $x_a = \left(B_{m-1}^\top B_{m-1}\right)^{-1} B_{m-1}^\top a$. Thus, $-\vec{C} \le \left(B_{m-1}^\top B_{m-1}\right)^{-1} B_{m-1}^\top a \le \vec{C}$ requires each coordinate of $x_a$ no larger than $C$. As a consequence, if $a \in \mathcal{H}_m$, then, $a \in \text{Span}_{[-C,C]}(\mathcal{B}_{m-1})$.

Combining the above analysis, we complete the proof. $\qquad\square$

### G.5 Proof of Lemma G.1

In this section, we aim to show that $\mathcal{B}_m$ is a $C$-approximate barycentric spanner of $\mathcal{A}_m$. To this end, we first show that $\mathcal{B}_m$ is a $C$-approximate barycentric spanner of $\mathcal{S}_m$ which is defined below and $\mathcal{A}_m \subseteq \mathcal{S}_m$ results in the desired claim. In fact, throughout our analysis, we stick with $\mathcal{S}_m$ rather than $\mathcal{A}_m$.

$$\mathcal{S}_1 = \mathcal{A}, \quad \mathcal{S}_m = \left\{a \in \mathcal{A} : \left\langle \widehat{\theta}_m, a_m^\star - a \right\rangle \le 2^{-m+1}, \ a \in \text{Span}_{[-C,C]}(\mathcal{B}_{m-1})\right\}, \ \forall m \ge 2. \quad (48)$$

Moreover, according to Lemma G.4, for all $m \ge 2$, $\mathcal{S}_m$ can also be equivalently rewritten as:

$$\mathcal{S}_m = \left\{a \in \mathcal{A} : \left\langle \widehat{\theta}_m, a_m^\star - a \right\rangle \le 2^{-m+1}, \ M_m a = \vec{0}, \ -\vec{C} \le \left(B_{m-1}^\top B_{m-1}\right)^{-1} B_{m-1}^\top a \le \vec{C}\right\}$$

**Lemma G.6.** *Let $C > 1$ and $\mathcal{B}_m$ is in Algorithm 2. For all $m \in \mathbb{N}$, $\mathcal{B}_m$ is a $C$-approximate barycentric spanner of $\mathcal{S}_m$.*

*Proof.* For all $m \in \mathbb{N}$, we define $\mathcal{B}'_m = \mathcal{B}_m \cup \left(\cup_{i \in \mathcal{I}_m} \frac{e_i}{\sqrt{T_m}}\right)$. We further define set $\mathcal{S}'_m = \mathcal{S}_m \cup \left(\cup_{i \in \mathcal{I}_m} \frac{e_i}{\sqrt{T_m}}\right)$, and matrix $B'_m = [b_1, \cdots, b_d]$ where $b_i$ is the $i$-th element of $\mathcal{B}'_m$. By these definitions, $b_i = \frac{e_i}{\sqrt{T_m}}$ for all $i \in \mathcal{I}_m$. The proof consists of three steps, and we first proceed step 1.

**Step 1: each solution of** (43) **is in $\mathcal{S}_m$.** According to the optimization problem (43), for each phase $m$, the solution is drawn from (see Eq. (47) for definition of $\mathcal{H}_m$)

$$\mathcal{H}_m \cap \left\{a \in \mathcal{A} : \left\langle \widehat{\theta}_m, a_m^\star - a \right\rangle \le 2^{-m+1}\right\}$$
$$= \text{Span}_{[-C,C]}(\mathcal{B}_{m-1}) \cap \left\{a \in \mathcal{A} : \left\langle \widehat{\theta}_m, a_m^\star - a \right\rangle \le 2^{-m+1}\right\}$$
$$= \mathcal{S}_m,$$

where the first equality holds due to Lemma G.4 and the last equality holds due to Eq. (48).

**Step 2: $\mathcal{B}'_m$ is a $C$-approximate barycentric spanner of $\mathcal{S}'_m$.** The proof idea of this step follows a similar arguments of Proposition 2.2 and Proposition 2.5 in [Awerbuch and Kleinberg, 2008]. In Algorithm 12, each update for matrix $A \in \mathbb{R}^{d \times d}$ always maintains $|\det(A)| > 0$, and thus $\mathcal{B}'_m$ spans $\mathbb{R}^d$. As $\mathcal{S}'_m \subseteq \mathbb{R}^d$, every element $a \in \mathcal{S}'_m$ can be linearly represented by $a = \sum_{i=1}^d w_i b_i$ for some coefficients $\{w_i\}_{i=1}^d$, and then we have

$$\left|\det\left(a, (B'_m)_{-i}\right)\right| = \left|\det\left(\sum_{i=1}^d w_i b_i, (B'_m)_{-i}\right)\right| = \left|\sum_{i=1}^d w_i \det\left(b_i, (B'_m)_{-i}\right)\right| = |w_i| |\det(B'_m)|.$$

---

[10]We have the uniqueness because one can treat $B_{m-1} x_a = a$ as an overdetermined linear system in terms of $x_a \in \mathbb{R}^{|\mathcal{B}_{m-1}|}$ and we have $\text{rank}(B_{m-1}) = |\mathcal{B}_{m-1}|$.

Step 1 shows that each solution of (43) is in $\mathcal{S}_m$, and thus $\mathcal{B}_m \subseteq \mathcal{S}_m$, which further implies $\mathcal{B}'_m \subseteq \mathcal{S}'_m$. Recall that before the termination, the algorithm will scan each column vector of matrix $A$ to ensure that no more replacement can be made. Hence, once Algorithm 12 terminates, we have that $\forall i \in [d]$, $\sup_{a \in \mathcal{S}'_m} |\det(a, (B'_m)_{-i})| \leq C |\det(B'_m)|$, and then $\forall i \in [d], |w_i| \leq C$, which implies that $\mathcal{B}'_m$ is a $C$-approximate barycentric spanner of $\mathcal{S}'_m$.

**Step 3: $\mathcal{B}_m$ is a $C$-approximate barycentric spanner of $\mathcal{S}_m$.** Here, we show that every $a \in \mathcal{S}_m$ can be written as a linear combination of elements *only* in $\mathcal{B}_m$ with coefficients in $[-C, C]$. Notice that we only need to consider $a \in \mathcal{S}_m - \mathcal{B}_m$ since if $a \in \mathcal{B}_m$, then, one can find a trivial linear combination by itself (recall that we assume $C > 1$). Then, we prove the desired claim for all $a \in \mathcal{S}_m - \mathcal{B}_m$ by contradiction. Suppose that there exists an arm $a \in \mathcal{S}_m - \mathcal{B}_m$ such that it cannot be linearly represented only by elements in $\mathcal{B}_m$ with coefficients in $[-C, C]$. As $\mathcal{S}_m \subseteq \mathcal{S}'_m$ and $\mathcal{B}'_m$ is a $C$-approximate barycentric spanner of $\mathcal{S}'_m$ (shown in step 2), there must exist coefficients $\{d_i\}_{i=1}^d$ to linearly represent $a$ as:

$$a = \sum_{i=1}^d d_i b_i, \quad \forall i \in [d], \ d_i \in [-C, C], \quad \text{and} \quad \exists i \in \mathcal{I}_m \text{ such that } d_i \neq 0. \tag{49}$$

As for Eq. (49), we assume without loss of generality that the coefficient index $j \in \mathcal{I}_m$ satisfies $d_j \neq 0$. Since $a \notin \mathcal{B}_m$, the second for-loop in Algorithm 12 ensures that $a$ can be represented as a linear combination of $\{b_i\}_{i \in [d]: i \neq j}$, and it would be put in $\mathcal{B}_m$ otherwise (recall that Algorithm 12 terminates the second for-loop, when it scans every column vector of $A$ and makes no replacement). Hence, there exist coefficients $\{c_i\}_{i \in [d]: i \neq j}$ such that

$$a = \sum_{i \in [d]: i \neq j} c_i b_i, \quad \forall i \neq j, \ c_i \in \mathbb{R}, . \tag{50}$$

Bridging Eq. (49) and Eq. (50), we have

$$d_j b_j = \sum_{i \neq j} (c_i - d_i) b_i \ \longrightarrow \ b_j = \frac{1}{d_j} \sum_{i \neq j} (c_i - d_i) b_i,$$

which implies that the $j$-th element of $\mathcal{B}'_m$, i.e., $b_j$ can be expressed as a linear combination of other elements of $\mathcal{B}'_m$, i.e., $\{b_i\}_{i \neq j}$. This leads to a contradiction because Algorithm 12 ensures $\mathcal{B}'_m$ to be linearly independent. As a consequence, every element of $a \in \mathcal{S}_m$ can be written as a linear combination of elements only in $\mathcal{B}_m$ with coefficients in $[-C, C]$, which completes the proof. $\square$

*Proof of Lemma G.1.* Recall from Eq. (3) that $\mathcal{A}_m$ is defined as

$$\mathcal{A}_1 = \mathcal{A}, \quad \mathcal{A}_m = \left\{ a \in \mathcal{A}_{m-1} : \left\langle \widehat{\theta}_m, a_m^\star - a \right\rangle \leq 2^{-m+1} \right\}, \quad \forall m \geq 2.$$

Since Lemma G.6 gives that $\mathcal{B}_m$ is a $C$-approximate barycentric spanner of $\mathcal{S}_m$, it suffices to show that $\mathcal{A}_m \subseteq \mathcal{S}_m$ to prove the claimed result. We prove this by induction. For the base case $m = 1$, $\mathcal{A}_1 \subseteq \mathcal{S}_1$ trivially holds based on definitions. Suppose that $\mathcal{A}_m \subseteq \mathcal{S}_m$ holds for $m \geq 2$. For $m + 1$, $\mathcal{A}_{m+1}$ is defined as:

$$\mathcal{A}_{m+1} = \left\{ a \in \mathcal{A}_m : \left\langle \widehat{\theta}_{m+1}, a_{m+1}^\star - a \right\rangle \leq 2^{-m} \right\}.$$

Since Lemma G.6, gives that $\mathcal{B}_m$ is a $C$-approximate barycentric spanner of $\mathcal{S}_m$, and the inductive hypothesis gives $\mathcal{A}_m \subseteq \mathcal{S}_m$, we have $\mathcal{A}_m \subseteq \text{Span}_{[-C,C]}(\mathcal{B}_m)$, which implies that

$$\begin{aligned} \mathcal{A}_{m+1} &= \left\{ a \in \mathcal{A}_m : \left\langle \widehat{\theta}_{m+1}, a_{m+1}^\star - a \right\rangle \leq 2^{-m} \right\} \\ &= \left\{ a \in \mathcal{A} : \left\langle \widehat{\theta}_{m+1}, a_{m+1}^\star - a \right\rangle \leq 2^{-m}, a \in \mathcal{A}_m \right\} \\ &\subseteq \left\{ a \in \mathcal{A} : \left\langle \widehat{\theta}_{m+1}, a_{m+1}^\star - a \right\rangle \leq 2^{-m}, \ a \in \text{Span}_{[-C,C]}(\mathcal{B}_m) \right\} \\ &= \mathcal{S}_{m+1}. \end{aligned}$$

Once the induction is done, the proof is complete. $\square$

## G.6  Proof of Theorem 4.2: Regret Analysis

Before proving Theorem 4.2, we first present technical concept and lemmas. We first formally introduce the concept of $C$-approximate optimal design, both of which help us to prove a stronger result of Eq. (4), paving the way to prove the main theorem. Let $\Delta(\mathcal{A})$ be the set of all Radon probability measures over set $\mathcal{A}$.

**Definition G.7** ($C$-approximate optimal design). *Suppose that $\mathcal{A} \subseteq \mathbb{R}^d$ is a finite and compact set. A distribution $\pi \in \Delta(\mathcal{A})$ is called a $C$-approximate optimal design with an approximation factor $C \geq 1$, if*

$$\sup_{a \in \mathcal{A}} \|a\|_{V(\pi;\mathcal{A})^{-1}}^2 \leq C \cdot d, \quad where \quad V(\pi;\mathcal{A}) = \sum_{a \in \mathcal{A}} \pi(a) aa^\top.$$

Then, the following lemma shows that if one computes a uniform design for set $\mathcal{B}$, a barycentric spanner for another set $\mathcal{A}$, then, playing arms in $\mathcal{B}$ can guarantee accurate estimations over $\mathcal{A}$.

**Lemma G.8** (Lemma 2 of [Zhu et al., 2022]). *Suppose that $\mathcal{A} \subseteq \mathbb{R}^d$ is a compact set that spans $\mathbb{R}^d$. If $\mathcal{B} = [b_1, \cdots, b_d]$ is a $C$-approximate barycentric spanner for $\mathcal{A}$, then, $\pi : \mathcal{B} \to \frac{1}{d}$ is a $(C^2 \cdot d)$-approximate optimal design, which guarantees*

$$\sup_{a \in \mathcal{A}} \|a\|_{V(\pi;\mathcal{B})^{-1}}^2 \leq C^2 \cdot d^2, \quad where \quad V(\pi;\mathcal{B}) = \sum_{b \in \mathcal{B}} \pi(b) bb^\top.$$

We then present the following lemma which provides a stronger result than that of Eq. (4). More specifically, we are supposed to show that $\|a\|_{V_m^{-1}} \leq C \cdot d / \sqrt{T_m}$ holds for all $a \in \mathcal{A}_m$ in Eq. (4), but we will show that this holds for all $a \in \mathcal{S}_m$. This is stronger since $\mathcal{A}_m \subseteq \mathcal{S}_m$ for all $m \in \mathbb{N}$.

**Lemma G.9.** *For $\mathcal{S}_m$ defined in Eq. (48) and $\mathcal{B}_m$ returned by Algorithm 12, setting $\pi_m(a) = \frac{1}{d}$ for each $a \in \mathcal{B}_m$ and playing each arm $a \in \mathcal{B}_m$ for $n_m(a) = \lceil T_m \pi_m(a) \rceil$ times ensure that*

$$\forall a \in \mathcal{S}_m, \quad \|a\|_{V_m^{-1}} \leq \frac{C \cdot d}{\sqrt{T_m}}. \tag{51}$$

*Proof.* For all $m \in \mathbb{N}$, we define $\mathcal{B}'_m = \mathcal{B}_m \cup \left( \cup_{i \in \mathcal{I}_m} \frac{e_i}{\sqrt{T_m}} \right)$ and

$$V(\pi_m; \mathcal{B}'_m) = \sum_{a \in \mathcal{B}'_m} \pi_m(a) aa^\top = \sum_{a \in \mathcal{B}_m} \pi_m(a) aa^\top + \sum_{a \in \mathcal{B}'_m - \mathcal{B}_m} \pi_m(a) aa^\top.$$

With this definition and also the definition $n_m(a) = \lceil T_m \pi_m(a) \rceil$, we have

$$
\begin{aligned}
V_m &= I + \sum_{a \in \mathcal{B}_m} n_m(a) aa^\top \\
&\succeq I + T_m \sum_{a \in \mathcal{B}_m} \pi_m(a) aa^\top \\
&\succ T_m V(\pi_m; \mathcal{B}'_m),
\end{aligned}
\tag{52}
$$

where the last step follows from the fact that

$$I = T_m \sum_{i=1}^d \frac{e_i}{\sqrt{T_m}} \frac{e_i^\top}{\sqrt{T_m}} \succ T_m \sum_{i \in \mathcal{I}_m} \pi_m \left( \frac{e_i}{\sqrt{T_m}} \right) \frac{e_i}{\sqrt{T_m}} \frac{e_i^\top}{\sqrt{T_m}} = T_m \sum_{a \in \mathcal{B}'_m - \mathcal{B}_m} \pi_m(a) aa^\top.$$

Since $\mathcal{B}'_m$ spans $\mathbb{R}^d$ and $\mathcal{B}'_m \subseteq \text{Span}_{[-C,C]}(\mathcal{B}'_m)$, set $\text{Span}_{[-C,C]}(\mathcal{B}'_m)$ also spans $\mathbb{R}^d$. Lemma G.8 gives that

$$\forall a \in \text{Span}_{[-C,C]}(\mathcal{B}'_m), \quad \|a\|_{(V(\pi_m; \mathcal{B}'_m))^{-1}} \leq C \cdot d.$$

Thus, one can use Eq. (52) to show that for all

$$\forall a \in \text{Span}_{[-C,C]}(\mathcal{B}'_m), \quad \|a\|_{V_m^{-1}} \leq \frac{\|a\|_{(V(\pi_m; \mathcal{B}'_m))^{-1}}}{\sqrt{T_m}} \leq \frac{C \cdot d}{\sqrt{T_m}}.$$

As $\mathcal{B}_m$ is a $C$-approximate barycentric spanner of $\mathcal{S}_m$, we have $\mathcal{S}_m \subseteq \text{Span}_{[-C,C]}(\mathcal{B}_m) \subseteq \text{Span}_{[-C,C]}(\mathcal{B}'_m)$, the proof is thus complete. $\qquad \square$

The following analysis will condition on the nice event $\mathcal{E}$, defined as:

$$\mathcal{E} = \left\{ \forall m \in \mathbb{N} : \left| \left\langle a, \widehat{\theta}_{m+1} - \theta \right\rangle \right| \leq 2^{-m-1} \text{ for all } a \in \mathcal{S}_m \right\},\tag{53}$$

where $\mathcal{S}_m$ is defined in Eq. (48).

**Lemma G.10.** *We have $\mathbb{P}(\mathcal{E}) \geq 1 - \delta$.*

*Proof.* One can show that

$$\forall a \in \mathcal{S}_m, \quad \left| \left\langle a, \widehat{\theta}_{m+1} - \theta \right\rangle \right| \leq \|a\|_{V_m^{-1}} \left\| \widehat{\theta}_{m+1} - \theta \right\|_{V_m} \leq \frac{C \cdot d}{\sqrt{T_m}} \left\| \widehat{\theta}_{m+1} - \theta \right\|_{V_m},\tag{54}$$

where the first inequality uses Cauchy-Schwartz inequality and the second inequality uses Lemma G.9.

Notice that the above inequality holds for all $m \in \mathbb{N}$. Finally, Theorem 20.5 of [Lattimore and Szepesvári, 2020] shows that with probability at least $1 - \delta$, for all $m \in \mathbb{N}$

$$\left\| \widehat{\theta}_{m+1} - \theta \right\|_{V_m} \leq 2\sqrt{\log(1/\delta) + d \log(T_m)}.\tag{55}$$

Combining Eq. (55) and Eq. (54), we have that $\forall a \in \mathcal{S}_m$,

$$\begin{aligned}
&\left| \left\langle a, \widehat{\theta}_{m+1} - \theta \right\rangle \right| \\
&\leq 2Cd\sqrt{\frac{\log(1/\delta) + d \log(T_m)}{T_m}} \\
&\leq \frac{1}{8C} \cdot 2^{-m} \sqrt{\frac{\log(1/\delta) + \log\left(256C^4 \cdot \frac{d^3}{4^{-m}} \log\left(\frac{d^3 4^m}{\delta}\right)\right)}{\log\left(\frac{d^3 4^m}{\delta}\right)}} \\
&= \frac{1}{8C} \cdot 2^{-m} \sqrt{\frac{4\log(4C) + \log\left(\frac{d^3 4^m}{\delta} \log\left(\frac{d^3 4^m}{\delta}\right)\right)}{\log\left(\frac{d^3 4^m}{\delta}\right)}} \\
&\leq \frac{1}{8C} \cdot 2^{-m} \sqrt{\frac{8C + 2\log\left(\frac{d^3 4^m}{\delta}\right)}{\log\left(\frac{d^3 4^m}{\delta}\right)}} \\
&\leq \frac{1}{8C} \cdot 2^{-m} \sqrt{\frac{16C \log\left(\frac{d^3 4^m}{\delta}\right)}{\log\left(\frac{d^3 4^m}{\delta}\right)}} \\
&= 2^{-m-1},
\end{aligned}$$

where the third inequality uses $4\log(4x) \leq 8x$ for all $x \in \mathbb{R}_{>0}$, the fourth inequality follows from $a + b \leq 2ab$ whenever $a, b > 1$ (here $a = 8C$ and $b = 2\log(d^3 4^m/\delta)$), and the last inequality holds due to $C > 1$.

Hence, the proof is complete. $\qquad \square$

**Lemma G.11.** *Suppose $\mathcal{E}$ occurs where $\mathcal{E}$ is in Eq. (53). For all $m \in \mathbb{N}$, $a^\star \in \mathcal{S}_m$ holds where $\mathcal{S}_m$ is defined in Eq. (48).*

*Proof.* We prove this by induction. For the base case $m = 1$, the claim holds trivially. Suppose that it holds for phase $m$ (i.e., $a^\star \in \mathcal{S}_m$) and then we aim to show $a^\star \in \mathcal{S}_{m+1}$. For all $a \in \mathcal{S}_m$, one can show

$$0 \leq \langle \theta, a^\star \rangle - \langle \theta, a \rangle \leq \left\langle \widehat{\theta}_{m+1}, a^\star \right\rangle - \left\langle \widehat{\theta}_{m+1}, a \right\rangle + 2 \times 2^{-m-1} = \left\langle \widehat{\theta}_{m+1}, a^\star - a \right\rangle + 2^{-m},$$

where the second inequality follows from the definition of $\mathcal{E}$ (see Eq. (53)) and $a^\star \in \mathcal{S}_m$ by inductive hypothesis. Then, we have

$$\left\langle \widehat{\theta}_{m+1}, a^\star - a^\star_{m+1} \right\rangle + 2^{-m} \geq 0.$$

The inductive hypothesis gives $a^\star \in \mathcal{S}_m$, and thus

$$a^\star \in \text{Span}_{[-C,C]}(\mathcal{B}_m) \cap \mathcal{A},\tag{56}$$

since Lemma G.6 gives that $\mathcal{B}_m$ is a $C$-approximate barycentric spanner of $\mathcal{S}_m$. Combining $\left\langle \widehat{\theta}_{m+1}, a^\star - a^\star_{m+1} \right\rangle + 2^{-m} \geq 0$ and Eq. (56), we have $a^\star \in \mathcal{S}_{m+1}$ according to definition of $\mathcal{S}_m$ in Eq. (48). Once the induction is done, the proof is complete. $\qquad\square$

**Lemma G.12.** *Suppose that $\mathcal{E}$ occurs where $\mathcal{E}$ is in Eq. (53). For each arm $a \in \mathcal{A}$ with $\Delta_a > 0$, it will not be in $\mathcal{S}_m$ for all phases $m \geq m_a + 1$, where $m_a$ is the smallest phase such that $\frac{\Delta_a}{2} > 2^{-m_a}$.*

*Proof.* Consider an arbitrary arm $a \in \mathcal{A}$ with $\Delta_a > 0$. Let $m_a$ be the smallest phase such that $\frac{\Delta_a}{2} > 2^{-m_a}$ (i.e., $\frac{\Delta_a}{2} \leq 2^{-(m_a-1)}$). Then, we will show that arm $a$ will be not in $\mathcal{S}_\tau$ for all $\tau \geq m_a$. Suppose that $a \in \mathcal{S}_{m_a}$ (if not, it does not impact the claim). One can show that

$$
\begin{aligned}
&\left\langle \widehat{\theta}_{m_a+1}, a^\star_{m_a+1} - a \right\rangle - 2^{-m_a} \\
&= \sup_{b \in \mathcal{S}_{m_a}} \left\langle \widehat{\theta}_{m_a+1}, b - a \right\rangle - 2^{-m_a} \\
&\geq \left\langle \widehat{\theta}_{m_a+1}, a^\star - a \right\rangle - 2^{-m_a} \\
&\geq \left\langle \theta, a^\star - a \right\rangle - 2^{-m_a+1} \\
&> \Delta_a - 2 \times \frac{\Delta_a}{2} \\
&= 0,
\end{aligned}
$$

where the first inequality uses Lemma G.11 that $a^\star \in \mathcal{S}_m$ for all $m \in \mathbb{N}$, the second inequality follows from the definition of $\mathcal{E}$ (see Eq. (53)), and the last inequality holds due to the choice of $m_a$.

According to the definition of $\mathcal{S}_m$ in Eq. (48), arm $a$ will not be in $\mathcal{S}_m$ for all $m \geq m_a + 1$ as long as $\mathcal{E}$ occurs. $\qquad\square$

**Lemma G.13.** *Let $m(t)$ be the phase in which round $t$ lies. Then, $m(t) \leq \log_2(t+1)$ for all $t \in \mathbb{N}$.*

*Proof.* We prove this by contradiction. Suppose that $\exists t \in \mathbb{N}$ that $m(t) > \log_2(t+1)$. Note that we can further assume $m(t) \geq 2$ since one can easily verify that for all $t$ such that $m(t) = 1$, $m(t) \leq \log_2(t+1)$ must hold. Recall that in phase $m(t)$, each active arm will be played for $m_{\ell(t)}$ times, we have

$$
t \geq \sum_{a \in \mathcal{B}_{m(t)-1}} \left\lceil \pi_{m(t)-1}(a) T_{m(t)-1} \right\rceil
$$

$$
\geq \frac{T_{m(t)-1}}{d} = 256 C^4 \cdot \frac{d^2}{4^{-(m(t)-1)}} \log\left( \delta^{-1} d^3 4^{m(t)-1} \right) \geq 64 C^4 \cdot d^2 (t+1)^2 \log\left( 4\delta^{-1} d^3 \right) > t,
$$

where the third inequality bounds $\ell(t)$ in the logarithmic term by $\ell(t) \geq 2$ and bound the other $\ell(t) > \log_2(t+1)$ by assumption. Therefore, once a contradiction occurs, the proof is complete. $\qquad\square$

**Lemma G.14.** *Let $m(t)$ be the phase in which round $t$ lies. Suppose that $\mathcal{E}$ occurs where $\mathcal{E}$ is in Eq. (53) and Algorithm 2 computes a $C$-approximate barycentric spanner with $C > 1$. For all $t \in \mathbb{N}$ and all $a \in \mathcal{A}$, if $a \in \mathcal{S}_{m(t)}$, then,*

$$
\Delta_a \leq \sqrt{\frac{64 \times 512 C^4 d^3 \log\left( \frac{d^3 4(t+1)^2}{\delta} \right)}{3t}}.
$$

*Proof.* If $a \in \mathcal{S}_{m(t)}$ is optimal, then, $\Delta_a = 0$ and the claim trivially holds. In what follows, we only consider arm $a \in \mathcal{S}_{m(t)}$ with $\Delta_a > 0$. Consider an arbitrary round $t \in \mathbb{N}$ and an arbitrary arm $a \in \mathcal{S}_{m(t)}$. Then, $t$ can be bounded by

$$
t \leq \sum_{s=1}^{m(t)} \sum_{a \in \mathcal{B}_s} \left\lceil \pi_s(a) T_s \right\rceil
$$

$$\leq 2 \sum_{s=1}^{m(t)} \sum_{a \in \mathcal{B}_s} \pi_s(a) T_s$$

$$= 512 C^4 \sum_{s=1}^{m(t)} \frac{d^3}{4^{-s}} \log\left(\frac{d^3 4^s}{\delta}\right)$$

$$\leq 512 C^4 d^3 \log\left(\frac{d^3 4^{m(t)}}{\delta}\right) \sum_{s=1}^{m(t)} \frac{1}{4^{-s}}$$

$$\overset{(a)}{\leq} 512 C^4 d^3 \log\left(\frac{d^3 4^{m(t)}}{\delta}\right) \sum_{s=1}^{m_a} \frac{1}{4^{-s}}$$

$$\overset{(b)}{\leq} \frac{64 \times 512 C^4 d^3 \log\left(\frac{d^3 4^{m(t)}}{\delta}\right)}{3\Delta_a^2}$$

$$\overset{(c)}{\leq} \frac{64 \times 512 C^4 d^3 \log\left(\frac{d^3 4^{\log_2(t+1)}}{\delta}\right)}{3\Delta_a^2}$$

$$\leq \frac{64 \times 512 C^4 d^3 \log\left(\frac{d^3 4(t+1)^2}{\delta}\right)}{3\Delta_a^2},$$

where the second inequality holds because $\pi_s(a) = \frac{1}{d}$ for all $s \in \mathbb{N}$ and all $a \in \mathcal{B}_s$, thereby $\pi_s(a) T_s \geq 1$, which gives $\lceil \pi_s(a) T_s \rceil \leq 2\pi_s(a) T_s$, the inequality (a) bounds $m(t) \leq m_a$ where $m_a$ is defined in Lemma G.12, the inequality (b) uses $\frac{\Delta_a}{2} \leq 2^{-m_a+1}$ to bound $m_a \leq \log_2(4/\Delta_a)$, and the inequality (c) follows from Lemma G.13 that $m(t) \leq \log_2(t+1)$.

Conditioning on $\mathcal{E}$, this argument holds for each $t, a$, which completes the proof. $\square$

*Proof of Theorem 4.2.* Once Lemma G.11 and Lemma G.14 hold, Theorem 3.1 gives that for any fixed $\delta \in (0,1)$, Algorithm 10 achieves the ULI guarantee with a function (omitting $C$ as it is a constant)

$$F_{\mathrm{ULI}}(\delta, t) = \mathcal{O}\left(\sqrt{\frac{d^3 \log(dt/\delta)}{t}}\right).$$

Therefore, the proof is complete. $\square$

## G.7 Proof of Theorem 4.2: Computational Analysis

As the second for-loop restarts repeatedly, and we first present the following lemma to bound the number of times that it restarts.

**Lemma G.15.** *Under the same setting of Lemma G.1, Algorithm 12 outputs a $C$-barycentric spanner by restarting the second for-loop for $\mathcal{O}\left(d^2 \log_C(d)\right)$ times.*

*Proof.* According to [Awerbuch and Kleinberg, 2008, Lemma 2.6], if $\mathcal{I}_m$ never changes after entering the second for-loop, then, the second for-loop restarts for at most $\mathcal{O}(d\log_C(d))$ times, and then the algorithm terminates. In Algorithm 12, if $\mathcal{I}_m$ changes, the second for-loop restarts. As set $\mathcal{I}_m$ is always non-increasing, it suffices to consider the worst case ($\mathcal{I}_m$ changes at most $O(d)$ times). Hence, the second for-loop restarts at most $O(d^2 \log_C(d))$ times. $\square$

Now, we are ready to show the computational complexity.

From Lemma G.15, Algorithm 12 restarts the second for-loop at most $O(d^2 \log_C(d))$ times. For each run of the second for-loop, the optimization oracle is invoked at most $d$ times. Thus, the number of calls to the oracle is at most $\tilde{O}(d^3 \log_C(d))$. Apart from the second for-loop, the first for-loop invokes the oracle $d$ times and computing the empirical best arm $a_m^\star$ requires once call to the oracle. Combining all together, we obtain the claimed bound.

# H   Omitted Details of Section 5

## H.1   Proof of Theorem 5.1

The proof of main theorem conditions on a nice event $\mathcal{E}$ in which some high-probability bounds hold simultaneously. We defer the formal definition of $\mathcal{E}$ to Appendix H.2.

The key to conducting policy elimination is to ensure that the estimated value functions will be closer to the true value functions as phases evolve. The following proposition gives us the desired result.

**Proposition H.1.** *Suppose that $\mathcal{E}$ occurs. For all $m \in \mathbb{N}$ and $\pi \in \Pi_m$, we have*

$$\left| \widetilde{V}_m^\pi - \mathbb{E}_{s_1 \sim \mu} \left[ V_1^\pi(s_1) \right] \right| \leq 2^{m-1}.$$

With the above result at hand, one can treat each policy as an arm and repeat the same arguments in Appendix D.2 (counterparts are Lemma D.7, Lemma D.8, and Lemma D.9) to get the following three lemmas.

**Lemma H.2.** *Suppose that $\mathcal{E}$ occurs. For each $m \in \mathbb{N}$, $\pi^\star \in \Pi_m$ holds.*

**Lemma H.3.** *Suppose that $\mathcal{E}$ occurs. For each policy $\pi$ with $\Delta_\pi > 0$, it will not be in $\Pi_m$ for all phases $m \geq m_\pi + 1$ where $m_\pi$ is the smallest phase such that $\frac{\Delta_\pi}{2} > 2^{-m_\pi}$.*

**Lemma H.4.** *Let $m(t)$ be the phase that round $t$ lies in. Then, $m(t) \leq \log_2(t+1)$ for all $t \in \mathbb{N}$.*

The next lemma shows that if a policy is not eliminated, then, the policy gap is in order of $\widetilde{\mathcal{O}}(t^{-1/2})$.

**Lemma H.5.** *Let $m(t)$ be the phase in which episode/round $t$ lies. Suppose that $\mathcal{E}$ occurs. For all $t \in \mathbb{N}$ and all $\pi \in \Pi$, if $\pi \in \Pi_{m(t)}$, then*

$$\Delta_\pi \leq \sqrt{\frac{S^3 A H^5 \log^2(tSAH/\delta)}{t}}.$$

*Proof.* If $\pi \in \Pi_{m(t)}$ is optimal, then, $\Delta_\pi = 0$ and the claim trivially holds. In what follows, we only consider policy $\pi \in \Pi_{m(t)}$ with $\Delta_\pi > 0$. From Lemma H.3, if a policy $\pi \in \Pi_{m(t)}$ is with $\Delta_\pi > 0$, then, $m(t) \leq m_\pi$ where $m_\pi$ is defined in Lemma H.3. Thus, the total number of episodes/rounds that such a policy $\pi$ is active is at most

$$
\begin{aligned}
t &\leq \sum_{s=1}^{m(t)} T_s \\
&\leq 2c_1 S^2 A H^4 \sum_{s=1}^{m(t)} 2^{2s} \log^2\left(2c_2 s^2 2^{2s} S^2 A H^4 |\Pi_{\texttt{all}}|/\delta\right) \\
&\leq 2c_1 S^2 A H^4 \log^2\left(2c_2 \log^2 t (t+1)^2 S^2 A H^4 |\Pi_{\texttt{all}}|/\delta\right) \sum_{s=1}^{m(t)} 2^{2s} \\
&\leq \mathcal{O}\left(\frac{S^2 A H^4 \log^2(tSAH|\Pi_{\texttt{all}}|/\delta)}{\Delta_\pi^2}\right) \\
&= \mathcal{O}\left(\frac{S^3 A H^5 \log^2(tSAH/\delta)}{\Delta_\pi^2}\right),
\end{aligned}
$$

where the second inequality holds due to $s \leq m(t) \leq \log_2(t+1)$, and the last step uses the fact that $\Pi_{\texttt{all}} = A^{SH}$. Rearranging the above, we complete the proof. $\qquad\square$

Now, Theorem 5.1 is immediate if we treat each policy as an arm and invokes Theorem 3.1.

## H.2 Construction of Nice Event

We extend the definitions of value function and action value functions to reward-dependent ones.

$$Q_h^\pi(s, a, r) = \mathbb{E}\left[\sum_{h'=h}^{H} r_{h'}(s_{h'}, a_{h'}) \mid s_{h'} = s, a_{h'} = a, \pi\right],$$

$$V_h^\pi(s, r) = \mathbb{E}\left[\sum_{h'=h}^{H} r_{h'}(s_{h'}, a_{h'}) \mid s_{h'} = s, \pi\right]. \tag{57}$$

Next, we start to construct high-probability event. We first construct an event that high-probability bounds occur for a single phase (i.e., a single execution of Algorithm 4), and then extend it to the case in which those bounds simultaneously hold for all phases.

**Lemma H.6.** *Suppose that Algorithm 4 is executed with input $(\delta, \Pi, T)$ where $\Pi \subseteq \Pi_{all}$. With probability at least $1 - \delta/5$, for all $(t, h, \pi, s, a) \in [T] \times [H] \times \Pi \times \mathcal{S} \times \mathcal{A}$:*

$$\left|\left[(\mathbb{P}_h - \widehat{\mathbb{P}}_{t,h})V_{h+1}^\pi\right](s, a)\right| \le H\sqrt{\frac{\log(10HSA|\Pi_{all}|T/\delta)}{2\max\{N_{t,h}(s, a), 1\}}}.$$

*Proof.* For $(s, a)$ has been visited, we apply Hoeffding's inequality and union bounds to complete the proof. For those $(s, a)$ that has not been visited yet, the claim trivially holds true. $\square$

**Lemma H.7.** *Suppose that Algorithm 4 is executed with input $(\delta, \Pi, T)$ where $\Pi \subseteq \Pi_{all}$. With probability at least $1 - \delta/5$, for all $(h, s, a, \pi, t) \in [H] \times \mathcal{S} \times \mathcal{A} \times \Pi \times [T]$,*

$$\left|\left[(\widehat{\mathbb{P}}_{t,h} - \mathbb{P}_h)\widehat{V}_{t,h+1}^\pi\right](s, a)\right| \le b_{t,h}(s, a).$$

*Proof.* For those $(s, a)$ that has not been visited yet, the claim trivially holds true. For $(s, a)$ has been visited, by Bernstein's inequality, with probability at least $1 - \delta'$,

$$\left|\left[(\widehat{\mathbb{P}}_{t,h} - \mathbb{P}_h)\widehat{V}_{t,h+1}^\pi\right](s, a)\right| \le \sum_{s' \in S}\left(\sqrt{\frac{2P(s'|s, a)\log(2/\delta')}{N_{t,h}(s, a)}} + \frac{2\log(2/\delta')}{3N_{t,h}(s, a)}\right)\widehat{V}_{t,h+1}^\pi(s')$$

$$\le \sum_{s' \in S} H\sqrt{\frac{2P(s'|s, a)\log(2/\delta')}{N_{t,h}(s, a)}} + \frac{2HS\log(2/\delta')}{3N_{t,h}(s, a)}$$

$$\le H\sqrt{\frac{2S\log(2/\delta')}{N_{t,h}(s, a)}} + \frac{2HS\log(2/\delta')}{3N_{t,h}(s, a)},$$

where the last inequality uses the Cauchy-Schwarz inequality. By taking a union bound, choosing a proper $\delta'$, and using the definition of $b_{t,h}(s, a)$, we complete the proof. $\square$

In the following high-probability bounds, we again assume that Algorithm 4 is executed with input $(\delta, \Pi, T)$ where $\Pi \subseteq \Pi_{all}$. By Azuma-Hoeffding's inequality and union bounds, with probability at least $1 - \delta/5$, for all $\pi \in \Pi$,

$$\mathbb{E}_{s_1 \sim \mu}\left[\sum_{t=1}^{T} V_1^\pi(s_1, b_t/H)\right] \le \sum_{t=1}^{T} V_1^\pi(s_{t,1}, b_t/H) + H\sqrt{8T\log(5|\Pi_{all}|/\delta)}. \tag{58}$$

For shorthand, we denote

$$\xi_{t,h} = \mathbb{P}_h\widehat{V}_{t,h+1}^{\pi_t}(s_{t,h}, a_{t,h}) - \widehat{V}_{t,h+1}^{\pi_t}(s_{t,h}). \tag{59}$$

By Azuma-Hoeffding's inequality and union bounds, with probability at least $1 - \delta/5$, for all $(t, h) \in [T] \times [H]$,

$$\sum_{t=1}^{T}\sum_{h=1}^{H-1} \xi_{t,h} \le H^2\sqrt{8T\log(5HT/\delta)}, \tag{60}$$

where $c_2 > 0$ is some absolute constant.

Again, by the fact that $|\Pi_{\mathtt{all}}| \geq |\Pi|$ and Azuma-Hoeffding's inequality and union bounds, with probability at least $1 - \delta_m/5$, for all $\pi \in \Pi$,

$$\left| \widetilde{V}^\pi - \mathbb{E}_{s_1 \sim \mu} \left[ \widehat{V}^\pi_{T,1}(s_1) \right] \right| \leq H \sqrt{8T \log \left( 10 |\Pi_{\mathtt{all}}|/\delta_m \right)} \tag{61}$$

**Definition H.8** (definition of $\widehat{\mathcal{E}}$, single phase). *Suppose that Algorithm 4 is executed with input $(\delta', \Pi, T)$ where $\Pi \subseteq \Pi_{\mathtt{all}}$. Let $\widehat{\mathcal{E}}$ be the event that all high probability bounds in Eq. (58), Eq. (60), and Eq. (61) hold simultaneously.*

Taking a union bound over those high-probability bounds, we have

$$\mathbb{P} \left( \widehat{\mathcal{E}} \right) \geq 1 - \delta'. \tag{62}$$

**Definition H.9** (definition of $\mathcal{E}$). *Let $\mathcal{E}$ be the event that when running Algorithm 3, all high probability bounds in Eq. (58), Eq. (60), and Eq. (61) hold for all phases $m \in \mathbb{N}$ simultaneously.*

Recall that Algorithm 3 runs Algorithm 4 in phases with input $(\delta_m, \Pi_m, T_m)$ where $\delta_m = \delta/(2m^2)$ and $\forall m \in \mathbb{N}, |\Pi_m| \leq |\Pi_{\mathtt{all}}|$ holds. By a union bound over all $m \in \mathbb{N}$, $\mathbb{P}(\mathcal{E}) \geq 1 - \delta$.

## H.3 Supporting Lemmas

Recall the reward-dependent value function given in Eq. (57), and we give the following lemmas.

**Lemma H.10.** *Suppose that Algorithm 4 is executed with input $(\delta', \Pi, T)$ where $\Pi \subseteq \Pi_{\mathtt{all}}$, and suppose $\widehat{\mathcal{E}}$ occurs. For all $(\pi, t, h, s) \in \Pi \times [T] \times [H] \times \mathcal{S}$, $V^\pi_h(s, b_t/H) \leq \widehat{V}^\pi_{h,t}(s)$ holds.*

*Proof.* Conditioning on $\widehat{\mathcal{E}}$, one can use Lemma H.6 and follow the same idea of Lemma 18 in [Azar et al., 2017] to complete the proof. $\qquad\square$

**Lemma H.11.** *Suppose that Algorithm 4 is executed with input $(\delta', \Pi, T)$ where $\Pi \subseteq \Pi_{\mathtt{all}}$, and suppose $\widehat{\mathcal{E}}$ occurs. For all policy $\pi \in \Pi$, the following holds.*

$$\forall \pi \in \Pi, \quad \left| \mathbb{E}_{s_1 \sim \mu} \left[ \widehat{V}^\pi_{T,1}(s_1) - V^\pi_1(s_1) \right] \right| \leq \mathbb{E}_{s_1 \sim \mu} \left[ V^\pi_1(s_1, b_T) \right].$$

*Proof.* One can show the following:

$$\mathbb{E}_{s_1 \sim \mu} \left[ \widehat{V}^\pi_{T,1}(s_1) - V^\pi_1(s_1) \right]$$

$$= \mathbb{E}_{s_1 \sim \mu} \left[ \widehat{Q}^\pi_{T,1}(s_1, \pi_1(s_1)) - Q^\pi_1(s_1, \pi_1(s_1)) \right]$$

$$\leq \mathbb{E}_{s_1 \sim \mu} \left[ [\widehat{\mathbb{P}}_{T,1} \widehat{V}_{T,2}](s_1, \pi_1(s_1)) - [\mathbb{P}_1 V_2](s_1, \pi_1(s_1)) \right]$$

$$\leq \mathbb{E}_{s_1 \sim \mu} \left[ b_{T,1}(s_1, \pi_1(s_1)) + [\mathbb{P}_1 \widehat{V}_{T,2}](s_1, \pi_1(s_1)) - [\mathbb{P}_1 V_2](s_1, \pi_1(s_1)) \right]$$

$$= \mathbb{E}_{s_1 \sim \mu, s_2 \sim \mathbb{P}_1(\cdot|s_1, \pi_1(s_1))} \left[ b_{T,1}(s_1, \pi_1(s_1)) + \widehat{V}_{T,2}(s_2) - V_2(s_2) \right]$$

$$\leq \cdots$$

$$\leq \mathbb{E}_{s_1 \sim \mu} \left[ V^\pi_1(s_1, b_T) \right],$$

where the second inequality follows from Lemma H.7.

Since those concentration bounds are also two-sided, the other side of desired claim can be similarly proved. Note that for the other side, one only need to consider $\widehat{Q}^\pi_{T,1}(s, a) \leq H$ for all $(s, a)$, and otherwise, the difference is negative, which implies that the claim trivially holds. $\qquad\square$

**Lemma H.12.** *Suppose that Algorithm 4 is executed with input $(\delta', \Pi, T)$ where $\Pi \subseteq \Pi_{\mathtt{all}}$, and suppose $\widehat{\mathcal{E}}$ occurs. For $\xi_{t,h}$ defined in Eq. (59), we have*

$$\sum_{t=1}^T \widehat{V}^{\pi_t}_{t,1}(s_{t,1}) \leq \sum_{t=1}^T \sum_{h=1}^{H-1} \xi_{t,h} + \sum_{t=1}^T \sum_{h=1}^{H-1} \left( 2 + \frac{1}{H} \right) b_{t,1}(s_{t,1}, a_{t,1}). \tag{63}$$

*Proof.* Conditioning on $\widehat{\mathcal{E}}$, we have

$$
\sum_{t=1}^{T} \widehat{V}_{t,1}^{\pi_t}(s_{t,1}) \le \sum_{t=1}^{T} \left( [\widehat{\mathbb{P}}_1^t \widehat{V}_{t,2}^{\pi_t}](s_{t,1}, a_{t,1}) + r_{t,1}(s_{t,1}, a_{t,1}) + b_{t,1}(s_{t,1}, a_{t,1}) \right)
$$

$$
= \sum_{t=1}^{T} \left( [\widehat{\mathbb{P}}_1^t \widehat{V}_{t,2}^{\pi_t}](s_{t,1}, \pi_t(s_{t,1})) + \left(1 + \frac{1}{H}\right) b_{t,1}(s_{t,1}, a_{t,1}) \right)
$$

$$
\le \sum_{t=1}^{T} \left( [\mathbb{P}_1 \widehat{V}_{t,2}^{\pi_t}](s_{t,1}, a_{t,1}) + \left(2 + \frac{1}{H}\right) b_{t,1}(s_{t,1}, a_{t,1}) \right)
$$

$$
\le \sum_{t=1}^{T} \left( \xi_{t,1} + \widehat{V}_{t,2}^{\pi_t}(s_{t,2}) + \left(2 + \frac{1}{H}\right) b_{t,1}(s_{t,1}, a_{t,1}) \right)
$$

$$
\le \cdots
$$

$$
\le \sum_{t=1}^{T} \sum_{h=1}^{H-1} \xi_{t,h} + \left(2 + \frac{1}{H}\right) \sum_{t=1}^{T} \sum_{h=1}^{H} b_{t,h}(s_{t,h}, a_{t,h}),
$$

where the first inequality holds due to Lemma H.7. $\qquad\square$

Then, we turn to bound two terms in Eq. (63). Note that Eq. (60) already gives the bound of the first term, and thus we only need to bound the second term. Before that, we first present an auxiliary lemma. The proof of this lemma can be found in [Jin et al., 2020, Lemma 10]

**Lemma H.13.** *Suppose that Algorithm 4 is executed with input* $(\delta', \Pi, T)$ *where* $\Pi \subseteq \Pi_{\mathtt{all}}$*. For all* $h \in [H]$*, the followings hold.*

$$
\sum_{(s,a) \in \mathcal{S} \times \mathcal{A}} \sum_{t=1}^{T} \frac{\mathbb{I}\{(s_{t,h}, a_{t,h}) = (s,a)\}}{\max\{1, N_{t,h}(s,a)\}} = \mathcal{O}\left(SA \log T\right),
$$

$$
\sum_{(s,a) \in \mathcal{S} \times \mathcal{A}} \sum_{t=1}^{T} \frac{\mathbb{I}\{(s_{t,h}, a_{t,h}) = (s,a)\}}{\sqrt{\max\{1, N_{t,h}(s,a)\}}} = \mathcal{O}\left(\sqrt{SAT}\right).
$$

With the above lemma in hand, one can show:

$$
\sum_{t=1}^{T} \sum_{h=1}^{H} b_{t,h}(s_{t,h}, a_{t,h})
$$

$$
= \sum_{h=1}^{H} \sum_{t=1}^{T} \left( H \sqrt{\frac{2S \log \iota}{\max\{1, N_{t,h}(s_{t,h}, a_{t,h})\}}} + \frac{2HS \log \iota}{3 \max\{1, N_{t,h}(s_{t,h}, a_{t,h})\}} \right)
$$

$$
= \sum_{h=1}^{H} \sum_{(s,a) \in \mathcal{S} \times \mathcal{A}} \sum_{t=1}^{T} \left( H\sqrt{2S \log \iota} \frac{\mathbb{I}\{(s_{t,h}, a_{t,h}) = (s,a)\}}{\sqrt{\max\{1, N_{t,h}(s_{t,h}, a_{t,h})\}}} + \frac{2HS\mathbb{I}\{(s_{t,h}, a_{t,h}) = (s,a)\} \log \iota}{3 \max\{1, N_{t,h}(s_{t,h}, a_{t,h})\}} \right)
$$

$$
= \mathcal{O}\left(SH^2 \sqrt{AT \log \iota} + HS^2 A \log(T) \log \iota\right),
$$

where the last step holds due to Lemma H.13.

Therefore, we have

$$
\sum_{t=1}^{T} \widehat{V}_{t,1}^{\pi_t}(s_{t,1}) = \mathcal{O}\left(SH^2 \sqrt{AT \log \iota} + HS^2 A \log(T) \log \iota\right). \tag{64}
$$

## H.4   Proof of Proposition H.1

We first consider a single execution of Algorithm 4 with input $(\delta, \Pi, T)$ where $\Pi \subseteq \Pi_{\mathtt{all}}$. With the above supporting results, we are now ready to prove the claimed result. The following proof will

condition on event $\mathcal{E}$. First, one can show that for all $\pi \in \Pi$:

$$
\left| \widetilde{V}^{\pi} - \mathbb{E}_{s_1 \sim \mu} \left[ V_1^{\pi}(s_1) \right] \right|
$$

$$
\leq \left| \widetilde{V}^{\pi} - \mathbb{E}_{s_1 \sim \mu} \left[ \widehat{V}_{T,1}^{\pi}(s_1) \right] \right| + \left| \mathbb{E}_{s_1 \sim \mu} \left[ \widehat{V}_{T,1}^{\pi}(s_1) \right] - \mathbb{E}_{s_1 \sim \mu} \left[ V_1^{\pi}(s_1) \right] \right|
$$

$$
\leq H \sqrt{8T \log(10|\Pi_{\mathtt{all}}|/\delta)} + \left| \mathbb{E}_{s_1 \sim \mu} \left[ \widehat{V}_{T,1}^{\pi}(s_1) \right] - \mathbb{E}_{s_1 \sim \mu} \left[ V_1^{\pi}(s_1) \right] \right|. \qquad \text{(By Eq. (61))}
$$

Then, we turn to bound the second term. Recall the reward dependent value function defined in Eq. (57), and we have

$$
\left| \mathbb{E}_{s_1 \sim \mu} \left[ \widehat{V}_{T,1}^{\pi}(s_1) - V_1^{\pi}(s_1) \right] \right| \leq \mathbb{E}_{s_1 \sim \mu} \left[ V_1^{\pi}(s_1, b_T) \right] \qquad \text{(By Lemma H.11)}
$$

$$
= H \mathbb{E}_{s_1 \sim \mu} \left[ V_1^{\pi}(s_1, b_T/H) \right]
$$

$$
\leq \frac{H}{T} \mathbb{E}_{s_1 \sim \mu} \left[ \sum_{t=1}^{T} V_1^{\pi}(s_1, b_t/H) \right]. \qquad (b_T \leq b_t, \forall t)
$$

From Eq. (58), we have

$$
\mathbb{E}_{s_1 \sim \mu} \left[ \sum_{t=1}^{T} V_1^{\pi}(s_1, b_t/H) \right] \leq \sum_{t=1}^{T} V_1^{\pi}(s_{t,1}, b_t/H) + c_1 H \sqrt{T \log \iota}
$$

$$
\leq \sum_{t=1}^{T} \widehat{V}_{t,1}^{\pi}(s_{t,1}) + c_1 H \sqrt{T \log \iota} \qquad \text{(By Lemma H.10)}
$$

$$
\leq \sum_{t=1}^{T} \widehat{V}_{t,1}^{\pi_t}(s_{t,1}) + c_1 H \sqrt{T \log \iota}. \qquad (\pi_t \in \mathrm{argmax}_{\pi \in \Pi} \widehat{V}_{t,1}^{\pi}(s_{t,1}))
$$

Combining the above with Eq. (64), we arrive at

$$
\mathbb{E}_{s_1 \sim \mu} \left[ \sum_{t=1}^{T} V_1^{\pi}(s_1, b_t/H) \right] = \mathcal{O} \left( SH^2 \sqrt{\frac{A \log \iota}{T}} + \frac{HS^2 A \log^2 \iota}{T} \right).
$$

Now, we consider a fixed phase $m$ where the input of Algorithm 4 is $(\delta_m, \Pi_m, T_m)$. There always exist two absolute constants $c_1, c_2 > 0$ for $T_m = c_1 2^{-2m} S^2 A H^4 \log^2 \left( c_2 2^{-2m} S^2 A H^4 |\Pi_{\mathtt{all}}| \delta^{-1} \right)$ such that

$$
\left| \widetilde{V}_m^{\pi} - \mathbb{E}_{s_1 \sim \mu} \left[ V_1^{\pi}(s_1) \right] \right| \leq \frac{2^{-m}}{2}.
$$

As conditioning on event $\mathcal{E}$, the above holds for all $m \in \mathbb{N}$. Thus, the proof is complete.

# I   Limitations

One limitation is that the regret bound implied by the ULI guarantee for linear bandits with infinitely-many arms is suboptimal, i.e., $\widetilde{\mathcal{O}}(d^{1.5}\sqrt{T})$ compared with lower bound $\Theta(d\sqrt{T})$ but note that this problem is long-standing open. This is discussed below Theorem 4.2. We also mention the limitation of Algorithm 3 (below Theorem 5.1) that it is computational inefficient and the regret bound implied by ULI guarantee is also suboptimal.

# J   Broader Impacts

This paper proposes a new metric for online RL, called uniform last-iterate, which could be helpful for high-stacks applications, since a near-optimal result under our metric ensures both good cumulative and instantaneous results.

