# OpenReview forum: "Uniform Last-Iterate Guarantee for Bandits and Reinforcement Learning"
_NeurIPS.cc/2024/Conference — NeurIPS 2024 poster_

### Official Review · Reviewer_rjuw · 2024-06-20

**Soundness:** 3
**Presentation:** 3
**Contribution:** 2
**Rating:** 6
**Confidence:** 4

**Summary:**

The paper proposes a new metric for sample efficiency in online learning called Uniform Last Iterate (ULI), show that it is stronger than existing metrics, and gives algorithms that achieve near-optimal ULI.

**Strengths:**

1. The paper proposes a new metric for online learning that characterizes not only the cumulative performance but also the instantaneous performance of the learner, which might be important in high-risk fields. The authors show that the proposed metric, called ULI, captures instantaneous performance and is strictly stronger than existing metrics: regret and uniform-PAC. This shows that their new metric is indeed valuable.
2. The authors then present algorithms that achieve near-optimal ULI in multi-armed bandit, linear bandits and tabular MDPs. The main insight is that algorithms based on action elimination achieve near-optimal ULI, while optimistic algorithms are proved to not achieve near-optimal ULI. This separation between optimism and action elimination is an interesting side effect.
3. The paper is very clearly written. Presenting a new concept is not easy, and the authors do a good job in explaining it, its motivation and its differences from existing metrics. The algorithms and proofs in the appendix are also easy to follow.

**Weaknesses:**

1. Although I quite like the definition of this new metric, the potential impact of this paper seems low. The insight about action elimination algorithms being better than optimistic algorithms is nice, but we have seen other cases where this is true and optimistic algorithms are not used in practice anyways. Unfortunately, I believe that the paper cannot get a higher score because of this reason. I would be happy to hear what the authors believe could be the future works that build on this paper.
2. While the paper and the appendix are well-written, I think that the structure of the paper could greatly improve so that the important things appear in the main text. There are many theorems but not one proof sketch. This is a shame because many of the claims are nice and simple to explain even in just two sentences. I think this would give much better intuition to the readers regarding the logic behind this new metric. In contrast, using a full page to describe an RL algorithm which is mostly standard seems like a huge waste of space. I do not feel like I gained any insight from the RL section (section 5), after reading the previous sections on bandits.
3. The RL algorithm is disappointing because it is not computationally efficient. For me, this means that the authors did not prove that near-optimal ULI can be achieved in MDPs. The result is over claiming of the contributions in the introduction, especially since near-optimal computationally efficient algorithms based on action elimination already exist in tabular MDPs. Is there a reason they do not achieve near-optimal ULI?

**Questions:**

see weaknesses

**Limitations:**

yes

---

> ### Author Rebuttal · Authors · 2024-08-07
>
> We thank the reviewer for the valuable feedback. Below we address your concerns.
>
> ---
> **Q1:** What the authors believe could be the future works that build on this paper?
>
> **A1:** We provide three future directions below and will include the discussion in our final version.
>
> - It could be interesting to investigate some empirical issues when deploying ULI algorithms in high-stakes fields.
> Typically, optimistic algorithms initially incur a lower regret than ULI algorithms, but their regret will exceed those of ULI algorithms at a certain juncture. At this juncture, ULI algorithms have eliminated all suboptimal arms, whereas optimistic algorithms continue exploring as time evolves. Hence, identifying this turning point could be beneficial for deploying ULI algorithms in high-stakes domains.
>
> - Design (computationally efficient) algorithms for MDPs with linear function approximation. The main challenging is to bypass any dependence on the number of states which is possibly infinite. Thus, generalizing our RL algorithm, which enumerates all deterministic policies, to the linear settings does not work.
>
>
> - Design (computationally efficient) algorithms for Episodic MDPs with only logarithmic dependence on $H$ (a.k.a. horizon-free). In our attempts, we use doubling trick on $\epsilon$ in existing $(\delta,\epsilon)$-PAC horizon-free RL algorithms. In this case, we run the algorithm in phases with $\epsilon,\epsilon/2,\epsilon/4,\ldots$. The main difficulty is to leverage the information learned from the previous phase to guide the algorithm to play an improved policy at the next phase as required by ULI.
> Thus, we conjecture that there might exist fundamental barriers to simultaneously achieve ULI guarantee and a logarithmic dependence on $H$.
>
> ---
>
>
> **Q2:** The structure and organization issues of the paper in the main text.
>
> **A2:** Thanks for this suggestion.
> We will use the extra page granted in the final version to improve the main text presentation.
>
>
>
> ---
> **Q3:** The RL algorithm is disappointing because it is not computationally efficient. For me, this means that the authors did not prove that near-optimal ULI can be achieved in MDPs. The result is over claiming of the contributions in the introduction, especially since near-optimal computationally efficient algorithms based on action elimination already exist in tabular MDPs. Is there a reason they do not achieve near-optimal ULI?
>
> **A3:** Great question.
> ULI condition requires algorithms to play increasingly better policies as time evolves.
> For action-elimination based algorithms, they typically focus on the cumulative performance such as regret, but does not control the single-round error that ULI requires.
> As a result, it is entirely possible for the algorithm to take a policy considerably worse after an action is eliminated.
> Therefore, we propose a more direct approach based on policy-elimination to achieve ULI.
> We left the design of an efficient RL algorithm with ULI guarantee as our future work.

---

> > ### Comment · Reviewer_rjuw · 2024-08-13
> >
> > Thank you for the response, I will keep my positive score.

---

### Official Review · Reviewer_upT1 · 2024-07-08

**Soundness:** 3
**Presentation:** 3
**Contribution:** 3
**Rating:** 6
**Confidence:** 3

**Summary:**

In this paper, the authors study algorithms with better performance metrics for both bandits and reinforcement learning. They propose a new metric, namely the uniform last-iterate guarantee, generalizing uniform-PAC, which can further ensure the last-iterate performance of the algorithm. The authors present algorithms achieving this guarantee in different scenarios. In the multi-armed bandits setting, the authors propose an elimination-based meta algorithm that can achieve a near-optimal uniform last-iterate guarantee. Together with the adaptive barycentric spanner technique, they then show that the algorithm can be extended to linear bandits with possibly infinite arms. Finally, the authors generalize their algorithm to the tabular Markov decision process, achieving a uniform last-iterate guarantee with near-optimal factors.

**Strengths:**

- The paper is generally well-written and easy to follow.

- The paper discusses a new metric for online learning algorithms. It is shown that the new uniform last-iterate guarantee is strictly stronger than the uniform-PAC guarantee, which I think is an interesting concept for study.

- The paper presents a meta-algorithm that translates the uniform last-iterate guarantee into adversarial bandits. This adaptation highlights significant implications for the new metric, which are compelling and merit further exploration.

**Weaknesses:**

- The reinforcement learning algorithm presented in the paper have to examine every policy, which is generally not computationally efficient.

**Questions:**

- Could the authors discuss potential barriers that might prevent the phase-based algorithm (He et al., 2021) from achieving a uniform last-iterate guarantee in linear bandits?

He, Jiafan, Dongruo Zhou, and Quanquan Gu. "Uniform-pac bounds for reinforcement learning with linear function approximation." Advances in Neural Information Processing Systems 34 (2021): 14188-14199.

**Limitations:**

- The authors have thoroughly discussed the limitations of their work.

---

> ### Author Rebuttal · Authors · 2024-08-07
>
> We thank the reviewer for the valuable feedback. Below we address your concern.
>
> ---
> **Q:** Could the authors discuss potential barriers that might prevent the phase-based algorithm (He et al., 2021) from achieving a uniform last-iterate guarantee in linear bandits?
>
> **A:**
> The main barriers for UPAC-OFUL (He et al., 2021) to achieve ULI is the optimistic arm selection rule. Specifically, UPAC-OFUL runs OFUL, an optimistic-based algorithm, in a multi-layer fashion.
> However, the arm selection rule established upon optimism, explores the bad arms unevenly across time steps. Consequently, it might play a significantly bad arm for some very large but finite $t$. The ULI guarantee, on the contrary, requires the algorithm to intensively conduct the exploration at the early stage.

---

### Official Review · Reviewer_vbbn · 2024-07-12

**Soundness:** 2
**Presentation:** 2
**Contribution:** 2
**Rating:** 5
**Confidence:** 2

**Summary:**

This paper introduces a stronger metric, uniform last-iterate (ULI) guarantee.

The authors demonstrate that a near-optimal ULI guarantee directly implies near-optimal cumulative performance across traditional metrics such as regret and PAC-bounds, but not the other way around.

The authors first provide two results for bandit problems with finite arms, showing that some algorithms can attain near-optimal ULI guarantees.

They also provide a negative result, indicating that optimistic algorithms cannot achieve a near-optimal ULI guarantee.

Finally, they propose other interesting algorithms that achieve a near-optimal ULI guarantee for linear bandits with infinitely many arms (given access to an optimization oracle) and the online reinforcement learning setting.

**Strengths:**

- Interesting new metric: Uniform last-iterate, which is a stricter version of uniform-PAC.

- Interesting inclusion result:
    - ULI includes uniform-PAC but the converse is not true.
    - Also, any optimism-based algorithms cannot satisfy the ULI standard.
    - ULI algorithms can still achieve tight minimax regret for many bandit instances (Section 3.1)

**Weaknesses:**

- Necessity of the ULI itself: basically this notation forces the learner to choose less explorative options. However, in my intuition, the only way to do less exploration later is to choose more sub-optimal arms often at the early stage where ULI is less restrictive. In your patient example in the Introduction section, does it make any major ethical difference?
    - As somewhat expected, an algorithm with ULI standard was not able to achieve the minimax optimal regret on the general linear bandit (with infinitely many arms) since the criterion prevents the algorithm from exploring, while in linear bandit it is important to explore each direction enough. Still, admitting the weakness shows honesty in their report.

- Clarity: As a first reader, it is not easy for me to understand why the $\log (t)$ term in the optimism-based algorithms guarantees that they fail the ULI standard. They said the log term makes the algorithm choose sub-optimal arms infinitely often, but does that directly induce that it does not satisfy the ULI condition? It would be great if authors could add a theorem for it, besides the lil'UCB result.

- They cannot achieve anytime sublinear expected regret, since their result is stricter than uniform-PAC.

**Questions:**

- What about the instance-dependent bound? UCB is strong since it also guarantees good instance-dependent bound. I guess most of the results in this paper are written in a minimax manner. It would be great if authors could provide the followings
    - Result of instance-dependent regret, such as the counterpart of Thm 2.6
    - Example of algorithms with optimal instance-dependent regret and satisfying ULI condition. If this is difficult, it would be great if authors could make an instance-dependent analysis for the algorithms in section 3.1.

- I want to ask for additional theorems about the impossibility of optimism-based algorithms, see the weakness section above.

- Necessity of the ULI: check the weakness section above. I somewhat feel ULI as 'unnecessarily strict' criterion.

**Limitations:**

The authors state their limitation in their paper.

No negative societal impact

---

> ### Author Rebuttal · Authors · 2024-08-07
>
> We thank the reviewer for the valuable feedback. Below we address your concerns.
>
> ---
> **Q1:** Necessity of the ULI itself. In your patient example in the Introduction section, does it make any major ethical difference?
>
> **A1:** Indeed, the ULI forces the algorithm to explore more in the beginning. Hence, it implies the necessity in safety-critical examples. Regarding the ethical difference: early-phase clinical trials could be conducted on animals, where ethical concerns are less stringent compared to later phases involving humans. Therefore, ULI is a desirable metric for safety-critical applications where playing a bad arm/policy at a late stage could lead to catastrophic consequences.
>
> ---
> **Q2:** Why the $\log t$ term in the optimism-based algorithms guarantees that they fail the ULI standard? Add additional theorems about the impossibility of optimism-based algorithms.
>
> **A2:** At a high-level, $\log t$ is increasing with time and forces the algorithm to play a bad arm indefinitely when $t$ evolves.
> We provide the formal proof in the following and will include it in our final version:
>
>
> *Proof.* We prove the claim by contradiction. Consider a $K$-armed bandit instance with at least one suboptimal arm, and let $\Delta>0$ be the minimum arm gap. Suppose there exists an optimism-based algorithm with $\log t$ in bonus term that can achieve the ULI guarantee in this setting. Then, for some fixed $\delta \in (0,1)$, with probability $\geq 1-\delta$, for all $t \in \mathbb{N}$, $\Delta_t \leq F_{ULI}(\delta,t)$. Based on Definition 2.5, we have $\lim_{t\to \infty}F_{ULI}(\delta,t)=0$ and $F_{ULI}(\delta,t)$ is monotonically decreasing w.r.t. $t$ after a threshold. Thus, $\exists t_0 \in \mathbb{N}$ such that $F_{ULI}(\delta,t) <\Delta$ for all $t \geq t_0$. In other words, the algorithm cannot play any suboptimal arm after $t_0$-th round. Recall that the bonus term is $\sqrt{\log t/N_a(t)}$ where $N_a(t)$ is the number of plays of arm $a$ before round $t$. For any suboptimal arm $a$, $N_a(t)$ should not increase after $t_0$-th round, but $\log t$ keeps increasing. This leads the bonus of arm $a$ goes to infinity, which will incur a play of arm $a$ at a round after $t_0$. This makes a contradiction.
>
> ---
> **Q3:** Result of instance-dependent regret, such as the counterpart of Thm 2.6.
>
> **A3:** All ULI results can be directly translated into instance-dependent regret bounds by invoking Theorem 2.7. We take the ULI result of SE-MAB in Theorem 3.2 as an example. According to ULI result of SE-MAB in Theorem 3.2, we have $F_{ULI}(\delta,t)=\text{polylog}(t/\delta)t^{-\kappa}=t^{-\frac{1}{2}}\sqrt{K\log(\delta^{-1}Kt)}$ with $\kappa=1/2$ in Theorem 2.7 and for all $T \in \mathbb{N}$, $R_T = O(\Delta^{-1}K \log(K\delta^{-1} \Delta^{-1}))$. Other ULI results in our paper can be similarly translated.
>
> ---
> **Q4:** Example of algorithms with optimal instance-dependent regret and satisfying ULI condition. If this is difficult, could authors
>  make an instance-dependent analysis for the algorithms in section 3.1?
>
> **A4:** It is unclear if an algorithm can simultaneously achieve optimal instance-dependent regret and ULI guarantee, but we note that the regret bounds of algorithms in Section 3.1 are near-optimal via a instance-dependent analysis provided below. The main difficulty is that the algorithm (e.g., optimism-based) with the optimal regret bound should mix exploration and exploitation, whereas ULI requires the algorithm (elimination-based) to explore more in the beginning, and [1] show that elimination-based algorithms cannot be optimal.
> It is interesting to study if there exists a separation between optimal regret and optimal ULI for future work.
>
> [1] On explore-then-commit strategies, NIPS, 2016.
>
> Below, we provide the formal instance dependent analysis for the algorithms. One can directly invoke Theorem 2.7 to get an instance-dependent regret bound in the form of $\widetilde{O}(K\Delta^{-1})$ (where $\Delta$ is the minimum gap) or making an instance-dependent analysis for specific algorithms to get a tighter bound as $\widetilde{O}(\sum_{a:\Delta_a>0} \Delta_a^{-1})$. The analysis is given in the following.
>
> **SE-MAB.** Recall that $N_a(t)$ is the number of plays of arm $a$ before $t$. With probability $\geq 1-\delta$, for all $t \in \mathbb{N}$, $R_t =\sum_{a:\Delta_a>0} \Delta_a N_a(t+1) = O(\sum_{a:\Delta_a>0}  \frac{\log(Kt/\delta)}{\Delta_a} )$ where the bound of $N_a(t)$ is given in Lemma D.4. Recall that $T_a$ is an upper bound for the number of plays of suboptimal arm $a$ until getting eliminated. By solving Eq. (12) in Lemma D.4, one can also derive a high-probability anytime bounded regret $R_t =\sum_{a:\Delta_a>0} \Delta_a T_a = O(\sum_{a:\Delta_a>0}  \frac{\log(K\Delta_a^{-1}\delta^{-1})}{\Delta_a} )$. Combining both bounds, we have
> $$\mathbb{P} \left( \forall t:R_t=O(\sum_{a:\Delta_a>0} \frac{\log(K\min\{\Delta_a^{-1},t\}\delta^{-1})}{\Delta_a}
> )\right) \geq 1-\delta.$$
>
> **PE-MAB.**
> Recall from Algorithm 7 (see Appendix D.2) that PE-MAB runs in phases $s=1,2,\ldots$ and in each phase $s$, it plays every arm $a$ in active arm set for $m_{s}=\lceil  2^{2s+1}\log(4Ks^2 \delta^{-1})  \rceil$ times. Let $s(t)$ be the phase that round $t$ lies in and we denote $s_a$ be the last phase that suboptimal arm $a$ survive. Thus, with probability $\geq 1-\delta$, for all $t \in \mathbb{N}$ ($\lesssim$ hides constant),
>
> $$R_t  = \sum_{a:\Delta_a>0} \Delta_a \sum_{s=1}^{  \min\\{s(t),s_a\\}  } m_{s} \lesssim \sum_{a:\Delta_a>0} \Delta_a \cdot \log(K\delta^{-1} (\min\\{s(t),s_a\\})^2 ) \sum_{s=1}^{  s_a  } 2^{2s+1} \lesssim \sum_{a:\Delta_a>0} \frac{\log(K\delta^{-1} \min\\{\log(t),\log(\Delta_a^{-1})\\} ) }{\Delta_a}$$
>
> where the last inequality uses $s(t) \leq \log_2(t+1)$ (Lemma D.9) and $\frac{\Delta_a}{2} \leq 2^{-(s_a-1)}$ (Lemma D.8).
>
> **PE-L.** As this algorithm is also phased-based, the instance-dependent regret bound can be derived by using a similar argument of PE-MAB, as shown above.

---

> > ### Comment · Reviewer_vbbn · 2024-08-12
> >
> > Thanks for your detailed rebuttal. I will change my score from 4 to 5.

---

### Official Review · Reviewer_HueZ · 2024-07-15

**Soundness:** 3
**Presentation:** 3
**Contribution:** 3
**Rating:** 6
**Confidence:** 3

**Summary:**

This paper introduces a new form of guarantee for MAB and RL algorithms named "Uniform Last Iterate (ULI)". Just like the uniform PAC guarantee introduced in Dann Lattimore and Brunskill (2017), it unifies the traditional sublinear-regret (with optimal rates) and PAC guarantees for such algorithms. However, unlike uniform-PAC, ULI guarantees are also shown to encapsulate the instantaneous perfromance of MAB/RL algorithms, ensuring that, with high probability, a very suboptimal arm/action is _never_ taken after large enough $t$ (with probability $\geq 1 - \delta$, the maximum suboptimality after round $t$ is bounded by a decreasing function of $t$).

* They show that ULI is strictly stronger than uniform PAC (and hence also unifies regret and PAC).
* They show that successive elimination (SE) and phased elimination (PE) algorithms achieve ULI for finite-arm and linear bandits (in the case of PE).
* They show that well-known optimistic algorithms for stochastic bandits do not achieve ULI, and that even some modifications (lil'UCB) do not achieve near-optimal-rate ULI.
* They show that a phased-meta algorithm used with well-known adversarial bandit algorithms such as EXP3.P can achieve ULI (Appendix F).
* They show that phased-policy-elimination with a modified version of UCB-VI can achieve ULI for tabular episodic MDPs.
* All the ULI guarantees are near-optimal (the ULI bound decays as $t^{-1/2}$) which leads to near optimal regret bounds $\tilde{O}(\sqrt{T})$,

**Strengths:**

* The ULI guarantee is a novel and interesting extension of existing work which unifies a number of known-guarantees applied to online bandit/RL algorithms (regret, PAC, uniform PAC).
* ULI encapsulates instantaneous performance in a way that existing guarantees do not, which is of interest in safety-critical applications.
* The ULI guarantee is very _applicable_ in the sense that existing well-known algorithms achieve it with mild conditions/modifications, and the authors successfully demonstrate this in the paper.

**Weaknesses:**

* The regret bounds obtained by ULI have the optimal dependency on $T$ (i.e. $\sqrt{T}$) but suboptimal dependence on other parameters (such as $d$ for linear bandits).
* The algorithm for episodic tabular RL is computationally inefficient (both limitations are acknowledged and drawn attention to by the authors in the paper, and are fairly minor considering the contribution).
* There is no experimental evaluation for the modified algorithms compared to the baseline algorithms.

**Questions:**

* Can you comment on where the UBEV algorithm of (Dann et al 2017) with the uniform PAC guarantee fails w.r.t the ULI guarantee (ULI even if not with near optimal rates)?

**Limitations:**

The limitations have been addressed adequately by the authors. No negative social impact.

---

> ### Author Rebuttal · Authors · 2024-08-07
>
> We thank the reviewer for the valuable feedback. Below we address your concern.
>
> ---
> **Q:** Can you comment on where the UBEV algorithm of (Dann et al 2017) with the uniform PAC guarantee fails w.r.t the ULI guarantee (ULI even if not with near optimal rates)?
>
> **A:** In fact, the UBEV algorithm is the RL version of lil’UCB, as both utilize the law of the iterated logarithm (LIL) to avoid the $\log t$ term in the bonus. When reducing UBEV to the MAB setting, it operates in a similar manner to lil’UCB. From Theorem 3.3, we know that lil’UCB (and thus UBEV) cannot be near-optimal ULI. However, it remains unclear whether lil’UCB (and UBEV) is ULI, as discussed in the text below Theorem 3.3.

---

### Decision · Program_Chairs · 2024-09-25

**Decision:**

Accept (poster)

**Comment:**

Overall a good paper with a stronger notion (than standard regret) proposed and characterized. Reviewers overall appreciated the work.